

# UK greenhouse gas measurements at two new tall towers for aiding emissions verification

Ann R. Stavert[1,2], Simon O'Doherty[2], Kieran Stanley[2], Dickon Young[2], Alistair J. Manning[3], Mark F. Lunt[5], Christopher Rennick[4] and Tim Arnold[4,5].

[1] Now at Climate Science Centre, CSIRO Oceans & Atmosphere, Aspendale, VIC, 3195, Australia
[2] School of Chemistry, University of Bristol, Bristol, BS8 1TS, United Kingdom
[3] Met Office, Exeter, Devon, EX1 3PB, United Kingdom
[4] National Physical Laboratory, Teddington, Middlesex, TW11 0LW, United Kingdom
[5] School of GeoSciences, University of Edinburgh, Edinburgh, EH9 3FF, UK

*Correspondence to*: Ann R. Stavert (ann.stavert@csiro.au)

**Abstract.** Under the UK focused Greenhouse gAs and Uk and Global Emissions (GAUGE) project, two new tall tower greenhouse gas (GHG) observation sites were established in the 2013/2014 Northern

Hemispheric winter. These sites were located at two existing telecommunications towers, Heathfield (HFD) and Bilsdale (BSD), utilised a combination of cavity ring-down spectroscopy (CRDS) and gas chromatography (GC) to measure key GHGs ($CO_2$, $CH_4$, CO, $N_2O$ and $SF_6$). Measurements were made at multiple intake heights on each tower. The inclusion of the two additional tower stations within the existing UK Deriving Emissions linked to Climate Change (DECC) network of four stations was found

to reduce the uncertainty of $CH_4$ UK emission estimates by between 10-20 %. $CO_2$ and $CH_4$ dry mole fractions were calculated from either CRDS measurements of wet air which were post corrected with an instrument specific empirical correction or samples dried to between 0.05 and 0.3 % $H_2O$ using a Nafion dryer, with a smaller correction applied for the residual $H_2O$. The impact of these two drying strategies was examined. Drying with a Nafion drier was not found to have a significant effect on the

observed $CH_4$ mole fraction; however, Nafion drying did cause a 0.02 $\mu$mol $mol^{-1}$ $CO_2$ bias. This bias was stable with sample $CO_2$ mole fractions between 373 and 514 $\mu$mol $mol^{-1}$ and for sample $H_2O$ up to 3.5 %. As the calibration and standard gases are treated in the same manner, this error is mostly calibrated out with the residual error below the World Meteorological Organization's (WMO) reproducibility requirements. Of more concern was the error associated with both default factory and

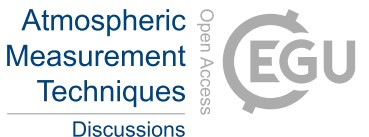
empirical instrument specific water correction algorithms. These corrections are relatively stable and reproducible for samples with $H_2O$ between 0.2 and 2.5 %, $CO_2$ between 345 and 449 µmol mol$^{-1}$ and $CH_4$ between 1743 and 2145 nmol mol$^{-1}$. However, the residual errors in these corrections increase to > 0.05 µmol mol$^{-1}$ for $CO_2$ and > 1 nmol mol$^{-1}$ for $CH_4$ (greater than the WMO internal reproducibility

guidelines) at higher humidities and for samples with very low (< 0.5 %) water content. These errors were also found to scale with the absolute magnitude of the $CO_2$ and $CH_4$ mole fraction. As such, water corrections calculated in this manner are not suitable for samples with low (< 0.5 %) or high (> 2.5 %) water contents and either alternative correction methods should be used or partial drying or humidification considered prior to sample analysis.

## 1 Introduction

The adverse effects of anthropogenically driven increases of greenhouse gas concentrations on global temperatures and climate have been well established (IPCC, 2013). Governmental efforts to curb these emissions include the UK 2008 Climate Change Act, which requires the UK to decrease its GHG

emissions by 80 % of 1990 levels by 2050 (Parliment of the United Kingdom, 2008 Chapter 27). This in turn motivated the Greenhouse gAs Uk and Global Emissons (GAUGE) project, which aimed to better quantify the UK $CO_2$, $CH_4$ and $N_2O$ emissions. These new emission estimates would then be used to assess the impact of emission abatement and reduction strategies. Key to the GAUGE project was combining new and existing GHG data streams, including high-density regional observation studies, tall

tower sites, moving platforms (ferry and aircraft) and satellite observations, with innovative modelling approaches.

This paper is divided into two sections. Firstly, it describes the establishment of two new UK GHG tall tower (TT) sites funded under the GAUGE project, provides an analysis of the observations made at the sites and outlines additional insights that these new observations provide when quantifying UK GHG

emissions and trends. Secondly, this paper investigates the error associated with empirical instrument specific water correction algorithms and the sample drying approach used at these TT sites. A further paper (Stanley, Stavert et al. in preparation) will discuss the integration of these new sites with the





existing UK Deriving Emissions linked to Climate Change (DECC) network (Stanley et al., 2018) funded by the UK Department of Business, Energy and Industrial Strategy (BEIS) and provide a full uncertainty analysis for data collected at all the DECC/GAUGE sites. While, a companion paper (Stavert, Stanley et al., in preparation) will discuss the integration and inter-calibration of all the $CO_2$,

$CH_4$, CO, $N_2O$ and $SF_6$ data streams including near surface, tall tower, ferry and aircraft measurements along with an analysis of the impact of identified site biases on UK GHG emission estimates.

Like the UK DECC network, the new sites, Bilsdale (BSD) and Heathfield (HFD), are equipped with a combination of cavity ring-down spectrometer (CRDS) and gas chromatograph (GC) instrumentation (Stanley et al., 2018). These instruments, along with the associated calibration gases (linked to

recognised World Meteorological Organization calibration scales) and automated sampling systems are located at the base of telecommunication towers within the UK. Further details of the sites and instruments used along with a description of the data collected to date are provided in the first part of this paper. As an example of the impact of these two new sites in terms of UK GHG emissions estimates, inversion based estimates of UK $CH_4$ emissions and their associated uncertainties, with and

without the new TT sites, are also included.

Recent advances in tuneable diode lasers used within CRDS instrumentation has led to a dramatic reduction in the cost of these instruments. This, along with their precision, stability, relative autonomy and robustness, has led to a rapid increase in their deployment in global, continental and regional GHG monitoring networks including the GAUGE network, the European Integrated Carbon Observing

System (ICOS) (Yver Kwok et al., 2015) and the Indianapolis Flux Experiment (INFLUX) (Turnbull et al., 2015). These instruments also claim the advantage of being able to measure un-dried ("wet") air samples which are then post corrected to "dry" values using an inbuilt algorithm (Rella, 2010).

Initially, it was hoped that the inbuilt $H_2O$ correction would remove the need for sample drying, inherent in most other methods (e.g. FTIR or NDIR) but subsequent studies have questioned its stability

over time and between instruments (Yver Kwok et al., 2015;Chen et al., 2010;Winderlich et al., 2010). In response to this, researchers have typically developed their own water corrections or have returned to sample drying in order to minimise the effect (Welp et al., 2013;Winderlich et al., 2010;Schibig et al.,





2015). As such the examination of any errors or biases induced by drying and water correction methods is essential for fully quantifying the uncertainty of CRDS measurements.

The CRDS instrumentation deployed at GAUGE and UK DECC Network sites initially relied on drying the sample with a Nafion water permeable membrane in combination with dry zero air as a counter

purge gas. Here, due to the moisture gradient between the sample and the counter purge, the water passed from the wet sample through the membrane to the dry counter purge. Drying in this manner has a history of successful application for the measurements of halocarbons (Foulger and Simmonds, 1979), $N_2O$ (Prinn et al., 1990) and $SF_6$ (Fraser et al., 2004). However, studies have found that $CO_2$ and $CH_4$ can also pass across a dry Nafion membrane (Chiou and Paul, 1988) and that this transport increases

with the water saturation of the membrane (Naudy et al., 2014). As the transport process is driven by a partial pressure difference between the sample and counter purge gas it is possible that changes in the sample $CO_2$ and $CH_4$ mole fraction relative to the counter purge gas, along with the $H_2O$ content of the sample, may alter the magnitude of any cross-membrane leakage.

A study by Welp et al. (2013) examined this issue and concluded that the leakage was small and well

within the WMO comparability guidelines. However, the drying approach used by Welp et al. (2013) is not directly comparable to that of the GAUGE sites as they used dry sample gas as the counter purge rather than zero air. That study also only examined two water contents (0 % or 2 % $H_2O$) and conducted only dry (0 % $H_2O$) experiments on samples with $CO_2$ and $CH_4$ mole fractions above ambient concentrations. Considering the importance of water in gas transport across the membrane (Chiou and

Paul, 1988) and the range of water contents observed in undried air samples measured within the DECC/GAUGE network (up to 3.5 % $H_2O$) further investigation of this issue was required.

As such, the second half of this paper aims to quantify the magnitude of Nafion $CO_2$ and $CH_4$ transport using the drying method used at the DECC/GAUGE TT sites along with errors associated with instrument specific water corrections. It also examines how these might change within the range of

$H_2O$, $CO_2$ and $CH_4$ mole fractions typically observed at these sites.




## 2. Tall tower observations and impact

### 2.1 Experimental — Site set up, sampling and calibration

### 2.1.1 Site descriptions

Two new tall tower sites, Heathfield (HFD; 50.977 °N, 0.231 °E) and Bilsdale (BSD; 54.359 °N, -1.150

°E) were established at existing telecommunication towers in December 2013 and January 2014, respectively. The general set up of these sites is similar to that described for the DECC sites in Stanley et al. (2018) and the locations of these two new sites relative to these sites described in Stanley et al. (2018) are shown in Figure 1.

Heathfield is located in rural East Sussex, 20 km from the coast. The closest large conurbation (Royal

Tunbridge Wells) is located 17 km NNE from the tower. The area surrounding the tower is > 90 % woodland and agricultural green space with some residential (0.7 %) and light industrial areas (0.3 %) (East Sussex in figures, 2006). Notable local industry includes a large horticultural nursery located only 200 m north of the tower.

Bilsdale is a remote moorland plateau site within the North York Moors National Park. It is 25 km

NNW of Middlesbrough (the closest large urban area) and 30 km from the coast. The tower is situated in a predominantly rural area, including moorland, woodland, forest and farmland (North York Moors National Park Authority, 2012;Chris Blandford Associates, 2011).

Inverted stainless steel intake cups were mounted at 42, 108 and 248 m.a.g.l. (metres above ground level) on the BSD tower and 50 and 100 m.a.g.l. at HFD. Air was pulled through the intake cups via ½ "

Synflex Dekabon metal/plastic composite tubing (EATON, USA) and a 40 µm filter (SS-8TF-40, Swagelok, UK) using a line pump (DBM20-801 linear pump, GAST Manufacturing, USA) operating at > 15 L min$^{-1}$. The instruments located at the sites sub-sampled from the tower intakes via a T-piece prior to the line pump. Further details can be found in Stanley et al. (2018).



### 2.1.2 Instrumentation

Both sites are equipped with a CRDS (G2401 Picarro Inc., USA) making high frequency (0.4 Hz) $CO_2$, $CH_4$, CO and $H_2O$ measurements. A GC coupled to a micro-electron capture detector (GC-ECD, Agilent GC-7890) is used to measure $N_2O$ and $SF_6$ every 10 mins. For further instrumental details, including flow diagrams and column details, see Stanley et al. (2018).

The sample lines, calibration and standard gas cylinders are linked to two multiport valves (EUTA-CSD10MWEPH, VICI Valco AG International, Switzerland), one for the CRDS and a second for the GC-ECD, the output of each valve is connected to the intakes of the instruments. Filters (7μm, SS-4F-7, Swagelok, UK) are located on the intake lines prior to the valve while a 2μm filter (SS-4F-2, Swagelok, UK) is located between the valve and the CRDS. The GC-ECD flow path, instrumentation and part numbers are described in detail in Stanley et al. (2018). However, in brief, air entering the GC-ECD system is first dried (Section 2.3.1) before flushing an 8 mL sample loop. The contents of the loop are transferred onto a combination of pre-, main and post chromatographic columns using P-5 carrier gas (a mixture of 5 % $CH_4$ in 95 % Ar; Air Products, UK).

The automated switching of valves and control of GC-ECD temperatures and flows, as well as logging the data and a range of other key parameters (flows, pressures, temperatures) is achieved using custom Linux based software (GCWerks, www.gcwerks.com). The CRDS instrument makes measurements at each intake height, switching between heights every 20 mins at BSD and 30 mins at HFD. While the GC-ECD measures only a single intake, initially the 108 m.a.g.l. intake at BSD (switched to the 248 m.a.g.l. intake on 17th March 2017) and the 100 m.a.g.l. intake at HFD. Other than the tower sample lines, all tubing within the system is 1/16 ", 1/8 " or ¼ " (o.d.) stainless steel (Supelco, Sigma-Aldrich, UK). A generalised diagram of the original sampling scheme for the two sites is shown in Figure 2.

### 2.1.3 Sample Drying

**GC-ECD**

All samples measured on the GC-ECD (air, standards and calibration) are dried using a Nafion permeation drier (MD-050-72S-1, Permapure, USA) prior to analysis. The counter purge gas for the



drier is generated from compressed room air. This air is dried to <0.005 % $H_2O$ by the compressor (50 PLUS M, EKOM, Slovak Republic) and a zero air generator (TOC-1250, Parker Balston, USA).

**CRDS**

In an attempt to minimise the water correction required for dry mole fraction CRDS measurements,

CRDS samples were initially dried using a Nafion in an identical manner to those of the GC-ECD. This resulted in air samples with water mole fractions between 0.05 and 0.2 % $H_2O$ depending on the original moisture content of the air. However, as outlined (Section 3.3.3), this method was found to be problematic. As such, the CRDS Nafion drying systems were removed (17[th] of June 2015 & 30th of September 2015 at BSD and HFD, respectively), undried air analysed and the data post corrected with

an instrument specific water correction.

**Calculating instrument specific water corrections at site**

Motivated by perceived weaknesses in the use of Nafion driers in this application, the decision was made to measure wet samples and correct using an instrument specific water correction. These corrections were determined in the field by conducting a droplet test, similar to those used in Yver

Kwok et al. (2015). In this test, a cylinder of dry natural air was humidified and the change in $CO_2$ and $CH_4$ mole fraction with water content examined. In brief, a 1.5 m length of 3/8 " Synflex Dekabon metal/plastic composite tubing (EATON, USA) was introduced between the standard cylinder outlet and the CRDS intake. Distilled water (0.7 mL) was injected through a septum located on a T-piece fixed on the "cylinder end" of the Dekabon tubing (See Figure S1 for flow diagram). This water evaporated

into the sample stream, with the $H_2O$ mole fraction typically peaking at up to 4.5 % (dependent on room temperature) before decreasing to pre-injection concentrations. The effect of this changing $H_2O$ concentration on the raw (without the inbuilt $H_2O$ correction) $CO_2$ and $CH_4$ concentrations was then observed.  The experiment was repeated in at least triplicate annually.

A water correction was then determined from a fit between the "wet"/mean "dry" ratio and the $H_2O$ of

the droplet test data and the equation given by Rella (2010). Here we defined "dry" data as any data with $H_2O$ < 0.001 %, as measured by the CRDS, and the remaining data as "wet". We use minute mean uncorrected CRDS $CO_2$ and $CH_4$ data for this analysis, that is, minute averaged data from the "co2_wet" and "ch4_wet" columns of the raw Picarro data files along with data from the "h2o" column.





This $H_2O$ data, unlike the "h2o_reported" data has been corrected for spectral self-broadening as detailed in Rella (2010). Data collected in the first 5 minutes immediately following the injection and data with minute mean standard deviations > 0.5 % $H_2O$, 0.05 µmol mol$^{-1}$ $CO_2$ or 0.4 nmol mol$^{-1}$ $CH_4$ were excluded from the fit. The fit was conducted using orthogonal distance regression weighted by

both the minute mean standard deviation of the $H_2O$ and gas of interest ($CO_2$ or $CH_4$). The resulting correction parameters are shown in Table 1. These corrections were then applied to minute mean observational data through the GCWerks software completely bypassing the built-in $CO_2$ and $CH_4$ water corrections.

### 2.1.4 Calibration and traceability

Calibration procedures for both the CRDS and GC-ECD are as described in detail in Stanley et al. (2018). In brief, CRDS measurements are calibrated against both a close-to-ambient standard and a set of three calibration cylinders, which span the typical ambient range (Table 2). Only a small number of elevated observations, < 0.4 % of the $CO_2$ and < 1.5 % of the $CH_4$ minute mean observations, were outside the range of the calibration cylinders. However, a much higher proportion of the CO

observations were outside the range of the calibration suites used at site, 28 % at BSD and 43 % at HFD, with the majority of these data points (> 98 %) below the lowest calibration cylinder. Assigning mole fractions to values outside the range of the calibration suite greatly increases the error and should be reflected in increased uncertainty estimates. Daily measurements of the ambient standard are used to account for any linear drift, while monthly measurements of the calibration suite are used to

characterise the nonlinear instrumental response. This calibration procedure is controlled by the GCWerks software and allows near real-time examination of calibrated data.

CRDS standards and calibration gases are all composed of natural air, either spiked or diluted with scrubbed natural air (TOC gas generator, Model No. 78-40-220, Parker Balston, USA) to achieve the required concentrations of $CO_2$, $CH_4$ and CO. This removes any pressure broadening effects inherent in

the use of non-matrix matched artificial standards (Nara et al., 2012). As the CRDS is an isotopologue specific method the calibration and standard cylinders were filled at Mace Head with well-mixed Northern Hemisphere air in an effort to ensure that the isotope ratios of the standard and calibration





gases were as close to those of the sampled air as possible (see Lee et al. (2006) or Nara et al. (2012) for further discussion).

GC-ECD measurements are made relative to a natural air standard of known $N_2O$ and $SF_6$ concentration. This standard is measured hourly and used to linearly correct the samples (Table 3). The instrumental nonlinearity response was characterised prior to deployment by dynamically diluting a high concentration standard with zero air and was repeated in the field at the BSD site on 30th September 2015.

GC-ECD and CRDS standards and calibration cylinders were calibrated before and where possible after deployment at the sites. If these two measurements agreed then a mean mole fraction was used, otherwise a linearly drift corrected mole fraction was used. The CRDS cylinders were calibrated through WMO linked calibration centres (either WCC-EMPA, Dübendorf, Switzerland or GasLab MPI-BGC, MPI, Jena, Germany). This ties the ambient measurements to the WMO $CO_2$ x2007 (Zhao and Tans, 2006), $CH_4$ x2004A (Dlugokencky et al., 2005) and CO x2014A (Novelli et al., 1991) scales. The calibration of the GC-ECD standards was conducted at either the AGAGE Mace Head laboratory or the University of Bristol laboratory and are reported here on the recently released SIO-16 $N_2O$ scale and the SIO $SF_6$ scale. Most cylinders were or will be calibrated before and after deployment and then mean of the two values used. Some cylinders, due to logistical constrains were only calibrated once (Table 3).

### 2.1.5 Instrument short-term precision and long-term repeatability

The short-term (1-minute) precision of the CRDS data was determined as the mean of the standard deviations of the 1-minute mean data. This was calculated for measurements of the standard cylinder and the calibration suite allowing the relationship between $CO_2$, $CH_4$ and CO mole fraction and short-term precision to be examined. Long-term repeatability of 20 minute means were calculated by examining the standard deviation of the daily standard cylinder measurements. This analysis included 18 cylinders covering a wide range of mole fractions (Table 2).

The mean absolute short-term precision for all cylinders was consistent between the two sites across all three gases. At BSD the short-term precision was $0.024 \pm 0.004$ µmol mol$^{-1}$ $CO_2$, $0.18 \pm 0.04$ nmol mol$^{-1}$ $CH_4$ and $4.2 \pm 0.7$ nmol mol$^{-1}$ CO while at HFD it was $0.021 \pm 0.004$ µmol mol$^{-1}$ $CO_2$, $0.22 \pm 0.04$





nmol mol$^{-1}$ CH$_4$ and 6 ±1 nmol mol$^{-1}$ CO ($\bar{x}$ ± 1σ). Both sites showed a small trend with the mean absolute precision increasing (i.e. becoming less precise) with increasing CO$_2$ and CH$_4$ mole fraction. However, this was not observed in the relative precision which remained unchanged at ~ 0.005 % for CO$_2$ and ~ 0.01 % for CH$_4$. This was not the case for CO where the relative precision improved with

increasing mole fraction from ~ 4 % at CO < 100 nmol mol$^{-1}$ to < 1.5 % at CO > 250 nmol mol$^{-1}$. We suspect that this tendency is inherent in the spectroscopic approach as the CO peak measured by the Picarro CRDS is much smaller than those of the CO$_2$ and CH$_4$ (Chen et al., 2013) and hence more susceptible to noise in the baseline particularly at low mole fractions.

Like short-term precision, mean long-term repeatability (calculated over a period of approximately a

year) is consistent between the two sites, 0.018 ± 0.009 and 0.013 ± 0.001 μmol mol$^{-1}$ CO$_2$, 0.20 ± 0.04 and 0.20 ± 0.01 nmol mol$^{-1}$ CH$_4$, and 1.1 ± 0.3 and 1.7 ± 0.3 nmol mol$^{-1}$ CO at BSD and HFD respectively ($\bar{x}$ ± 1σ).

Repeatability of individual injections on the GC instruments were calculated as the standard deviation of the hourly standard injection. These were found to be < 0.3 nmol mol$^{-1}$ and < 0.05 pmol mol$^{-1}$, for

N$_2$O and SF$_6$ respectively, and did not differ between the two sites.

## 2.2 Data analysis - quality control, statistical processing and modelling

### 2.2.1 Data quality control

A three-stage data flagging and quality control system was used for the HFD and BSD data. Initially, automated flags based on the stability of key parameters including cell temperature & pressure and

instrument cycle time (the time taken to collect and process each measurement) were applied. Here, data with a cycle time > 8 seconds were filtered out along with any data with cell pressure outside the range 45 ± 0.02 Torr or cell temperatures outside 140 ± 0.1°C. Secondly, a daily manual examination of the GC chromatograms and key GC/CRDS parameter values of each site were made. Data points were flagged if instrument parameters varied beyond thresholds determined to reduce their accuracy and a

reason for the removal was logged. Finally, all sites were reviewed simultaneously and the mixing ratio of the same gas from each site are overlaid to look for differences between sites. Any significant differences between the background values at each site were investigated by examining key



instrumental parameters, calibration pathways and 4-hourly air mass history maps to ensure that these differences represent true signals rather than instrumental or calibration driven artefacts. The hourly air mass history maps were produced using the Numerical Atmospheric dispersion Modelling Environment (NAME) Lagrangian dispersion model (Manning et al., 2011).

**2.2.2 Statistical processing, baseline fitting and seasonal cycles**

Statistical baselines and long term trends for each gas at each height were calculated using the CCGFILT Python package (Thoning, 2015). This is based on the curve fitting technique originally used by Thoning et al. (1989). In this package the data is initially fitted to a function combining simple polynomial (the long-term trend) and harmonic functions (annual seasonality). Secondly, the statistical baseline (smoothed curve) is defined using the short-term deviations from this function. These fit residuals are transformed into the frequency domain using a Fast Fourier Transform (FFT), filtered using a low-pass filter with a short term cut off of 80 days and a long term cut off of 667 days. The inverse FFT is then used to transform the data back into the time domain. To reduce the computational complexity of the fitting long term trends and the statistical baselines were determined using hourly $CO_2$, $CH_4$ and CO means, while the $N_2O$ and $SF_6$ data were fit at their native 10-minute frequency.

Annual seasonal cycles were determined as the difference between the statistically smoothed curve and the long-term trend. Diurnal cycles were calculated using detrended hourly mean data (i.e. hourly mean minus long term trend). These data were split into 24 hour-of-the-day bins for each season and the mean, 5[th] and 95[th] percentile of each bin determined. The mean value of all the hourly means at each sample height were then added to each bin mean for plotting.

**2.2.3 Modelling**

The observations can be combined with the output from atmospheric transport models, prior information and uncertainty assessments of each of the components in an inverse modelling framework to estimate regional emissions of each gas. Examples of different inverse methods that have been applied using the UK DECC network are given in Manning et al. (2011), Ganesan et al. (2015) and Lunt et al. (2016).



For $CH_4$ emissions during 2014 we used two different methods: the Inversion Technique for Emission Modelling (InTEM, Manning et al., 2011) and a reversible-jump Markov chain Monte Carlo (rj-MCMC) approach (Lunt et al., 2016). Both methods are based on Bayesian principles of using the atmospheric mole fraction data to update the prior information on emissions. The UK's 2015

submission to the UNFCCC for 2013 is used as the prior estimate of the magnitude and spatial distribution of emissions. Two-hourly mean mole fractions are used from each site, with the measurements filtered using the modelled stability of the atmosphere, so that only data from times when the atmosphere is thought to be well-mixed are included. The rj-MCMC approach estimates the form of the model-measurement uncertainty covariance matrix within the inversion itself, and only loose bounds

are set on these terms a priori. The InTEM method assumes fixed model-measurement covariance terms that are defined based on the modelled atmospheric stability. Monthly inversions were performed for the 1-year period under two set-ups, the first using only the four sites of the original UK DECC network, and the second incorporating the additional data from BSD and HFD. Output from each of these independent monthly inversions were averaged to derive annual statistics.

**2.3 Results and discussion**

**2.3.1 $CO_2$, $CH_4$ and CO key features**

The minute mean $CO_2$ observations range between a low of 379.50 to a high of 497.48 µmol mol$^{-1}$ $CO_2$ at Heathfield and 379.77 to 587.17 µmol mol$^{-1}$ $CO_2$ at Bilsdale. While the HFD mean $CO_2$ mole fraction, 407.5 µmol mol$^{-1}$, is slightly higher than the mean mole fraction of the BSD site, 404.7 µmol

mol$^{-1}$, the high $CO_2$ mole fractions observed at BSD are generally higher than those of the HFD site (Figures 3a & 4a). The high mole fraction events observed at BSD are generally sporadic — lasting only a couple of hours — and appear as a brief pulse relative to the normal diurnal cycle; a pattern indicative of a nearby point source. Considering BSD is remote from large conurbations, measured signals are expected to be dominated by biogenic sources. In this instance, we suspect high mole

fraction events at BSD are due to local heather (*Calluna vulgaris*) burning. These $CO_2$ events also typically coincide with periods of elevated $CH_4$ and CO, again suggesting a biomass burning source. In contrast, HFD is located in southern England just south of London (Figure 1). Here, high $CO_2$ events





are typically longer — 2 to 3 days — and drive a positive shift in the entire diurnal cycle, suggesting a change in the background mole fraction. Air histories, based on the output of the Numerical Atmospheric dispersion Modelling Environment (NAME) Lagrangian dispersion model, outlined in Manning et al. (2011), for these periods of elevated $CO_2$ typically show the source of the air to be from

over London or Europe. This difference in pollution event patterns is not apparent in the $CH_4$, CO, $N_2O$ or $SF_6$ data.

Both sites show a clear pattern in mean $CO_2$ mole fraction with intake height with the lowest height generally having the most elevated mole fractions, followed by the higher heights (Figures 3a & 4a). This trend is also apparent for $CH_4$ (and to a lesser degree CO), which is also measured at multiple

intakes and is most likely driven by the closer proximity to surface sources (Figures 3c & e and Figure 4c & e). The gradient in $CO_2$ and $CH_4$ mole fraction is most apparent during the early hours of the morning (see Figure 3b & d and Figure 4b & d). At these times the boundary layer is the lowest and $CO_2$ and $CH_4$ released from the surface sources during the night build up leading to relatively higher mole fractions at lower elevations. This difference disappears later in the day as the sun warms the

atmosphere and thermal mixing causes the boundary layer to rise.

The timings of the HFD and BSD seasonal cycles are similar, with $CO_2$ mole fractions highest in the colder months and lowest during the Northern Hemisphere summer; however, the magnitude of the summer draw down appears approximately 25 % larger at BSD than HFD (Figure 5a). Although both sites are located in areas of significant agriculture and green space the HFD site is more urbanised. This

appears to be reflected in a significantly reduced summer $CO_2$ uptake and slightly larger winter increase relative to the BSD site. The HFD CO seasonal cycle also shows an enhanced winter maximum relative to BSD while the summer minimum is consistent between the two sites. Suggesting that the HFD site is more sensitive to fossil fuel emissions.

As with the seasonal cycle, the shape of the $CO_2$ diurnal cycle is similar at both sites, with mole

fractions peaking near sunrise and the lowest $CO_2$ mole fractions observed in the late afternoon (Figures 3b and 4b). Again, the amplitude of these cycles varies between the sites with HFD, the more anthropogenically influenced site showing an approximately 3 μmol mol$^{-1}$ $CO_2$ higher maximum in the early morning.





Although there is a very large range in the minute mean $CH_4$ observations, 1841 to 3065 nmol mol$^{-1}$ at BSD and 1843 to 3877 nmol mol$^{-1}$ at HFD, > 99.99 % of measurements, are less than 2400 nmol mol$^{-1}$ $CH_4$, with only 6 events in the combined record exceeding this threshold. These events have been clipped from the data shown in Figures 3c and 4c for ease of viewing. Like $CO_2$, the $CH_4$ observations

show clear seasonal cycles with the mole fractions the highest in the winter months and the lowest in midsummer (Figure 5c). Interestingly, there is also a gradient in the amplitude of the $CH_4$ seasonal cycle, with the amplitude decreasing with sample intake height. This trend is not apparent in the $CO_2$ or CO data (Figure 5a and e). Whether this is a true vertical gradient in $CH_4$ seasonality or an artefact of the fitting protocol (possibly driven by the larger pollution event to seasonal cycle ratio in the $CH_4$ data)

is unclear. The $CH_4$ diurnal cycle peaks in the morning usually 1 to 2 hours after sunrise (this is after the $CO_2$ maximum) and then dips in the mid-afternoon (Figures 3d and 4d). The $CH_4$ diurnal cycle is also more pronounced and smoother in the HFD data and evident throughout the year, whereas the BSD cycle is only strongly apparent in the summer months. This could be linked to differences in the relative magnitude of key local sources/sinks of $CH_4$ between the two sites.

A known weakness of the statistical fitting procedure is that, due to the FFT used, any gaps in the record must be filled using interpolation before analysis. This can lead to spuriously high (or low) values near gaps within the fitted seasonal cycle if the ends of the gaps occur during a significantly elevated (or reduced) event. These "end effects" are very evident in the HFD 50 m intake record for $CH_4$ (and CO and $SF_6$) in late 2017 leading to the large spikes in the seasonal cycle evident in Figure 5c, e and i.

Of the 5 gases measured at HFD and BSD, CO is the only gas to show a decrease in the long-term trend of the statistically defined baseline mole fraction between 2013 and 2017, roughly -6 nmol mol$^{-1}$ yr$^{-1}$ (Figure 5f). In contrast, the $CO_2$ and $CH_4$ data increase by approximately 3 µmol mol$^{-1}$ yr$^{-1}$ and 7 nmol mol$^{-1}$ yr$^{-1}$ respectively (Figure 5b and d). The HFD long-term trend curves in $CO_2$, CO and $CH_4$ are generally positively offset relative to BSD - most likely driven by the prevalence of polluted air at the

HFD site. The minute mean CO mole fractions at BSD averaged 269 nmol mol$^{-1}$ but varied between 63 and 9500 nmol mol$^{-1}$. The mean HFD CO minute mean value was slightly higher, 277 nmol mol$^{-1}$, however the CO range was significantly narrower, 60 to 4850 nmol mol$^{-1}$.





### 2.3.2 N$_2$O and SF$_6$ key features

Mean N$_2$O mole fractions observed from the two intakes of comparable height, 108m at BSD and 100m at HFD, were found to be identical (329.5 nmol mol$^{-1}$) and the range of N$_2$O observations were also very similar, 326.6 to 340.0 and 326.4 to 338.5 nmol mol$^{-1}$ for BSD and HFD, respectively (Figures 3g

and 4g). The N$_2$O data from the higher 248m intake at BSD, has not only a higher mean value, 331.1 nmol mol$^{-1}$ but a narrower range, 329.3 to 335.0 nmol mol$^{-1}$. The 248m data analysed in this paper was from the 17$^{th}$ March 2017 (compared with late 2013 for the other heights) till the 19$^{th}$ June 2017. Hence this data is likely to be higher in N$_2$O relative to the 108 m data due to both the long-term trend in N$_2$O mole fraction and a seasonal bias, as it encompasses mainly the spring time maximum in the seasonal

cycle rather than a full cycle. The smaller range in the 248 m data is most likely driven by increased mixing with increasing altitude, which reduces the influence of local point sources.

The N$_2$O mole fraction seasonal cycle of both sites shows a unsual pattern with two maxima per year, one in early spring and a second in autumn (Figure 5g). Both the timings and amplitudes of these cycles and the long-term trend (Figure 5h) are very similar between the two sites. A previous study, Nevison et

al. (2011) examined the monthly mean N$_2$O seasonality of baseline mole fraction data at Mace Head (MHD, 53.327 °N, -9.904 °E, Figure 1), a remote site within the UK DECC network located on the west coast of Ireland. They found that although biogeochemical cycles predict a single thermally driven summer time maximum in N$_2$O flux (and hence mole fraction) (Bouwman and Taylor, 1996), they actually observed a late summer minimum, with a single N$_2$O concentration peak in spring. This was

attributed to the winter intrusion of N$_2$O depleted stratospheric air and its delayed mixing into the lower troposphere. In contrast, a UK focused inversion study Ganesan et al. (2015), found that N$_2$O flux seasonality is driven not just by seasonal changes in temperature but by agricultural fertilizer application and post-rainfall emissions. They predict the largest net N$_2$O fluxes will occur between May and August while agricultural fluxes will peak during spring for eastern England and summer time for

central England. However, the exact timings of these fluxes can vary year-to-year as they depend not only on the scheduling of agricultural fertilizer application but on rainfall and temperature. Like MHD, BSD and HFD are expected to experience a decrease in N$_2$O driven by stratospheric intrusion, which would account for the springtime maximum and summer minimum. However, both BSD and HFD are



located much closer to significant agricultural sources of $N_2O$ than MHD. Hence, it is likely that they are much more influenced by agricultural $N_2O$ fluxes. As such, it is possible that although a summer time maximum in $N_2O$ flux is completely offset by stratospheric intrusion, this summer time maximum may be so large that the residual autumn tail of this event appears as a second maximum at BSD and

HFD.

Clear diurnal cycles in $N_2O$ were observed at the HFD for the spring, summer and autumn months with the maximum $N_2O$ mole fraction occurring 2 hours after sunrise and the minimum in the mid-afternoon (Figure 4h). These cycles were not apparent at BSD (Figure 3h).

The long-term trend in the $SF_6$ mole fraction at BSD and HFD shows a gradual increase of 0.3 pmol

$mol^{-1}$ $yr^{-1}$ (Figure 5j). Although the predominant sources of $SF_6$ are electrical switchgear, which is not expected to have significant seasonality, there was a small seasonal cycle observed (Figure 5i). This cycle is more apparent in the BSD data and appears as a slight (0.1 to 0.15 pmol $mol^{-1}$) enhancement in $SF_6$ in the winter months. We hypothesise that this may be due to increased load on, and hence increased failure of, the electrical switchgear during the colder months. $SF_6$ mole fractions averaged 8.9

pmol $mol^{-1}$ at both BSD and HFD. While HFD, located closer to large conurbations than BSD, typically saw higher $SF_6$ pollution events. This was reflected in its larger range of 8.1 to 34.2 pmol $mol^{-1}$ compared with 8.1 to 22.9 pmol $mol^{-1}$ at BSD (Figures 3i and 4i).

### 2.3.3 Impact on UK GHG emissions estimates

As an example of the expected impact of the addition of these two additional TT sites on UK GHG emission estimates, the 2014 UK $CH_4$ flux has been estimated using both the InTEM and rj-MCMC methods. The result obtained from the InTEM method using only the 4 stations of the DECC network is 2.31 ± 0.77 Tg $yr^{-1}$ and when including the new GAUGE towers (6 stations in total) is 2.22 ± 0.70 Tg $yr^{-1}$. These flux estimates agree well with those of the rj-MCMC method, 2.07 (1.83–2.33) Tg $CH_4$ $yr^{-1}$

for 4 towers and 2.00 (1.81–2.20) Tg $CH_4$ $yr^{-1}$ for 6 stations. All four estimates are consistent with the bottom-up UK National Atmospheric Emissions Inventory estimate of 2.16 Tg $CH_4$ $yr^{-1}$ for 2014 (Brown et al., 2017). The inversion estimates include natural as well as anthropogenic components,





although for the UK the natural component is considered to be small (< 10 %). Although the mean flux estimate does not change significantly, the addition of the two stations reduced the UK total uncertainty by approximately 10 % in magnitude for the InTEM method and 20 % for the rj-MCMC method. Furthermore, the mean spatial distribution of emissions in the rj-MCMC approach was altered with the

addition of the extra two stations, reducing the emissions estimate over London and increasing it further north (Figure 6a). Figure 6b shows how the additional data from the extra two sites lead to a general reduction of uncertainty in those areas closest to these sites, whilst increasing the uncertainty in other areas, possibly due to inconsistencies in the modelling of methane mole fractions between different stations. The methane inversion results highlight the benefit of the additional TT data in reducing the

uncertainty on the emission estimates.

### 3. Sample drying and the use of empirical instrument specific water corrections

### 3.1 Experimental - Temporal stability and mole fraction dependence of instrument specific water corrections

The typical temporal stability and mole fraction dependence of the CRDS water correction was

examined using a laboratory based CRDS (G2301, Picarro Inc., USA; $CO_2$, $CH_4$ and $H_2O$ series). Here the water correction was determined using the droplet experiment, as described in Section 2.3.1. The mid-term and short-term stabilities were examined by repeating the experiment approximately weekly over a three-month period and daily for a 5-day period using a cylinder of dried ambient mole fraction air. A set of instrument specific water corrections was also determined in triplicate, using dried sub- and

above ambient air mole fraction cylinders.

### 3.2 Experimental - Nafion drying

### 3.2.1 Composition of the counter purge dry air stream

Measurements of the HFD, BSD and University of Bristol (UoB) laboratory Nafion counter purge contents were made using the HFD, BSD and UoB CRDS instruments, respectively. All counter purge

streams showed extremely low mole fractions of $CO_2 < 0.3$ µmol mol⁻¹, $CH_4 < 2$ nmol mol⁻¹ and $H_2O <$ 0.01 % (Figure S2). All these zero air streams have $CO_2$ and $CH_4$ mole fractions far lower than the 2015





mean global concentrations, 400.99 µmol mol$^{-1}$ $CO_2$ and 1840 nmol mol$^{-1}$ $CH_4$ (Dlugokencky and Tans, 2015), creating a clear and sizable partial pressure difference across the Nafion membrane for both species.

### 3.2.2 Experiment 1 — Quantifying $CO_2$ and $CH_4$ cross membrane transport with varying sample H$_2$O mole fraction

The magnitude and variability of Nafion cross membrane transport for ambient $CO_2$ and $CH_4$ mole fraction samples with changes in sample $H_2O$ concentration were investigated in a series of humidification experiments. In these experiments a system (Figure 7) was constructed allowing the humidification, using a Dew Point Generator (DPG; Licor LI-610 Portable Dew Point Generator, USA), of two high-pressure cylinders of dry near ambient $CO_2$ and $CH_4$ mole fraction (Table 4; UoB-06 and H-296). These cylinders were measured using a Picarro G2301 CRDS at the University of Bristol (UoB) laboratory. Cylinder delivery pressure was controlled using single stage high purity stainless steel Parker Veriflo regulators (95930S4PV3304, Parker Balston, USA) or TESCOM regulators (64-2640KA411, Tescom Europe). The system was constructed so that four different sample treatments were possible:

"*Wet*" — Direct measurements of the DPG output bypassing the Nafion

"*NafWet*" — DPG output through the Nafion

"*Dry*" — DPG output, cryogenically dried, bypassing the Nafion

"*NafDry*" — DPG output, cryogenically dried, through the Nafion

Using the humidification system, the dry cylinder air was humidified to a range of dew points between 2.5 and 30 °C (0.6 to 3.5 % $H_2O$). As the humidification process within the DPG can alter the composition of the gas stream the sample was also periodically cryogenically dried. This allowed the effect of the DPG on $CO_2$ and $CH_4$ mole fraction to be isolated from that of the Nafion.

In brief, the output of the cylinder regulator was plumbed to the input of the DPG. A T-piece connected prior to the DPG input vented any excess gas via a flow meter (F1, Fig. 7) ensuring that the DPG input remained at close to ambient atmospheric pressure throughout the experiment. The output of the DPG passed through a second T-piece with the overflow outlet also connected to a flow meter (F2) to ensure



that the CRDS input pressure remained near ambient. A third flowmeter (F3) was placed on the outflow of the Nafion counter purge. Flow meters F1 and F2 had a range of 0.1–1 L min$^{-1}$ (VAF-G1-05M-1, Swagelok, UK) while F3 had a smaller flow range 0.5–0.5 L min$^{-1}$ (FR2A12BVBN-CP, Cole-Palmer, USA). Typical output flows were 0.1, 0.3 and 0.3 L min$^{-1}$ for F1, F2 and F3 respectively. After F2 the

sample flow was further split using a T-piece, with half the flow passing through a cryogenic water trap before reaching a 4-port 2-position valve, V1 (EUDA-2C6UWEPH, VICI Valco AG International, Switzerland, actually a 6-port valve configured as a 4-port valve). The other half bypassed the water trap and connected directly to V1. One of the outputs of V1 went via the Nafion to a second identical valve, V2, while the second output went directly to V2. The first output of V2 connected directly to the

input of the CRDS while the second connected to a pump (PICARRO Vacuum pump S/N PB2K966-A) set to a flow rate matching that of the CRDS (0.3 L/min) to ensure uniform flow through both branches of the system. These valves were controlled manually using a VALCO electronic controller and universal actuator.

The cryogenic water trap consisted of a coil of ¼" diameter (I.D. 3.36mm) stainless steel tubing

immersed in a Dewar of silicone oil (Thermo Haake SIL 100, Thermo Fisher Scientific, USA). The silicone oil was cooled using an immersion probe (CC-65, NESLAB) to less than -50 °C. Other than the water trap and two short sections (< 10 cm) of ¼" (O.D.) polymeric plastic tubing immediately prior to and post the DPG, 1/16 " stainless steel tubing was used throughout the system.

The experiment was conducted in a temperature-controlled laboratory at 19 °C, and thus, at

temperatures lower than a number of the dew points used within the experiment. Hence, in order to avoid condensation forming on the walls of the tubing, all components of the system between the cylinder, excluding the water trap, and the outputs of V2 were contained within a chamber heated to > 32 °C. Tubing between the heated chamber and the input of the CRDS was also heated with heating tape to > 32 °C while the internal temperature of the CRDS was > 32 °C throughout the experiment.

Multiple measurement blocks of each sample treatment were conducted after a lengthy initial stabilisation period. This period allowed for the establishment of equilibrium between the water in the condenser block of the DPG and the sample gas and lasted between 2-5. The treatment blocks varied in length depending on the time required for the concentration to stabilise — this was particularly slow





when switching between *Nafion dry* and *Nafion wet* sample streams. At least 15 minutes of data were collected after the concentration stabilised.

It is important to note that the DPG was not independently calibrated but the $H_2O$ concentration was measured directly by the CRDS during the *Wet* experiments. These values were used as the reference
$H_2O$ concentration in all calculations and plots for all three experiments.

Flow rates, cylinder pressure, chamber temperature, $H_2O$ trap temperature and CRDS outlet valve settings were monitored closely and regularly logged.

### 3.2.3 Experiment 2 — Quantifying $CO_2$ and $CH_4$ cross membrane transport with varying sample $CO_2$ and $CH_4$ mole fraction

A second set of experiments identical to those outlined above was conducted. However, these used cylinders of above and below ambient mole fractions – spanning the typical range of mole fractions observed at the sites (Table 4, UoB-04 and H-306). Like the initial experiment, these experiments used the 4 sample treatments — *Wet*, *NafWet*, *Dry* and *NafDry* — and were conducted over a range of dew points.

### 3.2.4 Experiment 3 — Quantifying $CO_2$ and $CH_4$ cross membrane transport using measurements
### of the counter purge gas

Finally, a third experiment was conducted. It aimed to observe gas exchange across the Nafion membrane by measuring the counter purge gas before ($CP_{in}$) and after the Nafion ($CP_{out}$). For this experiment, two changes were made to the DPG sample flow path described previously (Figure 7).
Firstly, the water trap was moved from its original location and placed immediately prior to the CRDS intake. This ensured that any difference in $CP_{in}$ and $CP_{out}$ was driven solely by transport across the Nafion membrane rather than by an artefact of differing $H_2O$ mole fractions. Secondly, a multiport valve (EUTA-CSD10MWEPH, VICI Valco AG International, Switzerland) was placed between V2 and the CRDS intake, before the $H_2O$ trap. Two new T-pieces were included in the system. One was placed
between the TOC outlet and the Nafion and a second downstream of flowmeter F3. These were connected to the multiport valve allowing the sampling of the Nafion counter purge gas prior to ($CP_{in}$) and after the Nafion ($CP_{out}$). A fourth flowmeter (F4) was placed after the second new T-piece to ensure





that the flow out of the system remained positive and the CRDS was not sampling ambient air. As it was not possible to heat the new multiport valve to > 25 °C only dew points of 25 °C or less were used in this experiment.

The DPG was used to set the water content of the sample at a range of dew points between 5 and 25 °C,
this time sampling only ambient, or above ambient, mole fraction cylinders (UoB-15 and UoB-16; Table 4). The experiment was conducted in three stages. Firstly, the $H_2O$ content of the DPG humidified sample stream was allowed to stabilise. During this stage the $H_2O$ trap was removed from the Dewar of silicone oil and the CRDS measured an undried, Nafion bypassed sample, while the secondary pump maintained the flow of DPG sample through the Nafion. Secondly, the $H_2O$ trap was inserted into the
silicone oil and the water content monitored until 10 minutes of dry air (< 0.002 % $H_2O$) was obtained. Together these two stages took typically 2 to 3 hours to complete — allowing the Nafion time to equilibrate while ensuring that the $H_2O$ trap was drying the sample and the DPG had reached the required set point. The multiport valve was then used to switch between the $CP_{in}$ or $CP_{out}$ flows, measuring each for repeated 20 minute blocks (n > 3) at each dew point. The experiment was also
repeated with the DPG excluded (i.e. the dry cylinder, with $H_2O$ < 0.0001 %, connected directly to the Nafion) to obtain a sample with a dew point as close to 0 °C as possible.

### 3.2.5 Experimental data processing

All $CO_2$ and $CH_4$ data were corrected using the instrument specific water correction (Section 2.1.3). Minute mean values of all data were calculated from the raw 1.5 Hz data and exported from the
GCWerks software. Data processing was completed using code written using the Anaconda distribution of the Python programming language (Python Software Foundation, 2017;van Rossum, 1995) and a variety of standard packages including NumPy1.11.1 (Walt et al., 2011), SciPy 0.18.1 (Jones et al., 2001) and Matplotlib 2.0.2 (Hunter, 2007).

During the experiments, $CO_2$ mole fractions often took a significant amount of time (up to 5 hours) to
stabilise at each dew point – most likely due to a temperature driven shift in the carbonate buffering system within the condenser of the DPG. During Experiments 1 and 2 shorter stabilisation periods (30 to 60 mins) were also evident when switching the Nafion between wet and dry air streams due to the





drying/wetting of the membrane. Other studies using a similar approach (e.g. O'Shea et al., 2013) have not commented on this equilibration period. We suspect that this is due to the use of higher DPG flow rates. The O'Shea et al. (2013) experiment used flow rates of 2 L min$^{-1}$ (Bauguitte, 2015, per. comm.), double that used in our study, which may have increased the rate of equilibrium formation.

The mole fraction measured during each of the different sample treatments represented a different combination of possible effects. That is,

$$Wet = C_{True} + DPG + E_{Y\%}$$
$$NafWet = C_{True} + DPG + E_{Y\%} + N_{X\%}$$
$$Dry = C_{True} + DPG + H + E_{Y\%}$$
$$NafDry = C_{True} + DPG + H + E_{Y\%} + N_{0\%}$$

Where,

$C_{True}$ the true mole fraction of the cylinder

DPG the effect of the dew point generator

$E_{Y\%}$ is the CRDS water correction errors at $Y\%$ $H_2O$ content

$N_{X\%}$ and $N_{0\%}$ are the effect of the Nafion at $X\%$ $H_2O$ and 0% $H_2O$ respectively.

$H$ is the effect of the water trap

$X\%$ is the water content directly after the DPG

$Y\%$ is the water content at the CRDS

Hence, by looking at the differences between the sample treatments, it was possible to isolate each of these effects. For example, the difference between the *NafDry* and *Dry* treatments represents the effect of the Nafion with an incoming sample of 0 % $H_2O$ content ($N_{0\%}$) plus the error in the CRDS water correction at the $H_2O$ content of the sample exiting the Nafion ($E_{y\%}$). While, assuming that the effect of

the cryogenic water trap is negligible (this can be assumed as the water trap was maintained at temperatures above -78 °C (WMO, 2016)), then the trend in the CRDS water correction error with water content can be examined as the difference between *Wet* and *Dry* runs. Using this to characterise





the error due to the water content, knowing the water content and examining *Dry* less *NafWet* and *Dry* less *NafDry* it is possible to isolate the effect of the Nafion.

As per Experiments 1 and 2, the data collected to examine the composition of the counter purge in Experiment 3, represents a combination of effects.

$$CP_{in} = True_{CP}$$
$$CP_{out} = True_{CP} - N_{X\%}$$

Where,

$True_{cp}$ the true mole fraction of the counter purge gas

$N_{X\%}$ is the effect of the Nafion at $X\%$ $H_2O$ in the sample stream

$X\%$ is the water content of the counter purge gas before the Nafion

Hence the difference between the mean of $CP_{in}$ and the mean of $CP_{out}$ represents both any transport of $CO_2$ (or $CH_4$) through the Nafion membrane and the effect of the water correction.

In all three experiments, the mean of the final 15-minute period of each block of each sample type at
each dew point was calculated. The uncertainty of this mean was determined as the 95% confidence interval based on the larger of either the standard deviation of the minute means or average of the standard deviations of the minute means. Examples of the raw data collected during Experiment 1 and 2 as well as Experiment 3 are given in Figures 8 and 9, respectively.

As all three experiments were subject to initial equilibration periods the mean block values were
interpolated to remove any temporal drift. For Experiments 1 and 2 a spline interpolation method (SciPy.interpolate) was used, if n > 3 a cubic spline was used, otherwise a linear spline. While Experiment 3, which was not affected by the lengthy equilibrium period, used only linear interpolation, however, these pairs were first filtered and pairs with members that differed by more than 0.01 µmol mol$^{-1}$ $CO_2$ were excluded from the analysis.

For Experiments 1 and 2, the differences between sample types (*NafWet - Dry*, *NafDry - Dry* and *Wet - Dry*) were calculated as the mean difference between the interpolated block means. The uncertainties of these differences were estimated as the larger of either: the standard deviation of the mean difference or the uncertainties of the two sample types added together in quadrature (e.g. for the *NafWet - Dry*





difference the *NafWet* and *Dry* uncertainties were added together in quadrature). In Experiment 3 the $CP_{out}$ - $CP_{in}$ difference was calculated as the difference between the $CP_{out}$ and time adjusted $CP_{in}$ values and the uncertainty estimated as the combined uncertainty of the $CP_{in}$ and $CP_{out}$ values.

### 3.2.6 Key experimental assumptions

These experiments assume that any changes in the $CO_2$ or $CH_4$ mole fraction are driven solely by the Nafion drying processes or the CRDS water correction error. Other possible sources of error or bias included, adsorption and desorption effects within the regulator and walls of the tubing, gas solubility within the condenser of the dew point generator and instrumental drift.

Regulator and tubing adsorption and desorption effects has been previously examined by Zellweger and
Steinbacher (2017, personal communication). They found that for Parker Veriflo type regulators, as used in this experiment, the effects can be quite large, up to 0.5 µmol mol$^{-1}$ $CO_2$ or 2 nmol mol$^{-1}$ $CH_4$. But that these effects were only evident at flow rates < 250 ml min$^{-1}$ and after significant periods of stagnation (15 hours). Considering the high flow rates (> 1 L min$^{-1}$) and long flushing times (2 to 3 hours) used in our experiment it is highly unlikely that regulator effects would make a significant
impact on the results.

As discussed earlier in Section 3.2.2, a lengthy equilibration period was used at the start of each DPG run and following any change in DPG set point. This was to account for the dissolution of sample gas, in particular $CO_2$, in the DPG water chamber. After this initial equilibrium period there were no rapid changes in the $CO_2$ mole fraction with only a slow drift, apparent in the data. CRDS instrumental drift is
also typically very small and slow. For the UoB CRDS instrument, long-term measurements of target style standard cylinders have shown the drift to be < 0.001 µmol mol$^{-1}$ day$^{-1}$ $CO_2$ and < 0.03 nmol mol$^{-1}$ day$^{-1}$ $CH_4$. These drift rates are at least an order of magnitude smaller than the mole fraction differences observed in this study.

Although small, any time dependent drifts were accounted for by temporally interpolating between each
block of data. Also key to the design of this experiment is the examination of differences between two very similar mole fractions rather than absolute mole fractions. As such, any systematic errors that might drive a systematic offset cancel out and any mole fraction depended biases are minimised.





### 3.3. Results and discussion

### 3.3.1 Site specific water corrections

The annually determined instrument specific water corrections are typically very similar at each site, often within the 95 % confidence interval of the triplicate runs (Table 1), suggesting that the corrections

are fairly stable between years and instruments. The mean absolute residuals of the instrument specific water corrections are quite small, $\leq 0.02$ µmol mol$^{-1}$ $CO_2$ and $\leq 0.4$ nmol mol$^{-1}$ $CH_4$ (Table 1). These residuals are on average slightly smaller than those of the inbuilt correction and are notably smaller at higher $H_2O$ content. For example, the mean residuals for 2015 data from HFD with $H_2O > 2$ % are 0.04 and 0.09 µmol mol$^{-1}$ $CO_2$ and 0.4 and 1.2 nmol mol$^{-1}$ $CH_4$ for the new and inbuilt correction,

respectively.

Plots of the residuals typically show a common pattern, with the residual of zero at 0 % $H_2O$, before dipping below zero and then returning to zero at $H_2O$ between 0.2 and 0.5 % (Figure S3). Unlike other tests, the depth and width of this dip is more pronounced for BSD 2017. However, the BSD 2017 data both spans a wider range of $H_2O$ contents than the earlier tests (0 to 3.5 % c.f. 0 to 2.2 %) and has far

fewer data points in the 0.1 to 1 % $H_2O$ range (0.9 % of all data points c.f. 34 % and 27 % for BSD 2015 and 2016, respectively). The BSD 2017 0.1 to 1.0 % data points also have an average standard deviation an order of magnitude larger than those of 2015 and 2016 (Figure S3 a, b and c). Refitting the BSD 2017 correction using only data $H_2O < 2.2$ % decreases the depth of the deviation by 0.05 µmol mol$^{-1}$ $CO_2$ and 0.3 nmol mol$^{-1}$ $CH_4$ as well as decreasing its width slightly but the deviation remains.

This suggests that the dip is robust but the change in the shape of the dip between 2017 and 2016 may well be a fitting artefact.

Reum et al. (2017) previously identified this pattern in water correction residuals and linked it to a pressure sensitivity at low water vapour mole fractions. They proposed an alternative fitting function incorporating the "pressure bend" although they do not recommend using this fit for data collected

during the droplet test due to the paucity of stable data typically obtained between 0.02 and 0.5 % $H_2O$ and the effect of rapidly changing $H_2O$ on the cell pressure sensor. Fitting the HFD and BSD data using this function reduced the dip in the residual but did not significantly alter the mean residual. However, due to the limitations of the droplet water test as outlined by Reum et al. (2017) we have decided not to




implement this. Implementing a more controlled water test at the sites would allow the use of the new fitting function. But due to the complexity of such a test this would be logistically difficult at remote field sites.

It is also important to note that the magnitude of the dip observed by Reum et al. (2017) in their

controlled water tests, ~ 0.04 µmol mol$^{-1}$ $CO_2$ and 0.5 nmol mol$^{-1}$ $CH_4$, are 4 fold smaller than those observed for the HFD, BDS and UoB droplet tests. As such the increased residuals observed for our water corrections between 0.02 and 0.5 % $H_2O$ are likely to be primarily driven by the rapidly changing $H_2O$ content inherent in the droplet test rather than represent a true error in the water correction.

The poor performance of the CRDS pressure sensor at low $H_2O$ mole fractions, 0.02 to 0.5 % $H_2O$, is

not expected to be a large source of error for undried samples as the majority of these, 92 % of the BSD and HFD data, contain > 0.5 % $H_2O$. But this is likely to be a large source of error for Nafion dried samples where 95 % of HFD and 92 % of BSD are < 0.5 %. Calibration gases, although partially humidified to < 0.015 % $H_2O$, when flowed through a damp Nafion are still far drier than Nafion dried air samples which averaged 0.2 % $H_2O$. As such this effect will not be accounted for as part of the

calibration process. It is difficult to quantify this error, as it will vary with sample water content and the sensitivity of the individual instrument's pressure sensor to low $H_2O$ mole fractions. However, for BSD and HFD it is likely to be a systematic offset of the order of -0.05 to -0.1 µmol mol$^{-1}$ $CO_2$ and -1 to -2 nmol mol$^{-1}$ $CH_4$.

The sample mole fraction dependence of the CRDS water correction was examined by conducting water

droplet tests using dry cylinders of above and below ambient mole fractions (Section 3.1). Specific above and below ambient water corrections were calculated based on these data sets. The difference (residuals) between these two types of corrected data and the true dry mole fraction of each cylinder was then determined. If the water correction was independent of sample mole fraction then the residuals should be identical for both correction types. However, although the $CO_2$ and $CH_4$ mean residual does

not change significantly, the shape of the $CO_2$ residual changes slightly with the residual of the above mole fraction sample becoming more positive at higher $H_2O$ mole fractions while the below ambient mole fraction residuals become more negative.





The change in the difference between dry mole fractions calculated using the earliest instrument specific water correction and subsequent water corrections for each instrument with water concentration is shown in Figure 10a & b. For a typical air sample (1.5 % $H_2O$, 400 µmol mol$^{-1}$ $CO_2$ and 2000 nmol mol$^{-1}$ $CH_4$) shifting between the annual water corrections drives $CO_2$ and $CH_4$ changes of < 0.05 µmol

mol$^{-1}$ and < 1 nmol mol$^{-1}$. However, this difference does change with water content and can increase outside the WMO reproducibility bounds at higher (> 2.5 %) $H_2O$ contents. For example, the difference between $CO_2$ dry mole fractions calculated using the Bilsdale 2015 and 2017 $H_2O$ correction increases to 0.12 µmol mol$^{-1}$ at 2.5 % $H_2O$. It's also important to note that these differences will scale with $CO_2$ and $CH_4$ mole fraction. Nevertheless, at the range of ambient water contents observed at BSD and HFD

(0.1 to 2 %) these differences remain below the WMO comparability guidelines (WMO, 2016) for $CO_2$ and $CH_4$ mole fractions < 750 µmol mol$^{-1}$ and < 4000 nmol mol$^{-1}$, respectively, as observed in BSD and HFD air samples. This is not the case for samples dried with a Nafion drier. Here the CRDS is measuring samples with low water contents, for example 38 % at BSD and 42 % at HFD of the Nafion dried air samples had $H_2O$ < 0.1 %.

A comparison of the individual daily and weekly tests, Figures 10c & d and 10e & f, conducted using the UoB instrument, show the daily tests to be far more similar than the weekly tests. That is, the variability over the 3-month period of the weekly test is much larger than that of the 5-day period of the daily test. However, the variability of the weekly tests is similar to those of the annual tests, Figure 10a and b, suggesting that, within the bounds of the data typically observed at the BSD and HFD sites, the

use of annually derived instrument specific water corrections are sufficient. This may not be the case at sites with higher levels of humidity and $CO_2$ and $CH_4$ mole fractions where water corrections may need to be determined more frequently, perhaps even weekly. Alternatively, a higher level of uncertainty could be applied to measurements made at higher water contents.

**3.3.2 Quantifying the CRDS water correction error**

The change in the CRDS water correction with sample $H_2O$ content was characterised using the difference between the *Wet* and *Dry* DPG runs (*Wet – Dry = $E_{X\%}$*). This error typically had a shallow negative parabolic trend for both $CO_2$ and $CH_4$ (Figure 11) and was similar to the shape seen in the



residual of the CRDS water corrections (Figure S3) with the error negative at $H_2O$ mole fractions near 0.5 %, becoming more positive between 1 and 2 % $H_2O$ before dropping at higher $H_2O$ contents.

Although the UoB CRDS was not deployed in the field we expect the results of the DPG tests to be typical of most PicarroG2401 $CO_2$/$CH_4$ CRDS instrumentation. The DPG tests show that for ambient

and below ambient mole fraction samples the $CH_4$ error remained within the WMO internal reproducibility guidelines (WMO, 2016) at all water contents examined, that is 0.7 to 3.5 % $H_2O$, while the $CO_2$ error increased outside the guidelines for $H_2O > 2.5$ %. $CO_2$ errors increased rapidly outside this range reaching 0.3 $\mu mol\ mol^{-1}$ at 3.5 % $H_2O$. These results are broadly consistent with those of the droplet test residuals.

Unlike the ambient and below ambient samples, the CRDS water correction error of the above ambient sample, UoB-04, exceeded the WMO internal reproducibility guidelines for both $CO_2$ and $CH_4$ at most $H_2O$ mole fractions. For the $H_2O$ range of the BSD and HFD sites the error peaked at 0.1 $\mu mol\ mol^{-1}$ for $CO_2$ near 1.75 % $H_2O$ and at 2 $nmol\ mol^{-1}$ $CH_4$ near 2.25 % $H_2O$. As discussed earlier in Section 3.1, the absolute error in the CRDS water correction will scale with the absolute mole fraction of the sample

due to the structure of the correction. The CRDS correction was also optimised using a cylinder of significantly lower mole fraction (397.38 $\mu mol\ mol^{-1}$ $CO_2$ and 1918.73 $nmol\ mol^{-1}$ $CH_4$ compared with 515.4 $\mu mol\ mol^{-1}$ and 2579.5 $nmol\ mol^{-1}$). This shift in error/residual was also observed in the $H_2O$ droplet tests using higher mole fraction cylinders. Although it appears larger for the DPG tests, most likely due to the higher mole fractions used within these tests (515.4 and 2579.5 compared with 449.55

$\mu mol\ mol^{-1}$ $CO_2$ and 2148 $nmol\ mol^{-1}$ $CH_4$, respectively).

Considering the typical $H_2O$ mole fractions observed at the HFD and BSD sites, 0.05 to 2.5 % $H_2O$, any water driven error in the CRDS water correction is not likely to be a major source of uncertainty for these sites. Even at other DECC sites that are subject to higher humidity, for example the Angus site (Stanley et al., 2018) periods of high (> 2.5 % $H_2O$) water content are rare, < 0.03 % of the data record.

In contrast, as elevated $CO_2$ and $CH_4$ mole fractions are regularly observed at both the HFD and BSD sites, the increase in CRDS error with mole fraction is a source of concern and must be quantified as part of a full uncertainty analysis.



### 3.3.3 Quantifying Nafion cross membrane transport

**Experiments 1 and 2 — Examining the change in sample composition**

The *NafWet - Dry* and *NafDry - Dry* differences (Figure 12) represent the sum of the error in the CRDS water correction ($E_{Y\%}$) and transport across the Nafion membrane ($N_{X\%}$). Where Y% is the water

content measured at the CRDS (Figure 13).

Initially, it was assumed that the *Wet - Dry* experiments would be sufficient to characterise the error in the CRDS water correction for the *NafWet - Dry* and *NafDry - Dry* experiments. However, the water mole fraction range examined in each experiment differ significantly (0.5 to 3.5 % and 0 to 0.31 %, respectively; Figure 13) and the trends evident in the *Wet - Dry* experiments, along with those seen in

the residual of the CRDS droplet tests and by Reum et al. (2017), make it clear that it is not possible to simply extrapolate from one experiment to the other. Instead, the CRDS water correction error for the *Dry* experiment ($E_{Y\%}$, Y% = 0.002 ± 0.006 % $H_2O$) was determined from the mean residual of the UoB 2017 water correction for water values < 0.0026 % $H_2O$ (n = 437). This mean residual was approximately zero, -0.002 ± 0.01 µmol mol$^{-1}$ $CO_2$ and -0.005 ± 0.2 nmol mol$^{-1}$ $CH_4$ (Figure S3h).

The corrections for *NafDry* runs ($E_{Y\%}$, Y% = 0.04 ± 0.02 % $H_2O$; Figure 13a) are more difficult to estimate as the 0.02 to 0.1 % $H_2O$ region is poorly and sporadically sampled in the droplet test. The data points observed within this region of the water droplet test are also highly uncertain due to the adverse effect of rapid changes in $H_2O$ and low mole fractions of $H_2O$ on the pressure sensor. Estimates of the *NafDry* water correction error ($E_{Y\%}$) from the droplet tests (Figure S3g & h) differed between years for

both $CO_2$ (-0.12 ± 0.08 and -0.074 ± 0.009 µmol mol$^{-1}$, for 2016 and 2017, respectively) and $CH_4$ (-0.6 ± 0.3 and -1.20 ± 0.03 nmol mol$^{-1}$, for 2016 and 2017, respectively). These were determined as the mean residual for $H_2O$ between 0.02 to 0.06 % with n = 52 and 7, for 2016 and 2017, respectively. Due to limited number of measurements in 2017, the 2016 value was used as the estimate of $E_{Y\%}$ for *NafDry* runs (Table 5).

Estimating *NafWet* $E_{Y\%}$ (Y% = 0.19 to 0.31 % $H_2O$) values from the water droplet tests (Figure S3g & h) was even more challenging as there are so few data points between 0.19 and 0.31 % $H_2O$ — only 3 for 2016 and none for 2017. The general trend observed during droplet tests (and *Wet - Dry* experiments) also suggest that the CRDS water correction error will become more positive with



increasing $H_2O$ within the 0.19 to 0.31 % window. As such, the *NafWet* $E_{Y\%}$ was estimated using a linear extrapolation of the droplet test between 0.1 and 0.6 % $H_2O$. Using this method, the 2016 $CO_2$ results (-0.21 to -0.16 µmol mol$^{-1}$) agree fairly well with 2017 (-0.18 to -0.17 µmol mol$^{-1}$) while the 2016 $CH_4$ values (-1.5 to -1.3 nmol mol$^{-1}$) were more positive than those of 2017 (-2.3 to -2.1 nmol mol$^{-1}$).

$^{1}$). Again, there are far fewer points in the 2017 data set within this window, 3 in total, compared with 32 for the 2016 data set. As such the 2016 estimate was considered more robust and used in later calculations (Table 5).

A second estimate for the *NafDry* and *NafWet* $E_{Y\%}$ values was made from data found in Reum et al. (2017). These estimates, approximately zero $CO_2$ and $CH_4$ for *NafDry,* and -0.03 to -0.04 µmol mol$^{-1}$

$CO_2$ and -0.5 to -0.7 nmol mol$^{-1}$ $CH_4$ for *NafWet,* are between 2 or 4-fold smaller than the droplet test estimates (Table 5). Considering that the Reum et al. (2017) $H_2O$ data is far more stable than those of the droplet tests and the instability of the CRDS with rapid changes in $H_2O$, particularly at low $H_2O$ mole fractions, we consider the Reum et al. (2017) estimate to be closer to the true CRDS error at these water contents.

The Nafion cross membrane transport term ($N_{X\%}$) was isolated by deducting the CRDS $H_2O$ correction error ($E_{Y\%}$) from the *NafWet - Dry* and *NafDry - Dry* differences, where negative values represent a loss from the sample to the counter purge and positive values a gain. The resulting $N_{X\%}$ values were highly dependent on the estimate of CRDS error used. Dry $N_{X\%}$ values determined using the droplet tests, 0.07 ± 0.08 µmol mol$^{-1}$ $CO_2$ and -0.1 ± 0.4 nmol mol$^{-1}$ $CH_4$, were typically more positive and had larger

uncertainties than those determined using the Reum et al. (2017) data, -0.05 ± 0.02 µmol mol$^{-1}$ $CO_2$ and -0.7± 0.2 nmol mol$^{-1}$ (Table 5). Similarly, the range of wet $N_{X\%}$ estimates based on the droplet tests were again more positive than those based on Reum et al. (2017) data.

Although small, these changes are an order of magnitude smaller than the typical, 0.003 µmol mol$^{-1}$ $CO_2$ and 0.03 nmol mol$^{-1}$ $CH_4$, standard error of the 15 min blocks of data used in this experiment. The

estimates based on the Reum et al. (2017) data, which are expected to be the most reliable, are the most negative of all the estimates suggesting that there is likely to be loss across the membrane. However, due to the inability to satisfactorily characterise the CRDS water correction error at low $H_2O$ mole



fractions, this experiment does not provide a sufficiently robust method to precisely quantify and hence correct for any Nafion drying effects.

**Experiments 3 — Examining the change in counter purge composition**

Unlike earlier experiments, all measurement of the Nafion counter purge ($CP_{in}$ and $CP_{out}$) were

cryogenically dried to < 0.002 % $H_2O$ prior to CRDS analysis. This removed the need to use the CRDS water correction and hence any error associated with it. As such, any differences between the $CP_{in}$ and $CP_{out}$ samples can be attributed solely to transport across the Nafion membrane ($N_{X\%}$). The results of these experiments are shown in Figure 14.

The counter purge experiments conducted with both the ambient (UoB-15) and above ambient (UoB-

16) mole fraction cylinders show identical changes in $CO_2$ and $CH_4$ mole fractions, respectively. The wet sample $N_{X\%}$ difference is consistently positive for $CO_2$ with the $CP_{out}$ mole fraction an average of 0.021 ± 0.002 µmol mol$^{-1}$ ($\bar{x}$ ± 95 % conf. int., n > 19) higher than $CP_{in}$, reflecting a loss from the sample to the counter purge across the Nafion membrane (Figure 14a). Although small, this value is an order of magnitude larger than the average standard deviation of the 15 min block means (0.002 µmol

mol$^{-1}$ $CO_2$) making it well within the typical measurement precision. This difference decreases slightly with decreasing sample water content but it is never zero. Even with a dry sample, the $CP_{out}$ - $CP_{in}$ difference ($N_{X\%}$), - 0.015 ± 0.003 µmol mol$^{-1}$ $CO_2$, is still negative. This is in line with previous studies, which have found that, although water substantially increases membrane permeability, even dry membranes are permeable to $CO_2$ (Ma and Skou, 2007;Chiou and Paul, 1988). As earlier studies have

found that membranes can take more than a week to fully dry out (Chiou and Paul, 1988), it is also highly likely that the relatively brief length of this study (4 to 5 hours) was too short to remove all $H_2O$ from the membrane.

The $CH_4$ $CP_{in}$ and $CP_{out}$ mole fraction difference for both dry and wet samples is also slightly positive, 0.03 ± 0.01 and 0.04 ± 0.02 nmol mol$^{-1}$ $CH_4$, respectively (Figure 14c). However, unlike the $CO_2$

difference, this value is very close to the measurement precision, with the average $CH_4$ standard deviation of the 15-min block means of the order of 0.02 nmol mol$^{-1}$ $CH_4$.

The ~ 0.02 µmol mol$^{-1}$ loss of $CO_2$ across the Nafion membrane from the sample stream to the counter purge observed here, although small, is of the order of the WMO internal reproducibility requirements,





0.05 µmol mol$^{-1}$ in the northern hemisphere and 0.025 µmol mol$^{-1}$ in the southern hemisphere (WMO, 2016), and must be acknowledged. However, the calibration gases are also passed through the Nafion. These cylinders are very dry, $H_2O$ < 0.0001 %, equivalent to the 0 °C data points used in Experiment 3 (Figure 14a and b) and as such would be expected to show similar $CO_2$ loss across the Nafion

membrane, ~ 0.015 µmol mol$^{-1}$. Hence, as the bias is constant with sample $CO_2$ and $H_2O$ mole fractions and as a bias would be present in both the calibration gases (~ 0.015 µmol mol$^{-1}$) and samples (~ 0.02 µmol mol$^{-1}$) the majority of the bias will be calibrated out, with only a very small (≤ 0.005 µmol mol$^{-1}$) constant bias, of the order of the instrumental precision, remaining. This is not the case for the water correction bias, which varies with $H_2O$ (Section 3.3.2).

In contrast, the mean $CH_4$ Nafion bias, 0.04 ± 0.02 nmol mol$^{-1}$, is at least an order of magnitude smaller than the WMO internal reproducibility guidelines (WMO, 2016) and is extremely close to the typical measurement precision suggesting that it is not a bias of concern.

Considering how poorly the results from Experiment 1 and 2 characterise the $N_{X\%}$ value it is interesting to see that the $CO_2$ $N_{X\%}$ value obtained here is within the bounds of those estimates while the $CH_4$

estimates are not. It is not clear whether this is a fortuitous quirk or because the CRDS water correction error is two-fold larger for $CH_4$ than $CO_2$ relative to the typical atmospheric mole fractions.

## 4. Conclusions and future work

The newly established Bilsdale and Heathfield tall tower measurement stations provide important new data sets of GHG observations. These high-precision continuous in situ measurements show clear long

term increases in baseline $CO_2$, $CH_4$, $N_2O$ and $SF_6$ mole fraction and capture the seasonal and diurnal cycles of these key gases. As demonstrated for $CH_4$, we expect that these observations, when combined with regional inversion modelling, will significantly improve our ability to quantify UK greenhouse gas emissions — both reducing the uncertainty and improving the spatial and temporal resolution. Future work using this data is focusing on better estimates of UK GHG emissions with a particular emphasis

on the UK carbon budget.

An examination of the Nafion drying method found it to have a small inherent $CO_2$ bias of 0.02 µmol mol$^{-1}$; however, this bias did not vary significantly with sample water content > 0.8 % $H_2O$ or $CO_2$ mole





fraction. Even samples as dry as the calibration gases were affected by this Nafion bias, although to a smaller degree — ~ 0.15 µmol mol$^{-1}$ for $H_2O$ < 0.0001 % — as residual moisture remained in the membrane. Thus, as calibration gases are dried in an identical manner to the samples, this bias is mostly calibrated out with only a very small (≤ 0.005 µmol mol$^{-1}$) constant residual bias of the order of the

instrumental precision. As such, the Nafion drier itself when, used in this manner, does not contribute a significant bias to the resulting $CO_2$ observations.

In contrast, the errors associated with the CRDS water correction for samples with low water contents (< 0.5 %), like those dried using a Nafion drier, can be significant and difficult to adequately quantify using the current in field techniques. Hence, even though Nafion driers are not themselves an inherent

source of bias, for the CRDS instrumentation examined in this study the incomplete drying of the sample is a significant source of error. This may not be the case for other CRDS instrumentation or optical techniques that use alternative cell pressure sensors.

Undried air samples with water contents typically between 0.5 and 2.5 % $H_2O$ are not subject to this error. However, care must be taken to ensure that any drift in the instrumental water correction is

identified and accounted for through regular water tests. The necessary frequency of these water tests will depend on the stability of the individual instrument and the typical $CO_2$, $CH_4$ and $H_2O$ mole fractions at the given location and should be determined on a case by case basis.

Future improvements to the Bilsdale and Heathfield records include the addition of target tanks at the sites to allow independent long-term monitoring of instrument performance and the development of a

full uncertainty analysis incorporating calibration, instrumental, water correction and sampling errors.

## 5. Acknowledgements

This study was funded under the NERC Greenhouse Gas Emissions and Feedbacks Program as part of the Greenhouse gAs UK and Global Emissions (GAUGE) area grant number NE/K002449/1NERC. This grant also covered the establishment and early running costs of the stations. Operating costs of the

Bilsdale site after 17$^{th}$ September 2016 were funded by the UK Department of Business, Energy and Industrial Strategy (formerly the Department of Energy and Climate Change) through contract TRN1028/06/2015. The National Physical Laboratory (NPL) took responsibility for the Heathfield site





on the 30<sup>th</sup> September 2017 by the UK's Department for Business, Energy and Industrial Strategy as part of the National Measurement System Programme.

The authors would also like to acknowledge the support of Dr Carole Helfter and Dr Neil Mullinger from the NERC Centre for Ecology and Hydrology (CEH), Edinburgh, Scotland who helped to

5  maintain and run the Bilsdale site. Lastly the authors would like to thank Dr Joseph Pitt from the University of Manchester for the use of the dew point generator used during a series of preliminary studies.



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





**Figure 1: Locations of the GAUGE Bilsdale (BSD) and Heathfield (HFD) sites, shown in black and the UK DECC Mace Head (MHD), Ridge Hill (RGL), Tacolneston (TAC) and Angus (TTA) sites, shown in grey.**



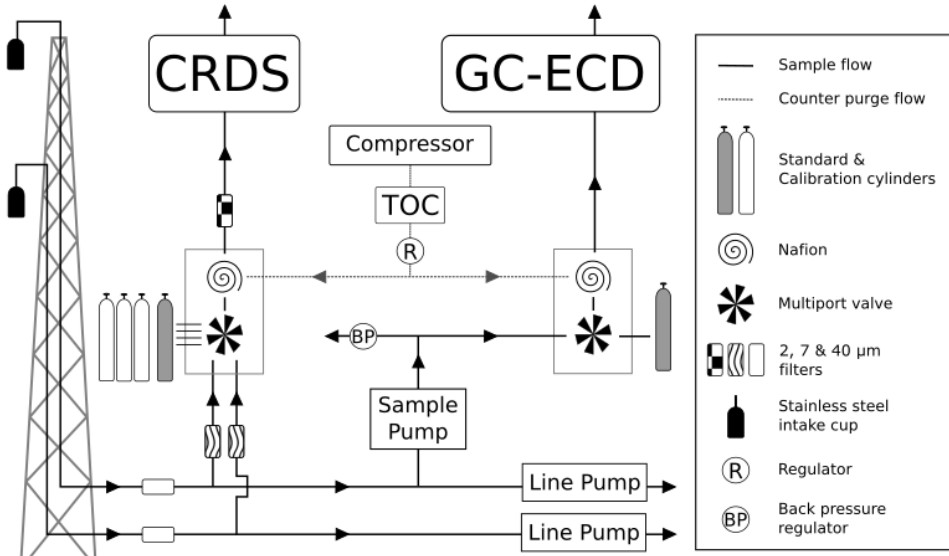

**Figure 2: A generalised schematic showing the initial Bilsdale and Heathfield site setup. Note that Bilsdale has three inlets, while Heathfield has only two as shown here. The Nafion drying system located downstream of the CRDS multiport valve was removed at both sites in 2015. Black arrows and lines show the direction of sample, standard and calibration gas flow. Grey dashed lines and arrows show the flow path of the Nafion counter purge gas.**




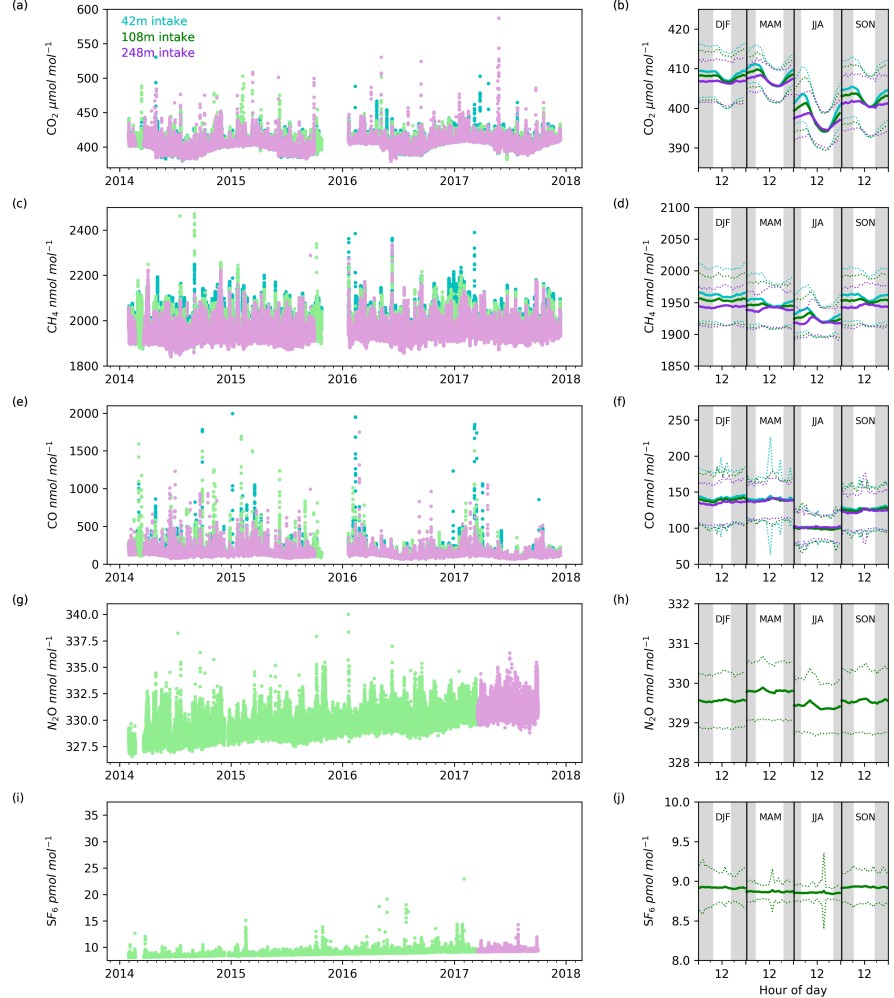

**Figure 3: Minute mean (a) CO$_2$, (c) CH$_4$ and (e) CO and 10 minute discrete (g) N$_2$O and (i) SF$_6$ observations and the mean diurnal cycle by season ± the 5$^{th}$ and 95$^{th}$ percentile for (b) CO$_2$, (d) CH$_4$, (f) CO, (h) N$_2$O and (j) SF$_6$ at the Bilsdale site for the 42m (blue), 108m (green) and 248m (purple) intake heights.**





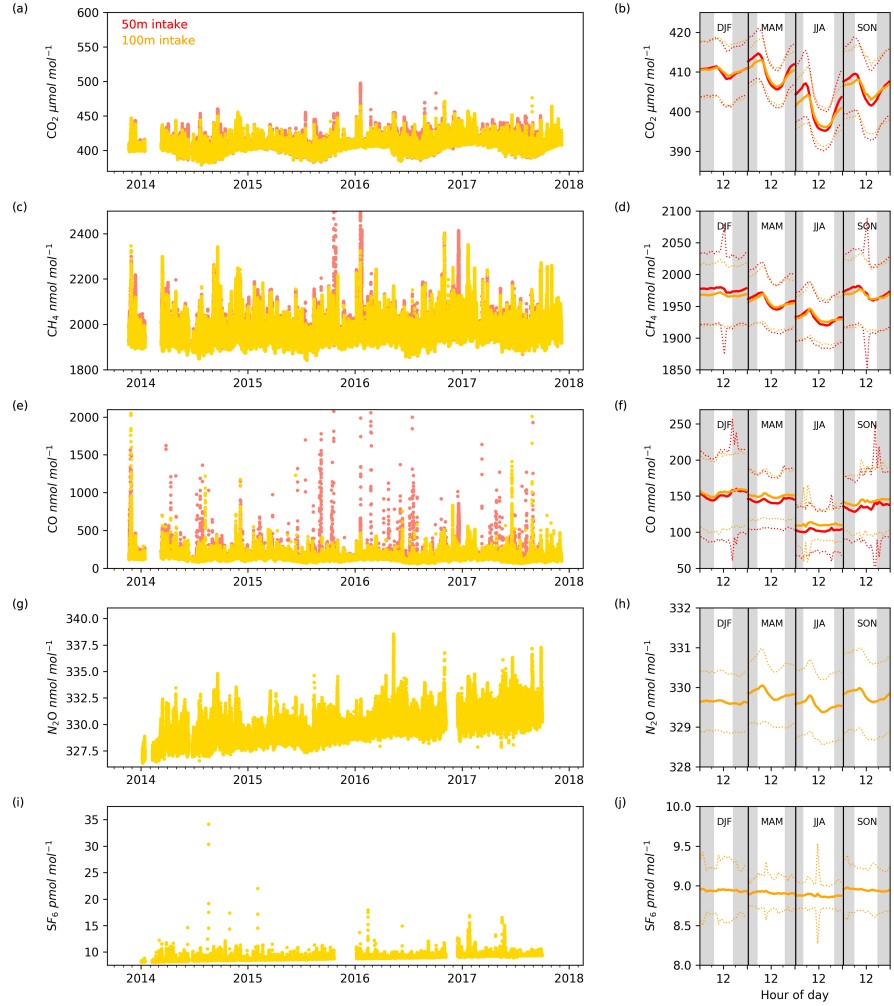

**Figure 4: Minute mean (a) $CO_2$, (c) $CH_4$ and (e) CO and 10 minute discrete (g) $N_2O$ and (i) $SF_6$ observations and the mean diurnal cycle by season ± the $5^{th}$ and $95^{th}$ percentile for (b) $CO_2$, (d) $CH_4$, (f) CO, (h) $N_2O$ and (j) $SF_6$ at the Heathfield site for the 50m (red) and 100m (yellow) intake heights.**



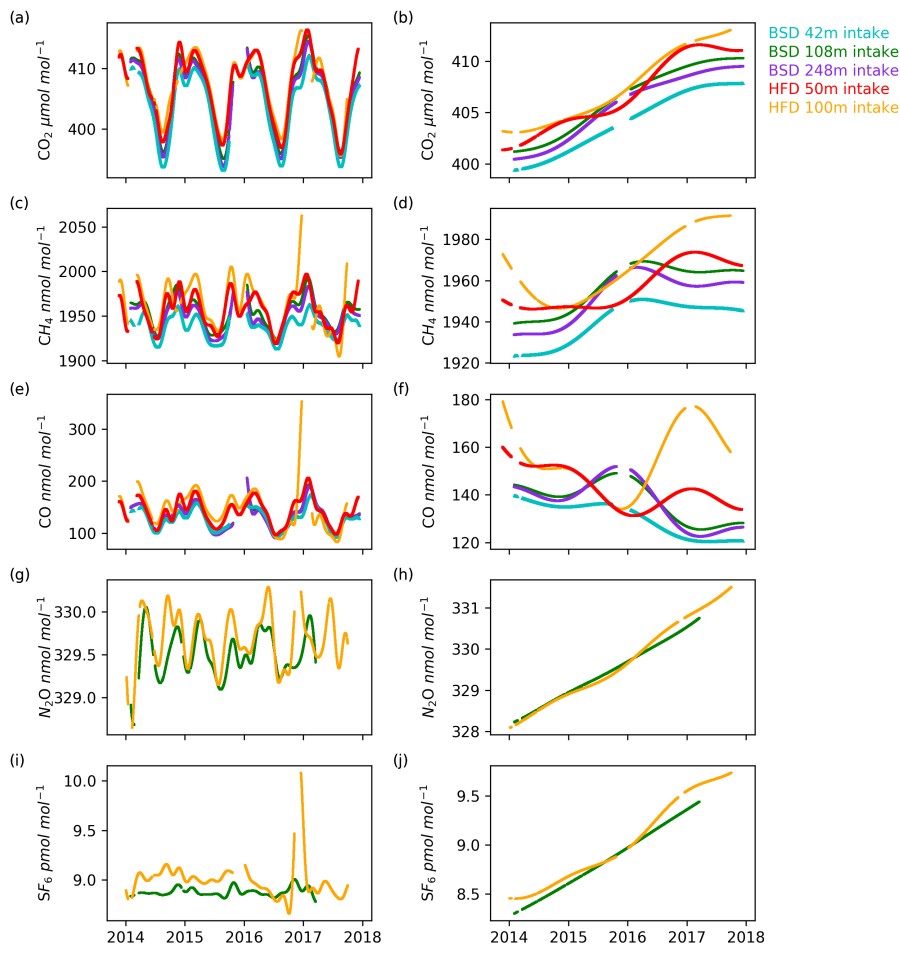

**Figure 5:** The seasonal cycle and long term trend for (a) & (b) $CO_2$, (c) & (d) $CH_4$, (e) & (f) CO, (g) & (h) $N_2O$ and (i) & (j) $SF_6$ of the Bilsdale 42m (blue), 108m (green) & 248m (purple) and Heathfield 50m (red) & 100m (yellow) intake heights. The large 2017 peak in the CO long term trend shown for the Heathfield 100m intake is an end effect artifact of the fitting process due to a large pollution event in early Jan 2017.



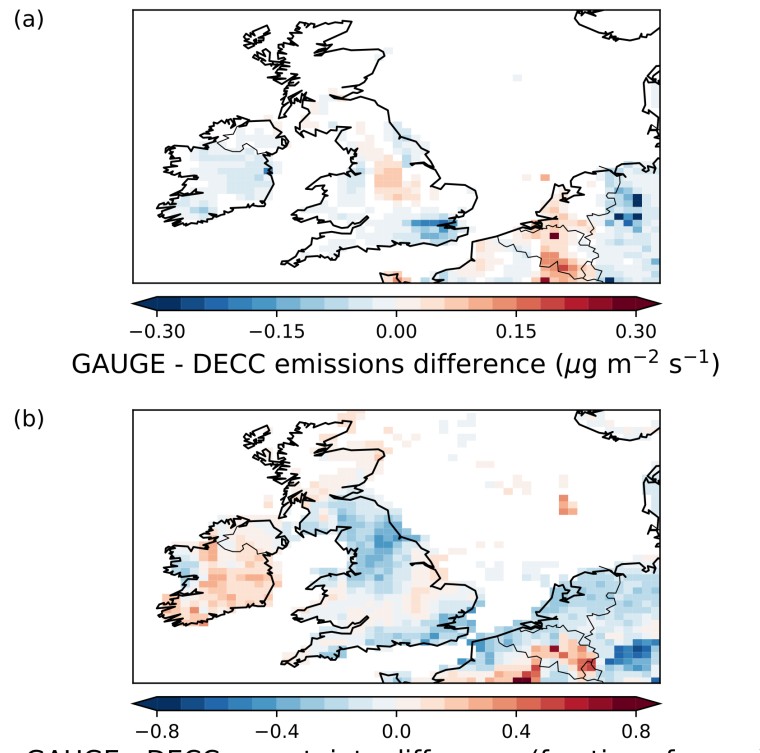

**Figure 6:** Maps of the posterior output from the rj-MCMC inversions showing (a): the difference in the spatial distribution of emissions between the 4-site DECC network inversion and the 6-site GAUGE network. Blue regions represent areas where emissions were reduced after the introduction of the additional data. (b) The difference in uncertainty between the DECC and GAUGE network inversions. Blue areas represent regions where the posterior uncertainty decreased after the introduction of the additional data, whilst red shows areas of increased uncertainty.





### (a) Experiment 1 & 2

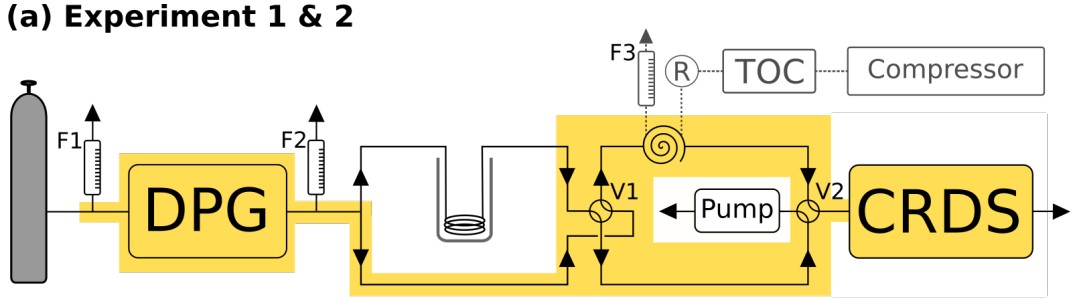

### (b) Experiment 3

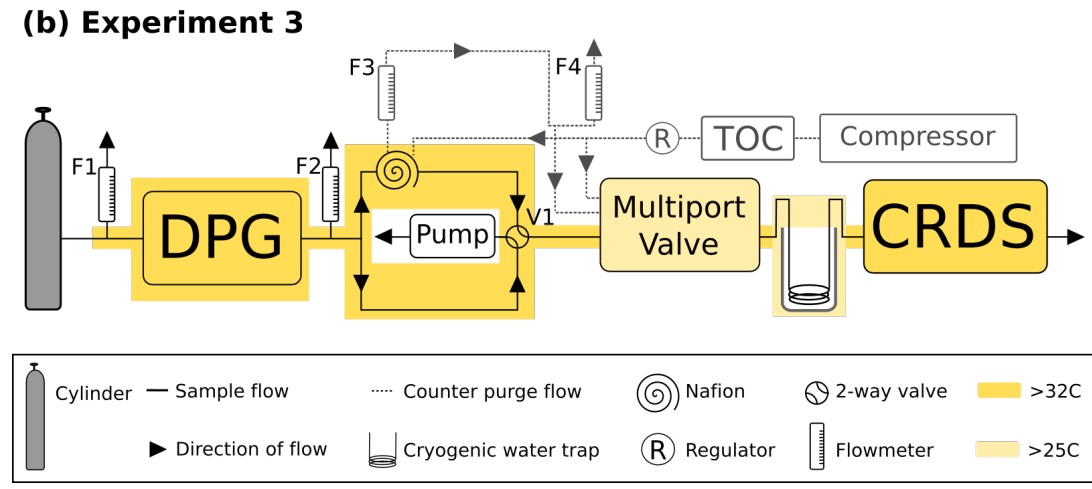

**Figure 7:** A schematic of the humidification system used in (a) Experiment 1 and 2 and (b) Experiment 3. The Black arrows and lines show the direction of sample gas flow. Grey dashed lines and arrows show the flow path of the Nafion counter purge gas. Heated zones are shown in yellow.





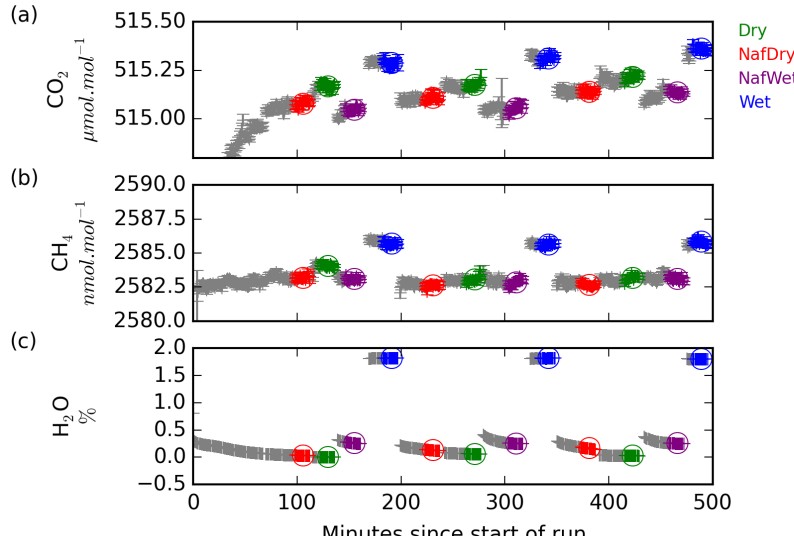

**Figure 8: The (a) CO$_2$, (b), CH$_4$ and (c) H$_2$O minute mean data obtained during measurements of cylinder UoB-04 with a DPG set point of 17.5°C. The raw data points are shown in grey with error bars representing the standard deviation of each minute mean, while the blue, purple, red and green points correspond to the stable periods of sample treatments Wet, NafWet, NafDry and Dry, respectively. The mean values used in further analysis are shown as open circles.**





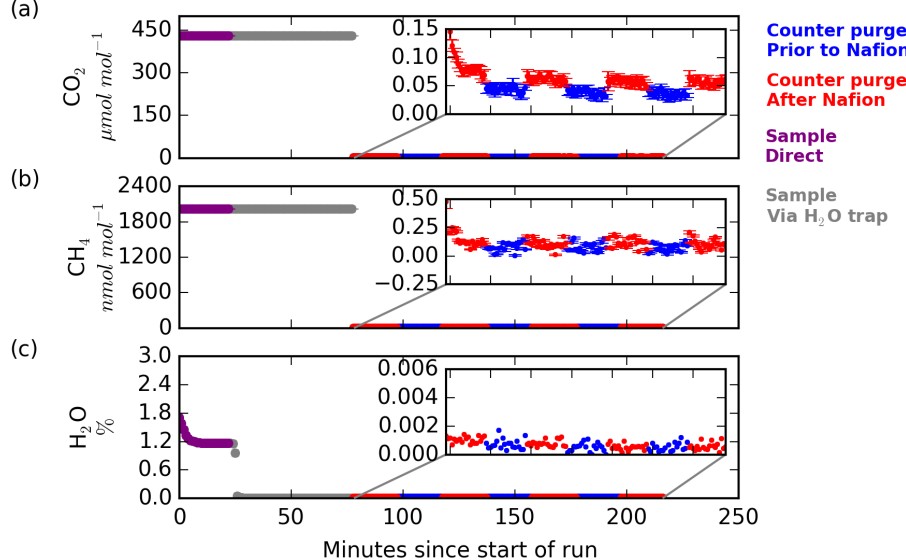

**Figure 9: The (a) $CO_2$, (b), $CH_4$ and (c) $H_2O$ minute mean data obtained during Experiment 3 for cylinder UoB-16 at a dewpoint of 10C. Error bars are ± 1 standard deviation of each minute mean. Purple and grey data points are the sample without and with the $H_2O$ trap, respectively, while blue and red data points are the Nafion counter purge before and after the Nafion, respectively.**





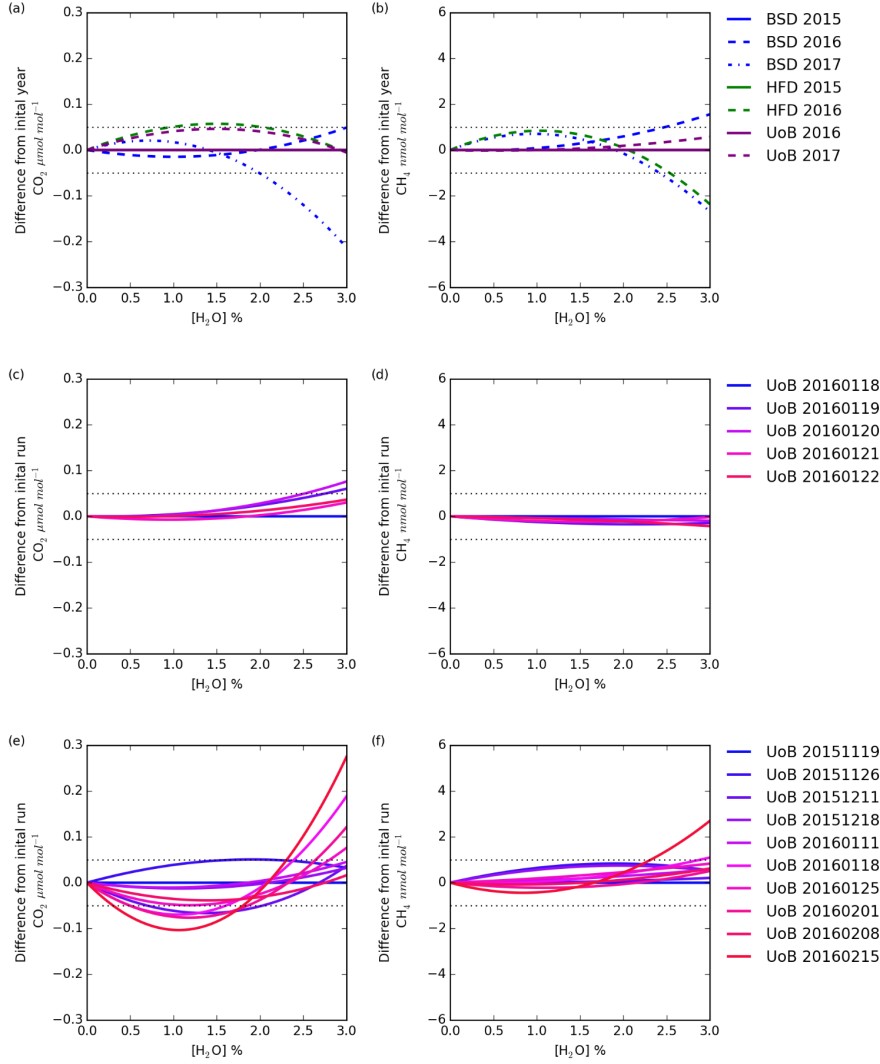

**Figure 10:** The change in the difference between dry mole fractions with water content calculated for $CO_2$ and $CH_4$ using (a) & (b) the first annual mean instrument specific water correction and subsequent annual corrections for each instrument and (c) & (d) the first individual water correction and subsequent corrections for the weekly and (e) & (f) daily tests conducted using UoB instrument.



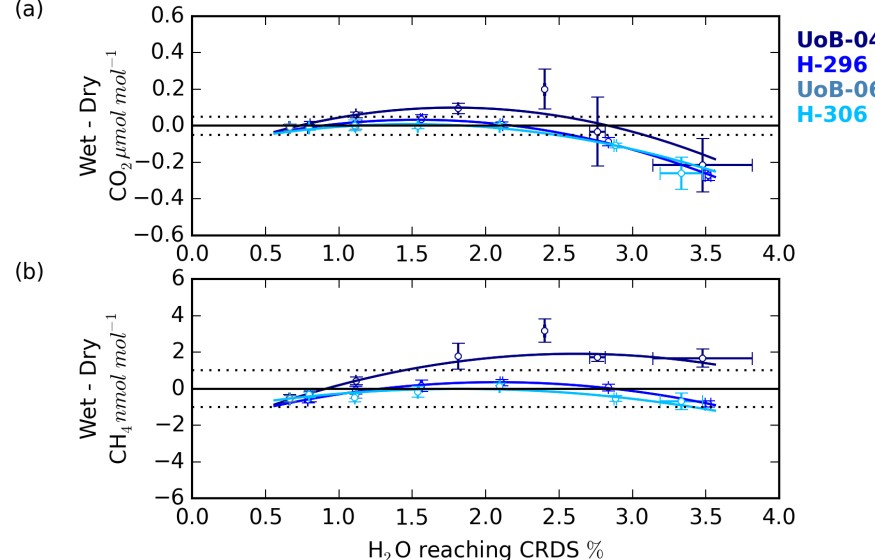

**Figure 11: The (a) CO₂ and (b) CH₄ change in the Wet – Dry sample treatment difference with sample water content for cylinders UoB-04, H-296, UoB-06 and H-306. Error bars are the larger of either the standard deviation of the mean difference or the uncertainties of the two sample types added together in quadrature.**



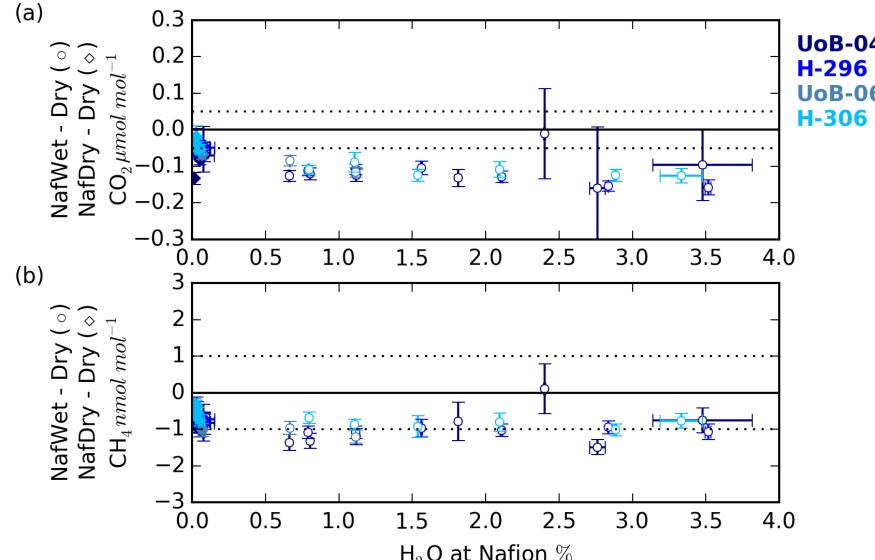

**Figure 12: The (a) CO₂ and (b) CH₄ change in the NafWet – Dry (open circle) and NafDry – Dry (closed diamond) sample treatment difference with sample water content for cylinders UoB-04, H-296, UoB-06 and H-306. Error bars are larger of either the standard deviation of the mean difference or the uncertainties of the two sample types added together in quadrature. Note the diamond markers are clustered near 0 % H₂O.**





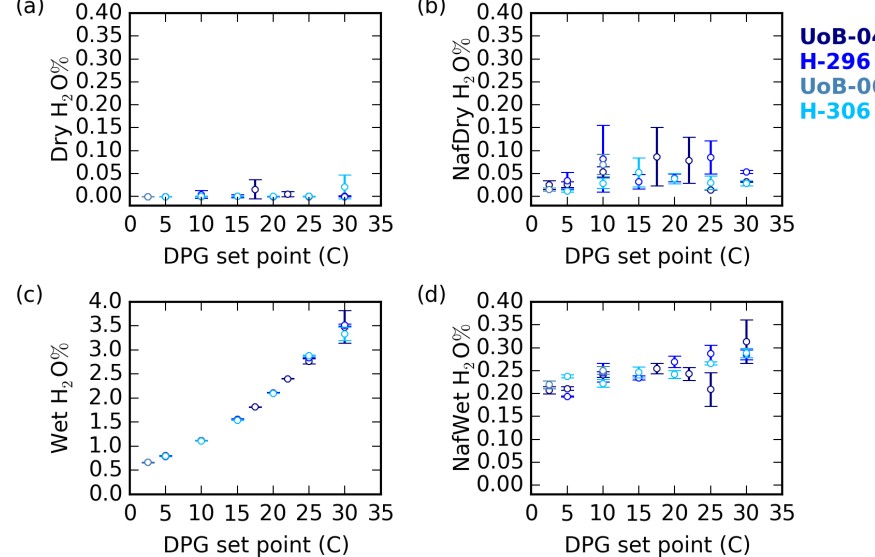

**Figure 13: The change in H₂O mole fractions with change in dew point generator set point for the (a) Dry, (b) NafDry, (c) Wet and (d) NafWet sample treatment type for sample cylinders UoB-04, H-296, UoB-06 and H-306. Error bars are ± 1 standard error n = 15.**





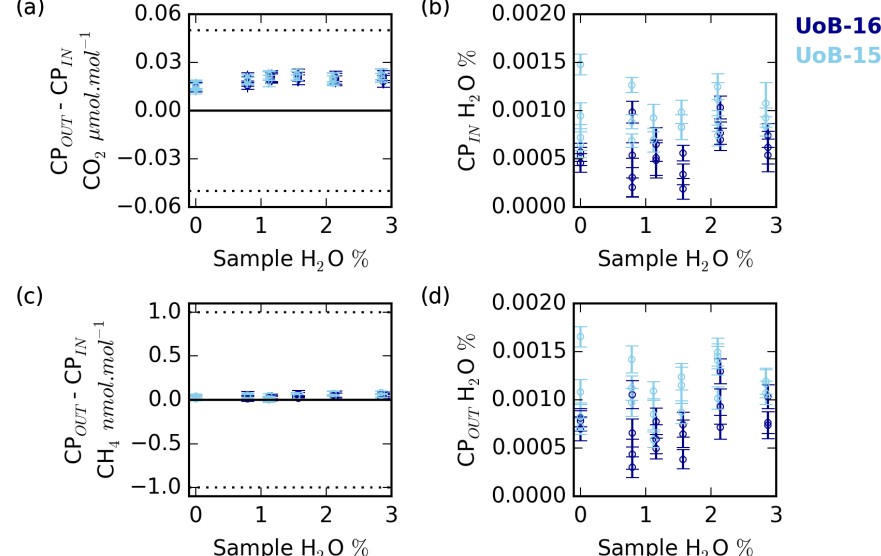

**Figure 14: Change in the counter purge in (CP$_{in}$) and out (CP$_{out}$) (a) CO$_2$ and (c) CH$_4$ mole fraction with sample water content for ambient (UoB-15) and above ambient (UoB-16) mole fraction cylinders. Change in (b) CP$_{in}$ and (d) CP$_{out}$ water content with changing sample water content. Note that the gas stream was cryogenically dried before analysis. Error bars are larger of either the standard deviation of the mean difference or the uncertainties of the two sample types added together in quadrature. The dotted lines in (a) and (c) are the respective WMO internal reproducibility guidelines.**



Table 1 – Instrument specific water corrections for the Bilsdale (BSD), Heathfield (HFD) and University of Bristol (UoB) CRDS instruments. The parameters shown are the mean ± the 95% confidence interval of tests repeated in triplicate. Water corrections labelled High and Low were determined using an above ambient and below ambient mole fraction cylinder, respectively, while the rest were determined using an ambient mole fraction cylinder. The mean residual is the mean of the absolute values of all residuals (1σ²).

| | | | A | | | B | | | Mean residual $CO_2$ μmol mol$^{-1}$ $CH_4$ nmol mol$^{-1}$ | n |
|---|---|---|---|---|---|---|---|---|---|---|
| $CO_2$ | BSD | 2015 | -0.0157 | ± | 0.0001 | 0.00018 | ± | 0.00008 | 0.016 (0.013) | 4 |
| | | 2016 | -0.01578 | ± | 0.00004 | 0.00022 | ± | 0.00002 | 0.016 (0.013) | 3 |
| | | 2017 | -0.01556 | ± | 0.00005 | 0.00008 | ± | 0.00002 | 0.018 (0.015) | 5 |
| | HFD | 2015 | -0.01559 | ± | 0.00007 | 0.00011 | ± | 0.00004 | 0.022 (0.023) | 3 |
| | | 2016 | -0.0154 | ± | 0.0001 | 0.00004 | ± | 0.00003 | 0.016 (0.017) | 1* |
| | | 2017 | | | | | | | | |
| | UoB | 2016 | -0.01577 | ± | 0.00007 | 0.00020 | ± | 0.00004 | 0.046 (0.06) | 17 |
| | | 2017 | -0.01558 | ± | 0.00008 | 0.00012 | ± | 0.00004 | 0.015 (0.015) | 3 |
| | | 2016 High | -0.0160 | ± | 0.0003 | 0.0003 | ± | 0.0001 | 0.054 (0.062) | 3 |
| | | 2016 Low | -0.01606 | ± | 0.00005 | 0.00030 | ± | 0.00002 | 0.038 (0.058) | 3 |
| $CH_4$ | BSD | 2015 | -0.0138 | ± | 0.0002 | 0.0005 | ± | 0.0001 | 0.204 (0.167) | 4 |
| | | 2016 | -0.0139 | ± | 0.0002 | 0.0006 | ± | 0.0001 | 0.204 (0.178) | 3 |
| | | 2017 | -0.01309 | ± | 0.00009 | 0.00014 | ± | 0.00002 | 0.199 (0.176) | 5 |
| | HFD | 2015 | -0.01274 | ± | 0.00002 | 0.00015 | ± | 0.00005 | 0.245 (0.217) | 3 |
| | | 2016 | -0.0119 | ± | 0.0005 | -0.0002 | ± | 0.0002 | 0.399 (0.310) | 1* |
| | | 2017 | | | | | | | | |
| | UoB | 2016 | -0.0139 | ± | 0.0001 | 0.00025 | ± | 0.00005 | 0.251 (0.244) | 17 |
| | | 2017 | -0.0139 | ± | 0.0001 | 0.00027 | ± | 0.00006 | 0.188 (0.199) | 3 |
| | | 2016 High | -0.01393 | ± | 0.00005 | 0.0004 | ± | 0.0001 | 0.054 (0.062) | 3 |
| | | 2016 Low | -0.01402 | ± | 0.00005 | 0.00028 | ± | 0.00008 | 0.038 (0.058) | 3 |

*The fitted parameter and 1σ² of a single test due to a leak in the septum





**Table 2 – CRDS calibration and standard cylinder mole fractions and usage start dates for the Heathfield (HFD) and Bilsdale (BDS) sites. Reported mole fractions are from the WCC-EMPA, Dübendorf, Switzerland. *Mole fraction measurement from GasLab MPI-BGC, Jena, Germany. ^Based on a single calibration episode.**

| | Cylinder | | CO$_2$ WMO x2007 $\mu$mol mol$^{-1}$ | CH$_4$ WMO x2004 nmol mol$^{-1}$ | CO WMO x2007 nmol mol$^{-1}$ | Start date – End date |
|---|---|---|---|---|---|---|
| BSD | Calibration Suite #1^ | Low | 379.2 | 1806.8 | 123.87 | 30/1/2014 -24/4/2015 |
| | | Ambient | 394.71 | 1889.4 | 131.13 | 20/2/2014 – 7/11/2015 |
| | | High | 456.51 | 2074.0 | 582.5 | 30/1/2014 – 24/4/2015 |
| | Calibration Suite #2* | Low | 379.51 | 1812.5 | 74.6 | 20/1/2016 – Current |
| | | Ambient | 418.63 | 2090.0 | 246.1 | 2/10/2015 – Current |
| | | High | 471.17 | 2404.8 | 469.2 | 2/10/2015 - Current |
| | Standard | H-239^ | 395.16 | 1900.8 | 117.58 | 30/01/2014 - 23/09/2014 |
| | | H-252 | 402.26 | 1905.7 | 140.91 | 23/09/2014 - 22/7/2015 |
| | | H-251 | 402.26 | 1906.3 | 145.0 | 22/7/2015-6/5/2016 |
| | | USN-20141394*^ | 399.31 | 1939.3 | 123.7 | 6/5/2016 -6/5/2016 |
| HFD | Calibration *^ | Low | 369.24 | 1845.9 | 128.78 | 16/12/2013 – Current |
| | | Ambient | 420.24 | 1993.8 | 321.7 | 16/12/2013 – Current |
| | | High1 | 441.26 | 2211.0 | 224.23 | 16/12/2013 - 27/1/2017 |
| | | High2 | 477.59 | 2282.1 | 104.65 | 24/2/2017 – Current |
| | Standard | H-240^ | 394.33 | 1881.6 | 121.15 | 16/12/2013 |
| | | H-255 | 402.12 | 1908.5 | 135.78 | 17/12/2014 |
| | | H-254 | 402.19 | 1908.2 | 138.72 | 21/10/2015 |
| | | H-285^ | 393.63 | 1927.9 | 104.65 | 21/9/2016 – Current |



**Table 3 – GC-ECD standard cylinder mole fractions and usage start dates**

| Site | Cylinder | N$_2$O SIO-16 nmol mol$^{-1}$ | SF$_6$ SIO-SF6 pmol mol$^{-1}$ | Start date | |
|------|----------|------|------|------|---|
| HFD | H-234 | 326.67 | 8.20 | 14/11/2013 | 5 |
| BDL | H-235 | 326.56 | 8.13 | 14/1/2014 | |
| | H-222 | 326.23 | 8.05 | 2/10/2015 | |





**Table 4 – The cylinders used during the humidification and UoB instrument specific water tests. Most measurements were made in-house and only corrected for linear drift against a standard calibrated at WCC-EMPA, Dübendorf, Switzerland and hence are simply indicative of the expected mole fractions. While those marked \* were calibrated at GasLab MPI-BGC, Jena, Germany and linked to the WMO x2007 $CO_2$ and x2004A $CH_4$ scales.**

| Test type | Cylinder | $CO_2$<br>µmol mol$^{-1}$ | $CH_4$<br>nmol mol$^{-1}$ |
|---|---|---|---|
| Humidification | H-306 | 372.5 | 1776 |
| | UoB-06 | 384.8 | 1975 |
| | UoB-15 | 399.3 | 1928 |
| | H-296 | 406.6 | 1947 |
| | UoB-16 | 430.7 | 2015 |
| | UoB-04 | 515.3 | 2585 |
| UoB instrument specific water correction | USN20104095* | 346.91 ± 0.06 | 1742.9 ± 0.3 |
| | H-283 | 379.1 | 1815 |
| | USN20104068* | 449.49 ± 0.05 | 2145.0 ± 0.4 |





**Table 5 – Cross- membrane transport (N$_{X\%}$) estimates at sample water content Y% from the difference of *NafDry* and *Dry* or *NafWet* and *Dry* runs and estimates of the error in the CRDS water correction at water mole fraction Y% (E$_{Y\%}$), where N$_{X\%}$ = Difference – E$_{Y\%}$. See text for further details. *Based on the 2016 replicate droplet test data ^Estimated from Reum et al. (2017)**

| | | | *NafDry -Dry* | *NafWet -Dry* |
|---|---|---|---|---|
| | | | $\overline{x} \pm 1\sigma^2$ | **Trend with increasing sample H$_2$O** |
| CO$_2$ μmol mol$^{-1}$ | | Difference | -0.05 ± 0.02 | Decreases from -0.08 to -0.16 |
| | E$_{Y\%}$ | Droplet test* | -0.12 ± 0.08 | Increases from -0.21 to -0.16 |
| | | Reum et al.^ | 0 | Decreases from -0.03 to -0.04 |
| | N$_{X\%}$ | Droplet test* | 0.07 ± 0.08 | Decreases from 0.13 to 0 |
| | | Reum et al.^ | -0.05 ± 0.02 | Decreases from -0.05 to -0.12 |
| | | Exp. 3 | -0.015 ± 0.003 | $\overline{x} \pm 1\sigma^2$ = -0.021 ± 0.003 (No trend) |
| CH$_4$ nmol mol$^{-1}$ | | Difference | -0.7 ± 0.2 | $\overline{x} \pm 1\sigma^2$ = -1.0 ± 0.3 (No trend) |
| | E$_{Y\%}$ | Droplet test* | - 0.6 ± 0.3 | Increases from -1.5 to -1.3 |
| | | Reum et al. | ~ 0 | Decreases from -0.5 to -0.7 |
| | N$_{X\%}$ | Droplet test | -0.1 ± 0.4 | Decreases from 0.5 to 0.3 |
| | | Reum et al. | -0.7 ± 0.2 | Increased from -0.5 to -0.3 |
| | | Exp. 3 | 0.03 ± 0.01 | $\overline{x} \pm 1\sigma^2$ = 0.04 ± 0.03 (No trend) |

