# Peer review of "UK greenhouse gas measurements at two new tall towers for aiding emissions verification"

_Atmospheric Measurement Techniques, 2018_

## Referee Comment (RC1) · Anonymous Referee #1 · 2 Aug 2018

Review Criteria: 1. Does the paper address relevant scientific questions within the scope of AMT? Yes, the paper provides a useful reference for greenhouse gas measurements at two sites and a discussion of systematic errors associated with water vapor that is important and timely.

2. Does the paper present novel concepts, ideas, tools, or data? It is worthwhile to document the analytical systems in use at these sites. Methods for dealing with water impacts on co2 and ch4 measurements are rapidly evolving and this group has worked hard to document and characterize systematic errors in their system. The paper describes a series of laboratory and field experiments that will provide useful information for researchers doing similar work.

3. Are substantial conclusions reached? The paper is overly long and the section

describing the experiments to characterize water vapor errors is difficult to follow. There are some significant conclusions, and the paper should be distilled to highlight those.

4. Are the scientific methods and assumptions valid and clearly outlined? The authors have worked hard to characterize their instrument, but substantial revision is needed to clarify the methods and results described in the paper. The statistical analysis of the data is flawed as described below.

5. Are the results sufficient to support the interpretations and conclusions? Generally yes, with some exceptions noted below.

6. Is the description of experiments and calculations sufficiently complete and precise to allow their reproduction by fellow scientists (traceability of results)? Generally yes.

7. Do the authors give proper credit to related work and clearly indicate their own new/original contribution? Generally yes, but a few key references are missing or incorrect as noted below.

8. Does the title clearly reflect the contents of the paper? yes

9. Does the abstract provide a concise and complete summary? yes

10. Is the overall presentation well-structured and clear? Improvement needed as described below.

11. Is the language fluent and precise? Improvement needed as described below.

12. Are mathematical formulae, symbols, abbreviations, and units correctly defined and used? Generally yes.

13. Should any parts of the paper (text, formulae, figures, tables) be clarified, reduced, combined, or eliminated? See specific suggestions below.

14. Are the number and quality of references appropriate? yes

15. Is the amount and quality of supplementary material appropriate? The description

of the laboratory experiments in the text should be shortened and distilled. Many of the details could be moved to the supplement.

Major comments:

A full uncertainty analysis is evidently planned for a future paper, but more work is needed in the current manuscript to summarize the various contributions to the uncertainty of the datasets described here. This manuscript provides many details about systematic errors related to water vapor, but a concise summary of the impacts on the measurements is lacking. Ideally the dataset would include time and site dependent estimates of this uncertainty component depending on the method of drying/water correction employed. Other uncertainty components such as larger errors for measurements outside of the calibration range should also be reported for individual measurements.

Although some of the uncertainties discussed here are significant compared to the WMO internal reproducibility guidelines, these errors/uncertainties are likely very small relative to the so-called observation (model-data mismatch) errors assigned in the inverse modeling. Some context about how the measurement uncertainty compares with model-representation errors would be helpful. It would also be helpful to see how the measurement uncertainty compares to signals of emissions. The WMO extended measurement compatibility goals should also be noted (+/- 0.2 ppm for CO2, +/- 5 ppb for CH4).

The statistical analysis of the time series data needs major improvement. The Thoning fit is not a good choice for this dataset, as is clearly evident in Figure 5. Specific suggestions are provided below.

The inversion analysis as currently presented is not compelling due to only minor reported improvement in total uncertainty and apparent flaws in the inversion framework that cause uncertainties in some regions to increase with additional data. Since AMT is not suitable for a detailed discussion of the inverse modeling methodology perhaps better to omit. A simpler presentation of how the additional sites improve the sensitivity to
surface fluxes could be substituted (i.e. a map of the total surface sensitivity/footprints estimated by the NAME model showing the impact of the additional sites).

Discussion of the lab water vapor tests is very hard to follow and should be reorganized and significantly shortened. Specific suggestions are given below. It is not necessary to exhaustively present results from experiments that were inconclusive in the body of the paper. Although researchers who are struggling with similar issues might benefit from this information, it should be relegated to the Supplement or to an Appendix in order to simplify the main paper. Despite having direct experience with analyzing results from these types of water corrections, I found the presentation difficult to understand.

Specific comments:

Page 1, line 27/28: "...this error is mostly calibrated out" is the 0.02 umol per mol error the remaining error after applying the calibration? Or is the nafion-related error « 0.02 ppm after calibration? In either case 0.02 ppm is nearly negligible and likely smaller than the total measurement error of the analytical system, which has not been adequately characterized.

Page 2, line 25: does AMT allow references to a paper in prep?

Page 6, line 18: CRDS dwell times at each level are surprisingly long

Page 7: line 1, "This air is dried to <0.005%..." I think this refers to the counterpurge air but not totally clear at this stage if it might refer to the sample air. It would be useful to state what is level of drying that is accomplished with the nafion for the GC channel.

Page 7: droplet test has weaknesses due to rapid changes of humidity that are inadequately resolved. Potential mismatchs/lags among co2/ch4/h2o channels. Also, I am not sure that Yver Kwok et al. 2015 is the best reference for this. I quickly checked and did not see any discussion of the droplet test in that paper. Maybe it would be better to cite the Rella 2013 AMT paper which describes several implementations of the droplet method. A citation for the Rella 2013 paper is currently lacking.

Page 7, line 15: A cylinder of air was not humidified. Instead, air from a cylinder was humidified.

Page 8: data collected in the first 5 minutes following the injection was excluded. What is the maximum humidity sampled after these data have been excluded? (i.e. is the maximum h2o value included in the fits significantly lower than the 4.5% value mentioned on page 7, line 20?) It is frustrating & confusing that the water corrections are discussed in multiple sections of this paper (here on page 8 and again in section 3.3.1).

Page 8, line 22: What are the calibration gases spiked with?

Page 9, line 1: It would be helpful to specify what is the maximum systematic error due to differences between sample and standard isotopic composition, since spiking standards can result in isotopic compositions that are outside the ambient range.

Page 9, line 5: What is the uncertainty associated with the non-linearity correction? Dynamic dilution is a complex procedure and likely to have non-negligible uncertainties. Why not use a set of gravimetric standards instead?

Page 10, line 9: It is not clear how the long-term repeatability numbers here are being computed. The parenthetical description (xbar minus +/- one sigma) is not an adequate description. Are these numbers the mean standard deviation computed over all the tanks over a year? In any case, instead of "long-term repeatability" it would be better to report the "long-term reproducibility", since a metric of the compatibility of the measurements over periods of months to years is needed. These terms are defined in the Guide to the expression of uncertainty in measurement: https://www.bipm.org/utils/common/documents/jcgm/JCGM_100_2008_E.pdf The difference between repeatability and reproducibility has to do with whether the conditions of the measurement are changed, and over timescales of months to years, standards are changing, ambient conditions, humidity levels are changing, etc. It seems unlikely that one could confidently interpret differences in CO2 in measurements made months or years apart at the level of 0.018 or 0.013 ppm.

Page 10, line 21: How was the cycle time threshold of 8 sec determined? Are those data that are filtered based on cycle time obviously bad? Page 11, section 2.2.2: The Thoning fits are most appropriate for remote sites, and the method often produces spurious results when gaps or large pollution events are present. These effects can be clearly seen in Figure 5. Figure 5 does not add any value to the paper and should be removed. More careful analysis is needed if seasonal cycles and trends are to be reported for these sites.

Page 12, line 19: Are the HFD mean $CO_2$ mole fraction of 407.5 ppm and the BSD mean of 404.7 the means over each entire record? If yes, then this quantity will seemingly be affected by the gaps and there is no use in reporting this quantity or interpreting the difference between the sites.

Page 12, line 25: The text implies that there are some high $CO_2$ events at BSD that do not have associated high CO. If that is the case then any such events are not likely to result from biomass burning and must have a different source (e.g. power plant plume).

Page 13, line 5: It seems unlikely that multi-day $CO_2$ enhancements resulting from pollution over London or Europe would not be associated with elevated CO. Could the elevated $CO_2$ result from advection of air from higher latitudes?

Page 13, line 10: The typical diurnal variation of $CO_2$ measured on tall towers is well understood, e.g., https://www.tandfonline.com/doi/pdf/10.3402/tellusb.v47i5.16070?needAccess=true

Page 14, Line 15: See comment above. The authors are correct that a major issue with the Thoning fitting routine is the underlying FFT, which requires interpolation across gaps. But since the fits are obviously flawed, why not use alternative methods to investigate the seasonal cycle and trends? A simple analysis using monthly mean or median values would be much more robust and simple to implement and explain. And/or box and whisker or fiddle plots could be used to describe the seasonal cycles and trends. Any months with significant gaps could/should be removed.

Page 14, line 20: It is not surprising that CO is decreasing. This has been widely reported. It would be interesting to compare the trends for all of these gases with background values from Mace Head.

Page 14, line 26: Again, box and whisker plots showing quantiles of the data would be helpful here to quantitatively describe any CO differences between sites. It is not useful to report he multi-year mean values without any uncertainties.

Page 15, N2O: There is a more recent Nevison et al paper that discusses N2O seasonality over the US (https://agupubs.onlinelibrary.wiley.com/doi/10.1002/2017GB005759)

Page 16, Line 13: SF6 seasonality might be driven by atmospheric transport.

Page 16, Line 23: The additional sites appear to have negligible impact on the inversions. The estimate of total emissions does not change significantly, and the uncertainties are only slightly reduced.

Page 17, discussion of Figure 6: Perhaps additional panels could be included to show the posterior flux distribution for the GAUGE and/or DECC cases. Figure 6a could be revised to show the magnitude of the redistribution relative to the mean. Otherwise it is some work for the reader to understand whether the redistribution is significant.

How can the addition of more data make the inversion estimate more uncertain in some regions? I think this can only be the case if the inversion framework underestimates the uncertainty with the 4-tower case. Some additional explanation is needed. Perhaps it would help to indicate which regions are significantly constrained using a footprint/sensitivity map.

Page 19, Line 27: Apparent typo. . . .lasted between 2-5 (minutes?)

Page 26, discussion of figure S3: The average residuals given in Table 1 do not adequately describe the uncertainty indicated in these plots.

Page 26, line 13: The reported large difference between the humidity of samples and

standards is a bit concerning. Could this be mitigated by using a longer nafion drier? And/or a chiller could be used to remove the bulk of the sample humidity upstream of the nafion drier as is done in the NOAA tall tower systems.

Page 26, line 15: "However it is likely to be a systematic offset of the order of -0.05 to -0.1 ppm..." based on what evidence? The residuals in S3? Or the Reum tests? Or something else?

Page 26, last paragraph: Are these below ambient and above ambient results shown somewhere?

Page 27, line 10: It is surprising that the maximum humidity at these sites is <2%.

Page 27, line 12: I understand that when the Nafion drier was installed then many of the air samples have H2O < 0.1%, but what are the implications? This paragraph is discussing Figure 10 a & b, and this particular figure does not seem to show any troubling implications for samples with H2O < 0.1%.

Page 27, line 22: It seems very impractical to calibrate high-humidity sites weekly. Also, I am not convinced that the droplet test is accurate at very high humidity. The daily tests at U of Br in Figure 10e show extremely large variability at humidity > 2.5%.

Page 28, line 6: Why was this test not performed at H2O < 0.7% since that is where the droplet test is unreliable due to rapidly changing H2O?

Page 28, line 21: The range of humidity for HFD and BSD here is different than given on Page 27 line 10 (max of 2.5% instead of 2.0%). Meanwhile the Introduction states that the DECC/GAUGE network observes samples with humidity of up to 3.5%.

Page 29, line 8: Clarify 0.5 to 3.5% in the Wet experiment versus < 0.31% for the others.

Page 29 & 30, discussion of Figures 12 & 13 and table 2. This discussion is extremely hard to follow. Since it seems that the droplet tests are highly uncertain below about

<0.3%, then consider just omitting that discussion and sticking with the estimates from Reum. Or perhaps omit this discussion altogether. The tepid conclusion at the top of page 31 does not warrant the amount of discussion present in the current manuscript. If the experiments failed or were inconclusive then why include them? If you feel it is important for the community to understand the pitfalls of your attempt in order to avoid similar futile attempts, then perhaps relegate this discussion to the supplement.

Page 29, line 23: "Although small, these changes are an order of magnitude smaller. . ." This is puzzling. . .did you mean to write: ". . .these changes are an order of magnitude larger"?

Page 31, line 25: The measured difference of 0.02 ppm is practically negligible and unlikely to impact any conceivable scientific analysis, except perhaps analysis of spatial gradients in the high southern hemisphere.

Page 32, line 3: "These cylinders are very dry, $H_2O$ < 0.0001%..." But, above on page 26, it states that standards are significantly humidified by the nafion, and so the difference should be even smaller than the 0.005 ppm that you report.

Page 32, line 8: "This is not the case for the water correction bias, which varies with $H_2O$ (section 3.3.2)."

Page 32, line 22: Unfortunately, the inversion results as presented here indicate only marginal improvement in the flux estimates. However this is likely due to limitations of the inversion framework. In particular, it seems that the uncertainty estimated for the 4 tower network is too small.

Page 33, line 33: I think the should read 0.015 umol/mol instead of 0.15.

Table 1: Mean residual is not an adequate diagnostic of the uncertainty, since the residuals vary strongly as a function of $H_2O$. Perhaps include an example plot for one of these calibration instances in the Supplement.

Table 2: ISO format for dates is YYYY-MM-DD

Figure 2: Include definition of TOC in the legend or the caption. It is mentioned in the text but should also be noted here.

Figure 8 & 9: Essentially no discussion of these figures is provided. Either describe these figures or omit.

Figure 11: Include the CO2 and CH4 values of the cylinders in the legend or the caption.

Table 2 & Table 3: What is the uncertainty of the assigned values for the calibration standards? The reproducibility of the NOAA scale for CO2 is estimated to be 0.03 ppm. But how well do the calibration centers propagate the scale?
* * *

---

## Referee Comment (RC2) · Anonymous Referee #2 · 7 Aug 2018

Journal: AMT

Title: UK greenhouse gas measurements at two new tall towers for aiding emissions verification

Author(s): Ann R. Stavert et al.

MS No.: amt-2018-140

MS Type: Research article

Special Issue: Greenhouse gAs Uk and Global Emissions (GAUGE) project (ACP/AMT inter-journal SI)

1. Does the paper address relevant scientific questions within the scope of AMT? Yes.

[Figure]

(see also 13. below)

2. Does the paper present novel concepts, ideas, tools, or data? Yes, valuable laboratory and field experiments/data.

3. Are substantial conclusions reached? Yes, but not for all parts of the paper (see also 13. below).

4. Are the scientific methods and assumptions valid and clearly outlined? Yes for methods – but the description of the work must be restructured and edited in order to improve clarity for the readers.

5. Are the results sufficient to support the interpretations and conclusions? Yes (for water / drying part).

6. Is the description of experiments and calculations sufficiently complete and precise to allow their reproduction by fellow scientists (traceability of results)? Yes – but I did not see anywhere the information from where the data is available.

7. Do the authors give proper credit to related work and clearly indicate their own new/original contribution? Yes.

8. Does the title clearly reflect the contents of the paper? No (see 13. below)

9. Does the abstract provide a concise and complete summary? Yes

10. Is the overall presentation well-structured and clear? No – text too long, parts should be omitted.

11. Is the language fluent and precise? Not always.

12. Are mathematical formulae, symbols, abbreviations, and units correctly defined and used? Yes.

13. Should any parts of the paper (text, formulae, figures, tables) be clarified, reduced, combined, or eliminated? Paper is too long and consists of several parts (description of

new measurement systems, emissions verification, errors related to drying of sample stream) of which the latter part is addressed in most detail and is in my opinion also the most interesting/relevant (and best fitting to the scope of AMT). In some parts, the discussion is difficult to follow. The authors should focus on the part related to water correction and keep the other parts to a minimum (just keep enough to support the drying part) or omit them.

14. Are the number and quality of references appropriate? Yes.

15. Is the amount and quality of supplementary material appropriate? Most parts that will be omitted, can find their place in Supplement.

General comments

The paper reports on the installations of GHG measurement systems at two new tall tower sites, describes their setups and their contribution to uncertainty reduction for UK CH4 emission estimates. In addition, the paper describes extensive tests and evaluates the impact of two different sample air drying strategies applied for CRDS instruments in the lab and installed at these two sites. From these subjects, I consider the latter the most relevant and best described in the paper. The text's length, the writing style and the text structure make it often difficult to follow the work and reasoning of the authors. This regarding, substantial improvements are necessary (i.e. shorten and restructure the text). Also, two of the three instruments measure carbon monoxide (CO) – why was this opportunity not used to evaluate the effects of drying / water corrections on the measurements of CO as well? The experiments and findings related to air drying strategies represent an important contribution to this field of research and should thus be accepted for publication after major corrections and only if the comments within this review are adequately and fully addressed.

Specific comments and technical corrections

Note on Technical corrections: in some cases, I have marked a word or formatting only

once, but make sure to apply the corrections throughout the text where relevant.

Page 1, Line 15 (1/15): "...were located...": I would rather go for "are located"

1/20: "by between" -> "by up to 20 %"

1/29: "..default factory.." it is well established that the default factory water correction is not to be used for the purposes described – this fact should be stated more clearly here.

2/17: write out the names of the gas molecules at first use.

3/3: need "While,"?

3/10: delete "recognised"

3/16: A dramatic reduction in the cost of these instruments could be debated. I suggest you keep only the next sentence, starting in line 17.

4/4: Please explain why dry zero air was chosen for the counter flow purge gas (instead of using the reflux method with sample air from the instrument's outlet). This is relevant because it is on one side true that the drying with dry (zero) air is more effective, but on the other side trace gas species gradients can influence the transition dynamics of these trace species across the Nafion membrane and thus the composition of the dried sample air as well. If I am not much mistaken, Nafion must be written as Nafion®.

5/18: correct to: "m a.g.l."

6/2: it would be advantageous, if you stated at least once the serial numbers of the Picarro instruments used in your setups.

7/7: Please summarize here briefly the reason for discontinuing the Nafion drying. State the main arguments/problems that are thus avoided or minimized. If the reader jumps to Section 3.3.3 to find out about it s/he will have a hard time finding the explanation. Sentence in 7/12 is similarly ambiguous.

7/26: Note: Picarro analyzers are not calibrated for H2O measurements and often show different positive or negative values close to zero at third place behind the comma. For the sake of completeness of information, it would be informative to know what the zero was for water on all three Picarro instruments.

8/2: please add a short justification/explanation on how the criteria were decided on (particularly the thresholds for 1 min mean standard deviations).

8/11: I am not convinced by the nomenclature for gases; what do you mean with "close-to-ambient standard"? From the continuation of the text, I guess this is a "working tank" that helps you to account/correct for short-term drift? You should then use "standard gas" for what you now call "calibration gas".

8/17: "greatly increases the error" – what does this explicitly mean? How much? Better reformulate, drop "greatly" and, if possible, add facts-based estimates.

8/21: no "target gas" seems to be used – for the appropriate QC, this is required. There is a comment on this in Chapter 4. – but it would be helpful if the authors comment on how the absence of a target gas is influencing the current performance of the system (i.e. what is the impact on the uncertainties, etc.).

9/8: In Table 3, the pre- and post-site calibration values should be given (where applicable).

10/2: The statement "...with the mean absolute precision increasing ( i.e. becoming less precise)" seems wrong to me.

10/22: the numbers for pressure and temperature are switched – correct.

12/15: 2.3 Results and discussion; 3.3 has the same name, which is confusing. A paper should have one Results/Discussion section.

13/19: What does "...green space..." refer to?

14/8-19: I can only second the comments by Anonymous Referee #1 on this subject.

14/23: I am reserved about calling a 3.5 years long data set as sufficient for calculating long-term trend curves.

15/2: put spaces between number and unit (e.g., 108 m).

17/25: you can safely drop "extremely low" without losing any information. What were the values for CO?

18:21: this would be an excellent place to state the maximum humidity of air in the cylinders.

19/3: typo in the flow range

19/14: analogously to I.D.; $\frac{1}{4}$" O.D.

19/17: please specify "polymeric plastic tubing" better and explain why it was necessary to include it (particularly as this introduces further possible "active" surfaces that can influence the performance of the system).

19/27: typo "2-5"

20/7: please be more explicit than "were monitored closely and regularly logged."

20/27: a flowmeter rather "monitors' than "ensures"

21/2: it would be helpful for the reader, if you added how this value translates to % H2O

21/7: please specify "stabilise".

21/16: I am not sure what you mean here – if we are talking about H2O < 0.0001 %, we go far below a dew point of 0 deg.C – please explain.

21/19: please clarify the frequency 1.5 Hz/data recording strategy here (as the Picarros record with 0.4 Hz).

22/7: you should number the equations

24/21: use "target gas"

25/14: wrong use of c.f. (cf) – use vs.

27/6: "WMO internal reproducibility"; also, decide what you want to use: internal reproducibility guidelines/requirements/bounds and use throughout the text consistently. Please also explain somewhere (best at first use) why you (correctly) aim for Internal reproducibility and not for the Recommended compatibility.

30/8: can you please explain in a bit more detail how you did this – and how you were able to estimate Nafion-related errors, since, to my knowledge, Reum et al. (2017) did not use one?

31/23: would you be able to discuss the role/importance of the trace gas species gradients between sample and counter flow gas? (see 4/4)

32/3: see 21/16

37: North and scale are missing.

38: all abbreviations used should be explained in the figure's caption

39: I presume "Hour of day" means local time? Better replace with "Hour (LT)". Was summer / winter time used as well?

41: see 14/8-19 comment regarding spikes

Figure S3: equations not legible

———————————————————

---

## Author Response (AR1)

**Reply to the comments of reviewer #1 of "UK greenhouse gas measurements at two new tall towers for aiding emissions verification"**

Ann R. Stavert1, Simon O'Doherty2, Kieran Stanley2, Dickon Young2, Alistair J. Manning3, Mark F. Lunt5, Christopher Rennick4 and Tim Arnold4,5.

1 Climate Science Centre, CSIRO Oceans & Atmosphere, Aspendale, VIC, 3195, Australia

2 School of Chemistry, University of Bristol, Bristol, BS8 1TS, United Kingdom

3 Met Office, Exeter, Devon, EX1 3PB, United Kingdom

4National Physical Laboratory, Teddington, Middlesex, TW11 0LW, United Kingdom

5 School of GeoSciences, University of Edinburgh, Edinburgh, EH9 3FF, UK

**General comments:**

A full uncertainty analysis is evidently planned for a future paper, but more work is needed in the current manuscript to summarize the various contributions to the uncer- tainty of the datasets described here. This manuscript provides many details about systematic errors related to water vapor, but a concise summary of the impacts on the measurements is lacking. Ideally the dataset would include time and site dependent es- timates of this uncertainty component depending on the method of drying/water correc- tion employed. Other uncertainty components such as larger errors for measurements outside of the calibration range should also be reported for individual measurements.

Although some of the uncertainties discussed here are significant compared to the WMO internal reproducibility guidelines, these errors/uncertainties are likely very small relative to the so-called observation (model-data mismatch) errors assigned in the in- verse modeling. Some context about how the measurement uncertainty compares with model-representation errors would be helpful. It would also be helpful to see how the measurement uncertainty compares to signals of emissions. The WMO extended measurement compatibility goals should also be noted (+/- 0.2 ppm for CO2, +/- 5 ppb for CH4). The statistical analysis of the time series data needs major improvement. The Thoning fit is not a good choice for this dataset, as is clearly evident in Figure 5. Specific suggestions are provided below.

The inversion analysis as currently presented is not compelling due to only minor re- ported improvement in total uncertainty and apparent flaws in the inversion framework that cause uncertainties in some regions to increase with additional data. Since AMT is not suitable for a detailed discussion of the inverse modeling methodology perhaps bet- ter to omit. A simpler presentation of how the additional sites improve the sensitivity to surface fluxes could be substituted (i.e. a map of the total surface sensitivity/footprints estimated by the NAME model showing the impact of the additional sites). Discussion of the lab water vapor tests is very hard to follow and should be reorganized and significantly shortened. Specific suggestions are given below. It is not necessary to exhaustively present results from experiments that were inconclusive in the body of the paper. Although researchers who are struggling with similar issues might benefit from this information, it should be relegated to the Supplement or to an Appendix in order to simplify the main paper. Despite having direct experience with analyzing results from these types of water corrections, I found the presentation difficult to understand.

The authors would like to thank reviewer 1 for their helpful comments. We have endeavoured to incorporate all the suggestions that they have made. The locations of changes to the paper are given as Page Number, Line number of the new paper while the reviewers comments contain the locations in the original paper.

As requested by the reviewer a summary of the errors associated with the drying and water correction methods used at the Heathfield and Bilsdale sites is now included (see Table 5). Estimates of the error associated with making observations outside the range of the calibration suite has also been included (Page 14, lines 13 - 23). Discussion of these errors relatively to the WMO extended compatibility goals, model-data mismatch error and the magnitude of emissions signals have also been included (Page 28, lines 20 - 26). As advised by the reviewer we have significantly altered the statistical analysis, no longer

using the Thoning et al. approach and instead using box plots of linearly detrended data. References to specific changes are included in the replies to the specific comments below. Also, as suggested by the reviewers, due to the measurement focused scope of AMT we have removed details of the inversion modelling and its results. These will be discussed more fully in a later paper.

Following the advice of both reviewers we have removed a large section of the text relating to the first set of Nafion® drying tests and the accompanying figures. As such some technical corrections are no longer applicable, these have been noted as such. We have also simplified and reordered the discussion of the water correction work. The paper has been significantly restructured with the Nafion® and water correction work no longer a separate section. Instead they are incorporated in to the main experimental and discussion sections. Even with the inclusion of the extra information requested by the reviewers these changes have reduced the length main body of the paper by 6 pages and removed 4 figures.

**Specific comments:**

Page 1, line 27/28: "...this error is mostly calibrated out" is the 0.02 umol per mol error the remaining error after applying the calibration? Or is the nafion-related error « 0.02 ppm after calibration? In either case 0.02 ppm is nearly negligible and likely smaller than the total measurement error of the analytical system, which has not been adequately characterized. The 0.02ppm error is the error prior to carrying out the calibration. We expect the error following the calibration to be ≪ 0.01ppm. This has been clarified in the text. It is important to note that this error is only removed through calibration because the calibration cylinders are also passed through the Nafion® which may not be the case for all monitoring sites. As such we feel it is important to highlight and quantify this possible source of error. We would politely disagree with the reviewer, and suggest that a 0.02ppm bias is far from negligible - a bias of this magnitude represents 40% of the WMO Southern Hemisphere inter-laboratory compatibility guideline. As the reviewer correctly notes, other measurement errors associated with an analytical system are most likely to be larger, hence, in conjunction with this bias they could well lead to a significant error within a data set.

As outlined in the text, a separate paper currently in preparation will characterise the measurement error of the analytical system of these two new sites along with the other four sites within the UK GHG monitoring network.

**Page 2, line 25: does AMT allow references to a paper in prep?**

We have removed this reference.

**Page 6, line 18: CRDS dwell times at each level are surprisingly long**

The length of the CRDS sampling period at each height was designed to obtain as long as possible period of continuous data at each height during an hour. That is, for the site with three intakes 20 minutes was spent at each height during each hour. Such a sampling strategy is not unusual for greenhouse gases, for example the INFLUX experiment measured from some intakes for up to 40 minutes of every hour (Richardson et al., 2017).

Page 7: line 1, "This air is dried to <0.005%..." I think this refers to the counterpurge air but not totally clear at this stage if it might refer to the sample air. It would be useful to state what is level of drying that is accomplished with the nafion for the GC channel. We have clarified and added extra information to the text.

Page 6, line 21 - 23, "The counter purge is dried to < 0.005 % H2O by the compressor (50 PLUS M, EKOM, Slovak republic) and a gas generator for total organic carbon systems (TOC-1250, Parker Balston, USA). Previous examinations of this drying method have found that samples are dried to < 0.0002 % H2O (Young, 2007)."

Page 7: droplet test has weaknesses due to rapid changes of humidity that are inadequately resolved. Potential mismatchs/lags among co2/ch4/h2o channels. Also, I am not sure that Yver Kwok et al. 2015 is the best reference for this. I quickly checked and did not see any discussion of the droplet test in that paper. Maybe it would be better to cite the Rella 2013 AMT paper which describes several implementations of the droplet method. A citation for the Rella 2013 paper is currently lacking.

We thank the reviewer for highlighting this oversight. We have replaced the reference to Yver Kwok et al. 2015 with a reference to Rella 2013 (Page 7, line 20).

Page 7, line 15: A cylinder of air was not humidified. Instead, air from a cylinder was humidified.

Corrected

Page 8: data collected in the first 5 minutes following the injection was excluded. What is the maximum humidity sampled after these data have been excluded? (i.e. is the maximum h2o value included in the fits significantly lower than the 4.5% value mentioned on page 7, line 20?) It is frustrating & confusing that the water corrections are discussed in multiple sections of this paper (here on page 8 and again in section 3.3.1).

The text has been clarified to include

Page 8, line 8, "This 5-minute cut-off reduced the maximum  $H_2O$  value included in the fit to 4 %  $H_2O$ ."

Following the restructuring of the paper, the water correction work is now split into two components, an experimental (Sections 2.3.4, 2.3.5 and 2.3.6) which outlines how the water correction tests were conducted and results & discussion (Sections 3.3 and 3.4) which details the results of the experiments and their implications.

Page 8, line 22: What are the calibration gases spiked with? The below has been added to the text.

Page 15, lines 4 -10, "The cylinder spiking and filling techniques of the calibration cylinders varied. The Heathfield calibration suite and the second Bilsdale calibration suite were filled at GasLab MPI-BGC Jena and consisted of natural air spiked using a combination of pure CO2 and a commercial mixture of 2.5 % CH4 and 0.5 % CO in synthetic air. The "high" calibration cylinder of the first calibration suite used at the Bilsdale site was filled with peak-hour ambient air at EMPA, Dübendorf, Switzerland while the "low" and "mid" cylinders were based on Mace Head air, in the case of the "low" this was diluted with scrubbed natural air."

**Page 9, line 1: It would be helpful to specify what is the maximum systematic error due to differences between sample and standard isotopic composition, since spiking standards can result in isotopic compositions that are outside the ambient range.**

We cannot comment directly on our measurement error as we do not have measurements of the isotopic composition for our calibration gases. However, we refer readers to other papers who investigate this topic in detail and give an estimate based on the literature. Page 15, lines 13-19, "The effect of an isotopic mismatch between the calibration standards and the sample has been examined in detail by Flores et al. (2017), Griffith (2018) and Tans et al. (2017). With Griffith (2018) showing that, for a sample of 400 µmol mol-1 CO2 and 2000 nmol mol-1 CH4, the error will range between 0.001 – 0.155 µmol mol-1 CO2 and 0.1 – 0.7 nmol mol-1 CH4 depending on the magnitude of the sample to standard mismatch. As such, we expect a worst-case scenario estimate of the error associated with our measurements < 0.04 % for both CO2 and CH4."

**Page 9, line 5: What is the uncertainty associated with the non-linearity correction? Dynamic dilution is a complex procedure and likely to have non-negligible uncertainties. Why not use a set of gravimetric standards instead?**

These non-linearity corrections are typically exceptionally stable with Hammer et al. (2008) finding differences of < 0.2 nmol mol-1 N2O between non-linearity corrections determined 10 years apart and over the range of interest (326 to 340 nmol mol-1 N2O) highly linear. While, studies including Hall et al. (2011) and van der Laan et al. (2009) have found SF6 to be very linear at mole fractions > 4 pmol mol-1 and for the instrumental response function to be very stable over time. As such, we expect the uncertainty of the non-linearity correction to be small. Although not discussed in detail in this paper this uncertainty will be addressed in the associated uncertainty analysis paper as outlined in the introduction.

Page 2, lines 23 – 26, "A further paper, currently in preparation, will discuss the integration of these new sites with the existing UK Deriving Emissions linked to Climate Change (DECC) network (Stanley et al., 2018) funded by the UK Department of Business, Energy and Industrial Strategy (BEIS) and provide a full uncertainty analysis for data collected at all the DECC/GAUGE sites."

The UoB group, along with other groups (e.g. the Schauinsland GAW station) have a history of successfully using dynamic dilution for the quantification of the N2O and SF6 non-linearity. The laboratories at the BSD and HFD sites are physically small making it difficult to store or use a set of dedicated gravimetric N2O and SF6 standards at each site and shipping a set of cylinders between the sites would be logistically difficult considering the remote locations of and limited access to the sites. In contrast, the dynamic dilution method is able to generate multiple calibration points using just two cylinders and is logistically simple to deploy and ship between sites. As such, the decision was made to use this approach. We have also added the information below

Page 15, line 23 – Page 16, line 3, "This approach, dynamic dilution, has a history of use in similar field locations (Hammer et al., 2008) and is able to generate multiple calibration points using just two cylinders. This greatly reduces the number of cylinders needed, a key concern for space-limited locations like BSD and HFD. An assessment of the uncertainty associated with this non-linearity approach will be included in a future paper currently in preparation. However, previous studies (Hall et al., 2011;van der Laan et al., 2009;Hammer et al., 2008) have found the ECD detector response to be extremely stable over time and very linear for both SF6 and N2O in the mole fraction range typical of the HFD and BSD stations. As such, we expect the uncertainty of the nonlinearity correction to be very small."

Page 10, line 9: It is not clear how the long-term repeatability numbers here are being computed. The parenthetical description (xbar minus +/- one sigma) is not an adequate description. Are these numbers the mean standard deviation computed over all the tanks over a year? In any case, instead of "long-term repeatability" it would be better to report the "long-term reproducibility", since a metric of the compatibility of the measurements over periods of months to years is needed. These terms are defined in the Guide to the expression of uncertainty in measurement:

https://www.bipm.org/utils/common/documents/jcgm/JCGM\_100\_2008\_E.pdf The difference between repeatability and reproducibility has to do with whether the conditions of the measurement are changed, and over timescales of months to years, standards are changing, ambient conditions, humidity levels are changing, etc. It seems unlikely that one could confidently interpret differences in CO2 in measurements made months or years apart at the level of 0.018 or 0.013 ppm.

We thank the reviewer for identifying the incorrect use of repeatability. We have now corrected this to reproducibility throughout the text.

A reproducibility of minute mean values at the level of 0.02  $\mu$ mol mol-1 CO2 over a period of years would indeed be unlikely however we are reporting the reproducibility of a 20-minute mean value which is much more stable. We have more clearly defined how these numbers have been calculated. See

Page 17, lines 3 - 4, "The long-term reproducibility of a 20-minute mean was estimated as the mean standard deviation of the daily 20-minute measurements of 4 standard cylinders at each site."

**Page 10, line 21: How was the cycle time threshold of 8 sec determined? Are those data that are filtered based on cycle time obviously bad?**

The 8-second cycle time threshold was determined after a close examination of the typical cycle times and their relationship to other key parameters. Longer cycle times were typically related to periods of more variable cavity pressure and outlet valve parameters. Measurements of cylinders with long cycle times were also typically significantly different, between 0.5 and 50  $\mu$ mol mol-1 CO2 from the mean cylinder value.

Page 11, section 2.2.2: The Thoning fits are most appropriate for remote sites, and the method often produces spurious results when gaps or large pollution events are present. These effects can be clearly seen in Figure 5. Figure 5 does not add any value to the paper and should be removed. More careful analysis is needed if seasonal cycles and trends are to be reported for these sites.

The Thoning fits and Figure 5 have been removed from the paper and replaced with a seasonal cycle analysis using box plots and simple trend estimate based on a linear fit to the data.

Page 18, Lines 2-6, "The long term trend in mole fraction at each site was estimated as the mean linear trend in the minute mean data over the period 2014-2017, inclusive. Seasonal and diurnal trends in the data were assessed using monthly and hour-of-day box plots of hourly means of detrended minute-mean data developed using the Python matplotlib.boxplot package. Here the long-term trend was removed by using a least-squares fit between a quadratic and the minute mean data."

Page 12, line 19: Are the HFD mean CO2 mole fraction of 407.5 ppm and the BSD mean of 404.7 the means over each entire record? If yes, then this quantity will seemingly be affected by the gaps and there is no use in reporting this quantity or interpreting the difference between the sites.

We thank the reviewer for noting this. References to mean mole fractions calculated in this manner have been removed.

Page 12, line 25: The text implies that there are some high CO2 events at BSD that do not have associated high CO. If that is the case then any such events are not likely to result from biomass burning and must have a different source (e.g. power plant plume).

We thank the reviewer for this observation and have included it in the text. Page 18, Lines 17 - 19, "While events that do not show corresponding high CO mole fractions, the majority of which occur in the higher two intakes, are likely to be driven by less local CO2 sources, for example power plants."

**Page 13, line 5: It seems unlikely that multi-day CO2 enhancements resulting from pollution over London or Europe would not be associated with elevated CO. Could the elevated CO2 result from advection of air from higher latitudes?**

We thank the reviewer for this comment as it demonstrates that our text was not written clearly. We had meant to say that the difference in pollution patterns, short sporadic events vs. prolonged periods of increased background mole fraction, was not common to all gases between the two sites rather than not common between the gases at an individual site. In fact, coincident 2-3 day periods of increased CO,  $CH_4$ ,  $N_2O$  and  $SF_6$  were observed at HFD. This has been clarified in the text.

Page 18, Lines 20 - 26, "HFD is located in southern England, just south of London (Figure 1). Here, high  $CO_2$  events are typically longer — 2 to 3 days — and coincide with elevated  $CH_4$ , CO,  $N_2O$  and  $SF_6$ . Rather than appearing as peaks superimposed on a background value these periods have a positive shift in the entire diurnal cycle, suggesting a change in the background mole fraction. Air histories, based on the output of the Numerical Atmospheric dispersion Modelling Environment (NAME) Lagrangian dispersion model, outlined in Manning et al. (2011), for these periods of elevated  $CO_2$  typically show the source of the air to be from over London or Europe."

Page 13, line 10: The typical diurnal variation of CO2 measured on tall towers is well understood, e.g., https://www.tandfonline.com/doi/pdf/10.3402/tellusb.v47i5.16070?needAccess=true We have clarified our text and included a reference to this paper. Page 19, Lines 1 -6, "Both sites show a clear relationship between  $CO_2$  mole fraction and intake height with the lowest height generally having the most elevated mole fractions, followed by the higher heights (Figures 3a & 4a). This trend, also apparent for CH4 and CO (Figures 3c & e and Figure 4c & e), is typical of tall tower measurements and is driven by proximity to surface sources (Bakwin et al., 1998). The gradient in  $CO_2$  and CH4 mole fraction is most apparent during the early hours of the morning (see Figure 3b & d and Figure 4b & d) when the boundary layer is the lowest."

Page 14, Line 15: See comment above. The authors are correct that a major issue with the Thoning fitting routine is the underlying FFT, which requires interpolation across gaps. But since the fits are obviously flawed, why not use alternative methods to investigate the seasonal cycle and trends? A simple analysis using monthly mean or median values would be much more robust and simple to implement and explain. And/or box and whisker or fiddle plots could be used to describe the seasonal cycles and trends. Any months with significant gaps could/should be removed.

All references to the Thoning fit have been removed and replaced with a discussion of seasonal cycles based on box plots.

Page 14, line 20: It is not surprising that CO is decreasing. This has been widely reported. It would be interesting to compare the trends for all of these gases with background values from Mace Head.

The -7 nmol mol-1 yr-1 trend observed in HFD and BSD CO is slightly larger than the -5 nmol mol-1 yr-1 trend apparent over the same period in the CO data collected at Mace Head. However, as HFD and BSD are on a different calibration scale to the MHD it is not possible to determine whether this is a real difference or a calibration artefact. As such, we decided not to include this in the text of the paper. However, as all three sites use common calibration scales for CO2, CH4, N2O and SF6 comparisons between the growth rates of these gases have been added to the text.

Page 20, Lines 1-7, "Of the 5 gases measured at HFD and BSD, CO is the only gas to show a decrease in mole fraction between 2013 and 2017, roughly -7 nmol mol-1 yr-1. In contrast, the CO2 and CH4 data increase by 2-3  $\mu$ mol mol-1 yr-1 and 5-9 nmol mol-1 yr-1 respectively, varying on the intake height. These agree well with the ~2  $\mu$ mol CO2 mol-1 yr-1 and ~8 nmol CH4 mol-1 yr-1 trends observed at Mace Head (MHD, 53.327 °N, -9.904 °E, Figure 1), a remote site within the UK DECC network located on the west coast of Ireland. The CO data collected at MHD is not on the NOAA x2014 CO calibration scale making direct comparisons between growth rates at the three sites meaningless."

Page 20, Lines 20-22, "The long-term trend, ~0.8 nmol  $N_2O$  mol-1 yr-1 (calculated using data from the 108 m and 100 m intakes at BSD and HFD over the period of coincident data collection, 2014 to mid 2016) also agrees well between the two sites and with MHD, also ~0.8 nmol  $N_2O$  mol-1 yr-1."

Page 21, Linese 17-18, "The long-term trend in the SF6 mole fraction at BSD and HFD shows a gradual increase of 0.3 pmol mol-1 yr-1 again agreeing well with MHD which showed an identical growth rate."

Page 14, line 26: Again, box and whisker plots showing quantiles of the data would be helpful here to quantitatively describe any CO differences between sites. It is not useful to report the multi-year mean values without any uncertainties.

This has been corrected and a discussion based on box plots included. Page 20, Lines 8-10, "While the range of minute mean CO mole fractions was significantly larger at BSD, 63 to 9500 nmol mol-1 than HFD, 60 to 4850 nmol mol-1, the high CO values observed at BSD were relatively rare. This is reflected in the smaller quantile spread of the BSD data compared with the HFD data (Figure 7e & f)."

**Page 15, N2O: There is a more recent Nevison et al paper that discusses N2O seasonality over the US (https://agupubs.onlinelibrary.wiley.com/doi/10.1002/2017GB005759) We thank the reviewer for highlighting this new Nevison paper. However, as our paper, and the original Nevison reference, focuses on the UK while the new paper examines the USA**

**Page 16, Line 13: SF6 seasonality might be driven by atmospheric transport.**

we feel the original paper is the more appropriate reference.

This increase in pollution events of SF6 during the winter months is seen across the combined DECC/GAUGE network (4 sites across the UK but not MHD) and is very intriguing. However, the air history maps for these events do not consistently indicate any one area in the UK or the continent suggesting that this is an emission rather than transport related increase. This has triggered further investigations in the inventory community but no conclusions have yet been made.

Page 21, Lines 21 -25, "This seasonal shift occurs across the wider DECC-GAUGE network and air history maps suggest that it is not associated with an obvious UK or continental region. As such, instead of an atmospheric transport driven shift we believe this to be a true change in emissions and hypothesise that this may be due to increased load on, and hence increased failure of, the electrical switchgear during the colder months."

Page 16, Line 23: The additional sites appear to have negligible impact on the inversions. The estimate of total emissions does not change significantly, and the uncertainties are only slightly reduced.

**Section removed.**

Page 17, discussion of Figure 6: Perhaps additional panels could be included to show the posterior flux distribution for the GAUGE and/or DECC cases. Figure 6a could be revised to show the magnitude of the redistribution relative to the mean. Otherwise it is some work for the reader to understand whether the redistribution is significant.

How can the addition of more data make the inversion estimate more uncertain in some regions? I think this can only be the case if the inversion framework underestimates the uncertainty with the 4-tower case. Some additional explanation is needed. Perhaps it would help to indicate which regions are significantly constrained using a footprint/sensitivity map. Following the comments of both reviewers the decision was made to remove the discussion of the effect of the two towers on UK flux estimates. This will be discussed in a separate paper.

Page 19, Line 27: Apparent typo. . . .lasted between 2-5 (minutes?) Section removed.

Page 26, discussion of figure S3: The average residuals given in Table 1 do not adequately describe the uncertainty indicated in these plots.

Table 1 now includes the interquartile range which better expresses the range of the uncertainties as indicated in plot S3.

Page 26, line 13: The reported large difference between the humidity of samples and standards is a bit concerning. Could this be mitigated by using a longer nation drier? And/or a chiller could be used to remove the bulk of the sample humidity upstream of the nation drier as is done in the NOAA tall tower systems.

Yes, the calibration gases could be further humidified using a longer Nafion® or the samples further dried using a chiller system. This suggestion has been incorporated into the conclusions and future work section (Page 28, Lines 15 - 16).

Page 26, line 15: "However it is likely to be a systematic offset of the order of -0.05 to -0.1 ppm. . ." based on what evidence? The residuals in S3? Or the Reum tests? Or something else?

This estimate was based on the residuals shown in Figure S3 (now Figure S5). This has been clarified in the text.

Page 23, Lines 17-20, "However, for BSD and HFD, assuming that the residuals of the droplet water tests are an accurate reflection of the likely error (Figure S3, we expect there to be a systematic offset of the order of -0.05 to -0.1  $\mu$ mol mol-1 CO2 and -1 to -2 nmol mol-1 CH4."

Page 26, last paragraph: Are these below ambient and above ambient results shown somewhere?

The coefficients for the above and below ambient tests are shown in Table 1 and plots of the residuals are included in the Supplement, Figure S6.

Page 27, line 10: It is surprising that the maximum humidity at these sites is <2%. This is a typo. It should have been 2.5%. This has been corrected (Page 25, Line 11).

Page 27, line 12: I understand that when the Nafion drier was installed then many of the air samples have H2O < 0.1%, but what are the implications? This paragraph is discussing Figure 10 a & b, and this particular figure does not seem to show any troubling implications for samples with H2O < 0.1%.

This paragraph was meant to convey that in light of the variability of the water correction over time, particularly at higher water contents, using a Nafion dryer to obtain a relatively low and stable sample water content was an advantage. This has been clarified.

Page 24, Lines 14 - 16, "In light of the temporal variability of the water correction over time, particularly at higher water contents, using a Nafion® dryer or alternative drying method to obtain a relatively low and stable sample water content would be an advantage."

Page 27, line 22: It seems very impractical to calibrate high-humidity sites weekly. Also, I am not convinced that the droplet test is accurate at very high humidity. The daily tests at U of Br in Figure 10e show extremely large variability at humidity > 2.5%.

As the reviewer has noted, performing weekly water calibrations would be a burdensome task and the droplet test can be unreliable at high humidities. This has been included in the text.

Page 24, Lines 24 – 27, "The impracticality of such a frequent testing regime along with the apparent unreliability of the droplet test at  $H_2O > 2.5$  % (See Figure S3 g) mean that an

alternative method, possibly partial drying, or a higher level of uncertainty may need be applied to measurements made at higher water contents."

Page 28, line 6: Why was this test not performed at H2O < 0.7% since that is where the droplet test is unreliable due to rapidly changing H2O?

We used the lowest stable set point of the dewpoint generator,  $2.5^{\circ}$ C, which equated to a water content of 0.6-0.7 %. As such, we were unable to use the dew point generator to examine water contents < 0.7 %.

Page 28, line 21: The range of humidity for HFD and BSD here is different than given on Page 27 line 10 (max of 2.5% instead of 2.0%). Meanwhile the Introduction states that the DECC/GAUGE network observes samples with humidity of up to 3.5%. The values reported on Page 27, line 10 was a typo. This has been corrected. Other sites within the DECC/GAUGE network, specifically the Angus site in Scotland, experienced humidities > 2.5%.

Page 29, line 8: Clarify 0.5 to 3.5% in the Wet experiment versus < 0.31% for the others. Section removed

Page 29 & 30, discussion of Figures 12 & 13 and table 2. This discussion is extremely hard to follow. Since it seems that the droplet tests are highly uncertain below about <0.3%, then consider just omitting that discussion and sticking with the estimates from Reum. Or perhaps omit this discussion altogether. The tepid conclusion at the top of page 31 does not warrant the amount of discussion present in the current manuscript. If the experiments failed or were inconclusive then why include them? If you feel it is important for the community to understand the pitfalls of your attempt in order to avoid similar futile attempts, then perhaps relegate this discussion to the supplement.

As suggested by the reviewer we have removed this section and the accompanying figures and table.

Page 29, line 23: "Although small, these changes are an order of magnitude smaller. ..." This is puzzling. . .did you mean to write: ". . .these changes are an order of magnitude larger"? As suggested by the reviewer we have removed this section and the accompanying figures and table.

Page 31, line 25: The measured difference of 0.02 ppm is practically negligible and unlikely to impact any conceivable scientific analysis, except perhaps analysis of spatial gradients in the high southern hemisphere.

Yes, as the reviewer has stated these differences are small, particularly in the light of the magnitude of the signals observed in the UK regional network. However, small errors can compound and individually may be a cause of concern at "clean air" locations as such we feel it is important to highlight them.

Page 32, line 3: "These cylinders are very dry, H2O < 0.0001%..." But, above on page 26, it states that standards are significantly humidified by the nafion, and so the difference should be even smaller than the 0.005 ppm that you report.

The relationship between water content and CO2 exchange discussed in this section and plotted on the X axis of Figure 9 is in reference to the water content of the sample (or standard) prior to it entering the Nafion®. Yes, the standards are humidified by the Nafion® but the CO2 exchange occurs prior/during this humidification process.

**Page 32, line 8: "This is not the case for the water correction bias, which varies with H2O (section 3.3.2)."**

This sentence has been removed.

Page 32, line 22: Unfortunately, the inversion results as presented here indicate only marginal improvement in the flux estimates. However this is likely due to limitations of the inversion framework. In particular, it seems that the uncertainty estimated for the 4 tower network is too small.

At the suggestion of the reviewers references to the inversion study have been removed from the paper.

Page 33, line 33: I think this should read 0.015 umol/mol instead of 0.15. Yes, this was incorrect. We thank the reviewer for noting this. It has been corrected Page 28, Line 6.

Table 1: Mean residual is not an adequate diagnostic of the uncertainty, since the residuals vary strongly as a function of H2O. Perhaps include an example plot for one of these calibration instances in the Supplement.

The plots of all of these calibration instances are included in the supplement – see Figures S5 and S6. In an effort to encompass the variability of the residual Table 1 has also been altered to include not just the mean residual but the 25th and 75th percentiles.

Table 2: ISO format for dates is YYYY-MM-DD

The date format in Table 2 (now Table 5) has been corrected.

**Figure 2: Include definition of TOC in the legend or the caption. It is mentioned in the text but should also be noted here.**

This has been noted in the caption of Figure 2.

**Figure 8 & 9: Essentially no discussion of these figures is provided. Either describe these figures or omit.**

Figure 8 has been removed. Figure 9 has been moved to the supplement (now Figure S4) and referred to specifically Page 11, Lines 22, 23, 28, Page 12, Line 4 and Page 13, Line7.

**Figure 11: Include the CO2 and CH4 values of the cylinders in the legend or the caption.**

These values are now included in the caption of Figure 9 (Now Figure 11). "The (a)  $CO_2$  and (b)  $CH_4$  change in the Wet – Dry sample treatment difference with sample water content for cylinders UoB-04 (515.3 µmol mol-1  $CO_2$  and 2585 nmol mol-1  $CH_4$ ), H-296 (406.6 µmol mol-1  $CO_2$  and 1947 nmol mol-1  $CH_4$ ), UoB-06 (384.8 µmol mol-1  $CO_2$  and 1975 nmol mol-1  $CH_4$ ) and H-306 (372.5 µmol mol-1  $CO_2$  and 1776 nmol mol-1  $CH_4$ ). Error bars are the larger of either the standard deviation of the mean difference or the uncertainties of the two sample types added together in quadrature."

Table 2 & Table 3: What is the uncertainty of the assigned values for the calibration standards? The reproducibility of the NOAA scale for CO2 is estimated to be 0.03 ppm. But how well do the calibration centres propagate the scale?

The uncertainties provided by the calibration centres have been added to Table 3. The values reported in Table 4 were made at the AGAGE Mace Head laboratory or the University of Bristol laboratory. An uncertainty assessment for scale propagation in this manner is outside the scope of this paper but will be included in the future uncertainty paper.

**General Comments**

The paper reports on the installations of GHG measurement systems at two new tall tower sites, describes their setups and their contribution to uncertainty reduction for UK CH4 emission estimates. In addition, the paper describes extensive tests and evaluates the impact of two different sample air drying strategies applied for CRDS instruments in the lab and installed at these two sites. From these subjects, I consider the latter the most relevant and best described in the paper. The text's length, the writing style and the text structure make it often difficult to follow the work and reasoning of the authors. This regarding, substantial improvements are necessary (i.e. shorten and restructure the text). Also, two of the three instruments measure carbon monoxide (CO) – why was this opportunity not used to evaluate the effects of drying / water corrections on the measurements of CO as well? The experiments and findings related to air drying strategies represent an important contribution to this field of research and should thus be accepted for publication after major corrections and only if the comments within this review are adequately and fully addressed.

The authors would like to thank Reviewer 2 for their helpful comments. We have endeavoured to reply to and incorporate all the suggestions that they have made. As requested by the reviewer we have removed and simplified large sections of the text. As such some technical corrections are no longer applicable, these have been noted as such. Reviewer 2 commented on the lack of discussion of the impact of Nafion® drying and the CRDS water correction on CO mole fractions. As the Nafion® drying experiments were conducted using a Picarro CRDS G2301, which does not measure CO, the effect of Nafion® drying on CO mole fractions was not investigated. While instrument specific CO water corrections were calculated, the large minute-mean variability inherent in the G2401 CO measurements (> 4 nmol mol-1) meant that the difference between the instrument specific and in-built correction was not statistically significant (Page 22, Lines 10-14). As such, these corrections were not presented in the body of the paper, however, further information can be found in Figure S5 of the supplementary. Even with the inclusion of the extra information requested by the reviewers these changes have reduced the length main body of the paper by 6 pages and removed 4 figures.

**Specific comments and technical corrections**

Page 1, Line 15 (1/15): "...were located...": I would rather go for "are located"

Removed.

1/20: "by between" -> "by up to 20 %" Section removed

1/29: "..default factory.." it is well established that the default factory water correction is not to be used for the purposes described – this fact should be stated more clearly here. Reference to default factory corrections has been removed from the abstract. Details of the issues associated with this correction are given in the introduction. Page 3, Lines 15-20, "Initially, it was hoped that the inbuilt water correction would remove the need for sample drying, inherent in most other methods (e.g. FTIR or NDIR) but subsequent studies questioned its stability over time and between instruments (Yver Kwok et al., 2015;Chen et al., 2010;Winderlich et al., 2010). In response to this, researchers have typically developed their own water corrections or have returned to sample drying in order to minimise the effect (Welp et al., 2013;Winderlich et al., 2010;Schibig et al., 2015)."

2/17: write out the names of the gas molecules at first use. Completed as requested Page 25, Line 15.

3/3: need "While,"? Section removed.

3/10: delete "recognised" Deleted.

3/16: A dramatic reduction in the cost of these instruments could be debated. I suggest you keep only the next sentence, starting in line 17. Sentence deleted.

4/4: Please explain why dry zero air was chosen for the counter flow purge gas (instead of using the reflux method with sample air from the instrument's outlet). This is relevant because it is on one side true that the drying with dry (zero) air is more effective, but on the other side trace gas species gradients can influence the transition dynamics of these trace species across the Nafion membrane and thus the composition of the dried sample air as well.

Page 3, Line 22 – 25, "For ease of servicing, the CRDS instrumentation at GAUGE and UK DECC Network sites was initially deployed using an identical drying method to that of the co-located GC instrumentation. This method relied on drying the sample with a Nafion® water permeable membrane in combination with dry zero air as a counter purge gas."

If I am not much mistaken, Nafion must be written as Nafion®. We have replaced all references to "Nafion" with "Nafion®".

5/18: correct to: "m a.g.l." Corrected throughout text.

**6/2: it would be advantageous, if you stated at least once the serial numbers of the Picarro instruments used in your setups.**

This information has been added to the text.

Page 5, Line 20, "Both sites are equipped with a CRDS (G2401 Picarro Inc., USA, CFKADS2094 and CFKADS2075 deployed at Bilsdale and Heathfield, respectively) making high frequency (0.4 Hz)  $CO_2$ ,  $CH_4$ , CO and  $H_2O$  measurements."

7/7: Please summarize here briefly the reason for discontinuing the Nafion drying. State the main arguments/problems that are thus avoided or minimized. If the reader jumps to Section 3.3.3 to find out about it s/he will have a hard time finding the explanation. Sentence in 7/12 is similarly ambiguous.

These have been clarified.

Page 6, Line 25 to Page 7, Line 6. "In an attempt to minimise the water correction required for dry mole fraction CRDS measurements, CRDS samples were initially dried using a Nafion® in an identical manner to those of the GC-ECD. This resulted in air samples with water mole fractions between 0.05 and 0.2 % H2O depending on the original moisture content of the air. However, due to concerns that the mole fraction gradient between the sample and the Nafion® counter purge might lead to CO2 transport across the Nafion® membrane this drying approach was discontinued. The CRDS Nafion® drying systems were removed on the 17th of June 2015 & 30th of September 2015 at BSD and HFD, respectively and undried air analysed and the data post corrected with an instrument specific water correction."

**And**

Page 7, Lines 18 - 19, "Motivated by the possibility of CO2 transport across the Nafion® membrane, the decision was made to measure wet samples and correct using an instrument specific water correction."

7/26: Note: Picarro analyzers are not calibrated for H2O measurements and often show different positive or negative values close to zero at third place behind the comma. For the sake of completeness of information, it would be informative to know what the zero was for water on all three Picarro instruments.

The below information has been added to the text.

Page 8, Lines 23-27, "As Picarro analysers are not calibrated for  $H_2O$  measurements when measuring dry air they often show different positive or negative values close to zero. For the instruments studied in this paper these "zero-water" values were 0.00001, -0.0003 and -0.002 for the Bilsdale, Heathfield and University of Bristol laboratory instruments respectively. These values were determined using measurements of cylinders of dry air where the first 120 minutes were ignored and the "zero-water" value calculated as the mean  $H_2O$  of the subsequent data (> 60 min)."

**8/2: please add a short justification/explanation on how the criteria were decided on (particularly the thresholds for 1 min mean standard deviations).**

The below information has been added to the text.

Page 7, Lines 4-9, "Data collected in the first five minutes immediately following the injection, the typical line equilibration period, were excluded from the fit. This avoids using data adversely effected by the effect of rapid changes in  $H_2O$  content on the cell pressure sensor, as identified by Reum et al. (2018) and the erroneous post-injection  $CO_2$

enhancement identified by (Rella et al., 2013). Again, due to cell pressure sensor concerns, data points with minute-mean  $H_2O$  standard deviations > 0.5 %  $H_2O$  were excluded. This 5-minute cut-off reduced the maximum  $H_2O$  value included in the fit to 4 %  $H_2O$ ."

**8/11: I am not convinced by the nomenclature for gases; what do you mean with "close- toambient standard"? From the continuation of the text, I guess this is a "working tank" that helps you to account/correct for short-term drift? You should then use "standard gas" for what you now call "calibration gas".**

As the reviewer has noted the standard cylinder is used to account for short-term drift and as such could also be called a "working tank" (Page 14, Line 7). The text has been updated to include a reference to this term. However, we would prefer to leave the labelling of the calibration cylinders unchanged. These cylinders are calibrated to WMO calibration scales and used to propagate these scales to the atmospheric observations (i.e. calibrate) as such we feel "calibration cylinders" is the most appropriate terminology. The use of the nomenclature "standard cylinder" and "calibration cylinders" in this way is also consistent with other related publications (e.g. Stanley et al. (2018)) and as such we feel it is important to maintain it.

**8/17: "greatly increases the error" – what does this explicitly mean? How much? Better reformulate, drop "greatly" and, if possible, add facts-based estimates. Added to the text.**

Page 14, Lines 13 - 23, "Assigning mole fractions to values outside the range of the calibration suite will increase the error. The magnitude of this error will depend on the magnitude of the mole fraction difference between the lowest calibration cylinder and the sample. This error has been estimated using measurements made at the Heathfield site of cylinders of known CO mole fractions, 6 and 57 nmol mol-1 CO below the lowest calibration cylinder. These show a percentage error of 2.41 and 3.09 %, respectively. A similar assessment of the error associated with samples above the highest calibration standard were made using cylinders 87 and 686 nmol mol-1 CO above the highest calibration standard. These correspond to percentage errors of 2.98 and 2.56%, respectively. As all the minute-mean CO measurements below of the calibration range are within 57 nmol mol-1 of the lowest calibration cylinder and the vast majority of minute-mean CO measurements above the calibration cylinder (99%) we expect that this error would typically be < 3%."

**8/21: no "target gas" seems to be used – for the appropriate QC, this is required. There is a comment on this in Chapter 4. – but it would be helpful if the authors comment on how the absence of a target gas is influencing the current performance of the system (i.e. what is the impact on the uncertainties, etc.).**

Target tanks, as defined by the World Meteorological Organisation (WMO, 2016), are an extremely useful quality control tool and "function as a warning that there might be a problem" but they are "not to be used to define a second, optional, path of traceability". As such, although they can be used to assess the uncertainty of the measurements, their use (or not) does not contribute to the uncertainty of any measurements. With this in mind, the authors believe that, although the use of a target tank would be a useful addition to the system, the lack thereof does not negatively impact the uncertainties of the measurements it merely impacts our ability to characterise these uncertainties.

This has been clarified in the text.

Page 29, Lines 3 - 8, "Future improvements to the Bilsdale and Heathfield records include the addition of target tanks at the sites. Although the use of target tanks do not directly influence measurement uncertainty they allow independent long-term monitoring of instrument performance and are a useful tool for assessing measurement uncertainty. The development of a full uncertainty analysis incorporating such target tank measurements, along with an assessment of the calibration strategy and any instrumental, water correction and sampling errors is also planned."

9/8: In Table 3, the pre- and post-site calibration values should be given (where applicable). Where available the pre- and post-site calibration values have been added to Table 3.

**10/2: The statement "... with the mean absolute precision increasing (i.e. becoming less precise)" seems wrong to me.**

We can understand why this sounds odd to the reviewer and have clarified the text. Page 16, Lines 23 – 24, "Both sites showed a small trend with the mean absolute precision worsening with increasing CO2 and CH4 mole fraction."

10/22: the numbers for pressure and temperature are switched – correct. Page 17, Line 17, Corrected.

12/15: 2.3 Results and discussion; 3.3 has the same name, which is confusing. A paper should have one Results/Discussion section.

The paper has been restructured to remove the second Results/Discussion section.

**13/19: What does "...green space..." refer to?**

Green space refers to protected areas of undeveloped landscape this can include areas of native vegetation along with parks and reserves, however, for clarity the term has been replaced with "native vegetation" (Page 19, Lines 9-10).

14/8-19: I can only second the comments by Anonymous Referee #1 on this subject. As detailed in our reply to reviewer #1 the FFT based curve fitting method has been removed from the paper and diurnal and seasonal trends are now examined using box plots.

**14/23: I am reserved about calling a 3.5 years long data set as sufficient for calculating longterm trend curves.**

Section removed

**15/2: put spaces between number and unit (e.g., 108 m). Corrected throughout the text.**

**17/25: you can safely drop "extremely low" without losing any information. What were the values for CO?**

Removed "extremely low". Carbon monoxide was <12 nmol mol-1, this information has been added to the text (Page 7, Line 11).

**18:21: this would be an excellent place to state the maximum humidity of air in the cylinders.**

This section has been removed and instead the typical humidity of the dried cylinders has been included in an earlier part of the paper.

Page 7, Line 21, "...air from a cylinder of dry (<0.002 % H2O) natural air..."

**19/3: typo in the flow range**

Corrected Page 11, Line 6, "...had a flow range of 0.1–0.5 L min-1..."

19/14: analogously to I.D.; 1 " O.D. Corrected Page 10, Line 25, "...coil of ¼ " diameter (I.D. 0.12 ")..."

19/17: please specify "polymeric plastic tubing" better and explain why it was necessary to include it (particularly as this introduces further possible "active" surfaces that can influence the performance of the system).

We understand that this does introduce possible active surfaces, however, due to the fittings of the DPG this was unavoidable. The text has been updated to address this and to specify the exact tubing type.

Page 11, Lines 9-12, "Other than the cryogenic water trap and two short sections (< 10 cm) of ¼" (O.D.) "Bev-a-line" plastic tubing immediately prior to and post the DPG, 1/16" stainless steel tubing was used throughout the system. Due to the air output and input connections of the DPG the use of the plastic tubing was unavoidable."

19/27: typo "2-5" Section removed.

20/7: please be more explicit than "were monitored closely and regularly logged." The text has been updated to Page 12, lines 9-11, "Flow rates, cylinder pressure, chamber temperature and H2O trap temperature were manually logged after each valve position

change and when the water trap was inserted into the silicone oil bath."

20/27: a flowmeter rather "monitors' than "ensures" Section removed.

21/2: it would be helpful for the reader, if you added how this value translates to % H2O This equates to approximately 2.9% H2O. This value has been added to the text Page 11, Line 19.

**21/7: please specify "stabilise".**

Page 11, Lines 18- 19, "A stable water content was defined as one where the standard deviation of the minute mean values was <  $0.003 \ \% H_2O$  for a 15-min period."

21/16: I am not sure what you mean here – if we are talking about H2O < 0.0001 %, we go far below a dew point of 0 deg.C – please explain. Sentence removed.

**21/19: please clarify the frequency 1.5 Hz/data recording strategy here (as the Picarros record with 0.4 Hz).**

Picarro's operate by taking an individual measurement of each gas of interest, in the case of a G2401 instrument CO2, CH4, H2O and CO, in series. For our instruments, these measurements are typically 0.7 sec (approx. 1.5 Hz) apart. Instead one, perhaps more correctly, could consider the time it takes to complete a full cycle through all the gases of interest. For our instruments, this typically takes 2.5 sec (approx. 0.4 Hz). We have updated the text to reflect this.

**22/7: you should number the equations**

Equations numbered

**24/21: use "target gas"**

The authors would prefer not to use the phrase "target gas" as this is a specific term reserved for a long term surveillance tank which is completely independent of the calibration and drift correction of the instrument (WMO, 2016). The tank referred to here is used to drift correct the measurements and cannot be classed as a target tank. Instead, as these cylinders are measured periodically over multiple years the raw uncorrected measurements can be used to estimate typical instrumental drift. This has been clarified in the text.

Page 13, Line 26 - "For the UoB CRDS, raw long-term measurements of standard cylinders have shown the typical instrumental drift to be <  $0.001 \ \mu mol \ mol^{-1} \ day^{-1} \ CO_2$  and <  $0.03 \ nmol \ mol^{-1} \ day^{-1} \ CH_4$ ."

25/14: wrong use of c.f. (cf) – use vs. Corrected Page 22, Line 18.

27/6: "WMO internal reproducibility"; also, decide what you want to use: internal reproducibility guidelines/requirements/bounds and use throughout the text consistently. Please also explain somewhere (best at first use) why you (correctly) aim for Internal reproducibility and not for the Recommended compatibility.

Corrected throughout to the text to "WMO internal reproducibility guidelines". Text added to clarify the choice of internal reproducibility guidelines.

Page 4, Lines 17 – 22, "The importance of these errors are assessed in comparison to the WMO internal reproducibility guidelines (WMO, 2016) which incorporate not only the instrumental precision but uncertainties related to other components sample collection and measurement including drying. These internal reproducibility guidelines are typically half the WMO recommended compatibility goals which, unlike the reproducibility guidelines, are driven by the need for compatibility between datasets."

30/8: can you please explain in a bit more detail how you did this – and how you were able to estimate Nafion-related errors, since, to my knowledge, Reum et al. (2017) did not use one?

Section removed

**31/23: would you be able to discuss the role/importance of the trace gas species gradients between sample and counter flow gas? (see 4/4)**

Added to the text.

Page 26, Lines 6 – 11, "Nafion® membranes, when combined with a dry counter purge gas stream, can be used to effectively dry air samples. This drying process is driven by the moisture gradient between the "wet" sample and the dry counter purge. In a similar manner, as long as the membrane is permeable to the gas, a sample to counter purge gradient in any other trace gas species will also drive exchange. In an effort to quantify the magnitude of  $CO_2$  and  $CH_4$  exchange a series of experiments measuring the composition of the Nafion® counter purge gas were conducted."

**32/3: see 21/16**

Clarified

Page 27, Lines 11 - 12 "These cylinders are very dry,  $H_2O < 0.0001$  %, equivalent to the driest conditions studied in the DPG experiments"

**37: North and scale are missing.**

North and scale have been added. Figure 1.

**38: all abbreviations used should be explained in the figure's caption**

Caption now reads "Figure 2: A generalised schematic showing the initial Bilsdale and Heathfield site setup of the cavity ringdown spectrometer (CRDS) and the Gas Chromatograph – Electron Capture Dectector (GC-ECD) including the gas generator (TOC) and back pressure regulator (BP). Note that Bilsdale has three inlets, while Heathfield has only two as shown here. The Nafion® drying system located downstream of the CRDS multiport valve was removed at both sites in 2015. Black arrows and lines show the direction of sample, standard and calibration gas flow. Grey dashed lines and arrows show the flow path of the Nafion® counter purge gas."

**39: I presume "Hour of day" means local time? Better replace with "Hour (LT)". Was summer / winter time used as well?**

Data were not adjusted for summer/winter time daylight savings. All data were recorded and plotted on UTC as such, "Hour of day" has been replaced with "Hour (UTC)". Figure 6.

**41: see 14/8-19 comment regarding spikes**

This figure has been replaced with two using box plots (Figures 6 and 7) rather than a FFT based curve fitting method to examine diurnal and seasonal trends. This has removed the spikes.

**Figure S3: equations not legible**

Equation text font size has been increased on Figures S5 and S6.

10

Correspondence to: Ann R. Stavert (ann.stavert@csiro.au)

Abstract. Under the UK focused Greenhouse gAs and Uk and Global Emissions (GAUGE) project, two new tall tower greenhouse gas (GHG) observation sites were established in the 2013/2014 Northern

- 15 Hemispheric winter. These sites located at existing telecommunications towers utilised a combination of cavity ring-down spectroscopy (CRDS) and gas chromatography (GC) to measure key GHGs (CO2, CH4, CO, N2O and SF6). Measurements were made at multiple intake heights on each tower. CO2 and CH4 dry mole fractions were calculated from either CRDS measurements of wet air which were post corrected with an instrument specific empirical correction or samples dried to between 0.05 and 0.3 %
- 20 H2O using a Nafion® dryer, with a smaller correction applied for the residual H2O. The impact of these two drying strategies was examined. Drying with a Nafion® drier was not found to have a significant effect on the observed CH4 mole fraction; however, Nafion® drying did cause a 0.02 µmol mol-1 CO2 bias. This bias was stable for sample CO2 mole fractions between 373 and 514 µmol mol-1 and for sample H2O up to 3.5 %. As the calibration and standard gases are treated in the same manner, the 0.02 µmol
- 25 mol-1 CO2 bias is mostly calibrated out with the residual error ( $\ll 0.01 \mu mol mol-1 CO2$ ) well below the World Meteorological Organization's (WMO) reproducibility requirements. Of more concern was the error associated with the empirical instrument specific water correction algorithms. These corrections are relatively stable and reproducible for samples with H2O between 0.2 and 2.5 %, CO2 between 345 and

| 6  | Put u u                                                            |
|----|--------------------------------------------------------------------|
| U  | Deleted: were                                                      |
| •( | Deleted: two                                                       |
| Ó  | Deleted: ,                                                         |
| í  | Deleted: Heathfield (HFD) and Bilsdale (BSD),                      |
| -6 | Deleted. The inclusion of the two additional tower stations within |

**Deleted:** The mclusion of the two additional tower stations within the existing UK Deriving Emissions linked to Climate Change (DECC) network of four stations was found to reduce the uncertainty of  $CH_4$  UK emission estimates by between 10-20 %.

449  $\mu$ mol mol-1 and CH4 between 1743 and 2145 nmol mol-1. However, the residual errors in these corrections increase to > 0.05  $\mu$ mol mol-1 for CO2 and > 1 nmol mol-1 for CH4 (greater than the WMO internal reproducibility guidelines) at higher humidities and for samples with very low (< 0.5 %) water content. These errors also scale with the absolute magnitude of the CO2 and CH4 mole fraction. As such,

water corrections calculated in this manner are not suitable for samples with low (< 0.5 %) or high (> 2.5 %) water contents and either alternative correction methods should be used or partial drying or humidification considered prior to sample analysis.

**Deleted: were**

eletea: found

**1 Introduction**

5

- 10 The adverse effects of anthropogenically driven increases of greenhouse gas concentrations on global temperatures and climate have been well established (IPCC, 2013). Governmental efforts to curb these emissions include the UK 2008 Climate Change Act, which requires the UK to decrease its GHG emissions by 80 % of 1990 levels by 2050 (Parliment of the United Kingdom, 2008 Chapter 27). This in turn motivated the Greenhouse gAs Uk and Global Emissons (GAUGE) project, which aimed to better
- 15 quantify the UK carbon dioxide (CO2), methane (CH4) and nitrous oxide (N2O) emissions. These new emission estimates would then be used to assess the impact of emission abatement and reduction strategies. Key to the GAUGE project was combining new and existing GHG data streams, including high-density regional observation studies, tall tower sites, moving platforms (ferry and aircraft) and satellite observations, with innovative modelling approaches.
- 20 This paper describes the establishment of two new UK GHG tall tower (TT) sites funded under the GAUGE project. Here we provide an analysis of the observations made at the sites and investigate the error associated with empirical instrument specific water correction algorithms and the Nafion®-based sample drying approach used at these TT sites. A further paper, currently in preparation, will discuss the integration of these new sites with the existing UK Deriving Emissions linked to Climate Change (DECC)
- 25 network (Stanley et al., 2018) funded by the UK Department of Business, Energy and Industrial Strategy (BEIS) and provide a full uncertainty analysis for data collected at all the DECC/GAUGE sites. A second companion paperalso in preparation, will discuss the integration and inter-calibration of all the CO2,

| D             | eleted: is divided into two sections. Firstly, it                                                                                                        |
|---------------|----------------------------------------------------------------------------------------------------------------------------------------------------------|
| D             | eleted: ,                                                                                                                                                |
| D             | eleted: es                                                                                                                                               |
| - (D          | eleted:                                                                                                                                                  |
| D
oł
tr | eleted: sites and outlines additional insights that these new
bservations provide when quantifying UK GHG emissions and
ends. Secondly, this paper |
| D             | eleted: s                                                                                                                                                |
|               | eleted: (Stanley, Stavert et al.                                                                                                                         |
| D             | eleted: )                                                                                                                                                |
| - (D          | eleted: While, a                                                                                                                                         |
| D             | eleted: (Stavert, Stanley et al.,                                                                                                                        |
| . ( n         | eleted: )                                                                                                                                                |

[revised manuscript text omitted]
 CO2 < 0.3 µmol mol-1, CH4 < 2 nmol mol-1, CO < 12 nmol mol-1 and H2O < 0.01 % (Figure \$1). All these zero air streams have CO2 and CH4 mole fractions far lower than the 2015 mean global concentrations, 400.99 µmol mol-1 CO2 and 1840 nmol mol-1 CH4 (Dlugokencky and Tans, 2015;Dlugokencky, 2015). While the CO mole fraction is significantly lower than the minimum

15 CO mole fractions typically observed at the HFD and BSD sites,  $\sim 60 \text{ nmol mol}^{-1}$ As such there is a clear and sizable partial pressure difference across the Nafion® membrane for all three species.

**2.3.4 Calculating instrument specific water corrections,**

Motivated by the possibility of CO2 transport across the Nafion® membrane, the decision was made to measure wet samples and correct using an instrument specific water correction. These corrections were

- 20 determined in the field by conducting a droplet test, similar to those described in Rella et al. (2013). In this test, air from a cylinder of dry (<  $0.002 \ \% H_2O$ ) natural air was humidified and the change in CO2 and CH4 mole fraction with water content examined. In brief, a 1.5 m length of 3/8 " Synflex Dekabon metal/plastic composite tubing (EATON, USA) was introduced between the standard cylinder outlet and the CRDS intake. Distilled water (0.7 mL) was injected through a septum located on a T-piece fixed on
- 25 the "cylinder end" of the Dekabon tubing (See Figure \$2 for flow diagram). This water evaporated into the sample stream, with the H2O mole fraction typically peaking at up to 4.5 % (dependent on room

| Del | eted |  | meti | hoo |
|-----|------|--|------|-----|
|-----|------|--|------|-----|

| Formatted: Not Superscript/ Subscript     |  |
|-------------------------------------------|--|
| Deleted: , creating                       |  |
| Deleted: both                             |  |
|                                           |  |
| Deleted: at site                          |  |
| Formatted: Heading 3                      |  |
| Deleted: perceived weaknesses in          |  |
| Deleted: use                              |  |
| Deleted: driers in this application       |  |
| Deleted: used in Yver Kwok et al. (2015). |  |
| Formatted: Subscript                      |  |

temperature) before decreasing to pre-injection concentrations. The effect of this changing  $H_2O$  concentration on the raw (without the inbuilt  $H_2O$  correction)  $CO_2$  and  $CH_4$  concentrations was then observed. The experiment was repeated in at least triplicate annually.

Data collected in the first five minutes immediately following the injection, the typical line equilibration

- 5 period, were excluded from the fit. This avoids using data adversely effected by the effect of rapid changes in H2O content on the cell pressure sensor, as identified by Reum et al. (2018) and the erroneous postinjection CO2 enhancement identified by (Rella et al., 2013). Again, due to cell pressure sensor concerns, data points with minute-mean H2O standard deviations > 0.5 % H2O were excluded. This 5-minute cutoff reduced the maximum H2O value included in the fit to 4 % H2O.

[revised manuscript text omitted]
 CPin values were linearly interpolated and the *CPout* - *CPin* difference calculated as the difference between the CPout and time adjusted CPin values
- 10 and the uncertainty estimated as the combined uncertainty of the  $CP_{in}$  and  $CP_{out}$  values.

**Key experimental assumptions**

These experiments assume that any changes in the  $CO_2$  or  $CH_4$  mole fraction are driven solely by the Nafion® drying processes. Other possible sources of error or bias included, adsorption and desorption effects within the regulator and walls of the tubing, gas solubility within the condenser of the dew point

15 generator and instrumental drift.

Regulator and tubing adsorption and desorption effects has been previously examined by Zellweger and Steinbacher (2017, personal communication). They found that for Parker Veriflo type regulators, as used in this experiment, the effects can be quite large, up to 0.5  $\mu$ mol mol-1 CO2 or 2 nmol mol-1 CH4. But that these effects were only evident at flow rates < 250 ml min-1 and after significant periods of stagnation (15

20 hours). Considering the high flow rates (> 1 L min-1) and long flushing times (2 to 3 hours) used in our experiment it is highly unlikely that regulator effects would make a significant impact on the results. As discussed earlier, a lengthy equilibration period was used at the start of each DPG run and following any change in DPG set point. This was to account for the dissolution of sample gas, in particular CO2, in

the DPG water chamber. After this initial equilibrium period there were no rapid changes in the CO2 mole 25 fraction with only a slow drift, apparent in the data. CRDS instrumental drift is also typically very small

and slow. For the UoB CRDS instrument, long-term measurements of target style standard cylinders have shown the drift to be  $< 0.001 \ \mu mol \ mol^{-1} \ day^{-1} \ CO_2$  and  $< 0.03 \ nmol \ mol^{-1} \ day^{-1} \ CH_4$ . These drift rates are at least an order of magnitude smaller than the mole fraction differences observed in this study.

13

Although small, any time dependent drifts were accounted for by temporally interpolating between each block of data. Also key to the design of this experiment is the examination of differences between two very similar mole fractions rather than absolute mole fractions. As such, any systematic errors that might drive a systematic offset cancel out and any mole fraction depended biases are minimised.

**5 2.3.8 Calibration and traceability**

Calibration procedures for both the CRDS and GC-ECD are as described in detail in Stanley et al. (2018). In brief, CRDS measurements are calibrated using a close-to-ambient standard ("working tank") and a set of three calibration cylinders, which span the typical ambient range (Table 3). Only a small number of elevated observations, < 0.4 % of the CO2 and < 1.5 % of the CH4 minute mean observations, were

10 outside the range of the calibration cylinders. However, a much higher proportion of the CO observations were outside the range of the calibration suites used at site, 28 % at BSD and 43 % at HFD, with the majority of these data points (> 98 %) below the lowest calibration cylinder.

Assigning mole fractions to values outside the range of the calibration suite will increase the error. The magnitude of this error will depend on the magnitude of the mole fraction difference between the closest

- 15 calibration cylinder and the sample. This error has been estimated using measurements made at the Heathfield site of cylinders of known CO mole fractions, 6 and 57 nmol mol-1 CO below the lowest calibration cylinder. These show a percentage error of 2.41 and 3.09 %, respectively. A similar assessment of the error associated with samples above the highest calibration standard were made using cylinders 87 and 686 nmol mol-1 CO above the highest calibration standard. These correspond to percentage errors of
- 20 2.98 and 2.56 %, respectively. As all the minute-mean CO measurements below of the calibration range are within 57 nmol mol-1 of the lowest calibration cylinder and the vast majority of minute-mean CO measurements above the calibration range are within 686 nmol mol-1 of the highest calibration cylinder (99 %) we expect that this error would typically be < 3 %.

Daily measurements of the ambient standard are used to account for any linear drift, while monthly 25 measurements of the calibration suite are used to characterise the nonlinear instrumental response. This calibration procedure is controlled by the GCWerks software and allows near real-time examination of calibrated data, Deleted: against both

**Deleted:**

| Deleted:
estimates. | and should be reflected in increased uncertainty |
|------------------------|--------------------------------------------------|
| Formatte               | d: Superscript                                   |
| Formatte               | d: Superscript                                   |
| Formatte               | d: Not Highlight                                 |
| Formatte               | d: Not Highlight                          |
| Formatte               | d: Not Highlight                          |
| Formatte               | d: Not Highlight                                 |
| Formatte               | d: Superscript                                   |
| Formatte               | d: Not Highlight                                 |
| Formatte               | d: Superscript                                   |
| Formatte               | d: Not Highlight                                 |
| Formatte               | d: Highlight                                     |
All CRDS standards and calibration gases are composed of natural air, some spiked or diluted with scrubbed natural air (TOC gas generator, Model No. 78-40-220, Parker Balston, USA) to achieve the required concentrations of CO2, CH4 and CO. All standard cylinders were filled at Mace Head with well- mixed Northern Hemisphere air. The cylinder spiking and filling techniques of the calibration cylinders

- 5 varied. The Heathfield calibration suite and the second Bilsdale calibration suite were filled at GasLab MPI-BGC Jena and consisted of natural air spiked using a combination of pure CO2 and a commercial mixture of 2.5 % CHd and 0.5 % CO in synthetic air. The "high" calibration cylinder of the first calibration suite used at the Bilsdale site was filled with peak-hour ambient air at EMPA, Dübendorf, Switzerland while the "low" and "mid" cylinders were based on Mace Head air, in the case of the "low" this was
- 10 diluted with scrubbed natural air. Using natural air based calibration and standard gases removes any j pressure broadening effects inherent in the use of non-matrix matched artificial standards (Nara et al., 2012). As the CRDS is an isotopologue-specific method filling the cylinders in such a manner ensures j that the isotopic composition was as close to those of the sampled air as possible. The effect of an isotopic j mismatch between the calibration standards and the sample has been examined in detail by Flores et al. /
- 15 (2017) Griffith (2018) and Tans et al. (2017) With Griffith (2018) showing that, for a sample of 400  $\mu$ mol mol-1 CO2 and 2000 nmol mol-1 CH4, the error will range between 0.001 0.155  $\mu$ mol mol-1 CO2 and 0.1 0.7 nmol mol-1 CH4 depending on the magnitude of the sample to standard mismatch. As such, we expect a worst-case scenario estimate of the error associated with our measurements < 0.04 % for both CO2 and CH4.
- 20 GC-ECD measurements are made relative to a natural air standard of known N2O and SF6 concentration. This standard is measured hourly and used to linearly correct the samples (Table 4). The instrumental nonlinearity response was characterised prior to deployment by dynamically diluting a high concentration standard with zero air and was repeated in the field at the BSD site on 30th September 2015. This approach, dynamic dilution, has a history of use in similar field locations (Hammer et al., 2008) and is able to
- 25 generate multiple calibration points using just two cylinders. This greatly reduces the number of cylinders needed, a key concern for space-limited locations like BSD and HFD. An assessment of the uncertainty associated with this non-linearity approach will be included in a future paper currently in preparation. However, previous studies (Hall et al., 2011;van der Laan et al., 2009;Hammer et al., 2008) have found

**Deleted: re all**

| Formatted: Subscript                                                                             |
|--------------------------------------------------------------------------------------------------|
| Formatted: Subscript                                                                             |
| Deleted: This                                                                                    |
| Deleted: isotopologue                                                                            |
| Deleted: the calibration and                                                                     |
| Deleted: standard                                                                                |
| Deleted: were filled at Mace Head with well-mixed Northern Hemisphere air in an effort to |
| Formatted: Not Highlight                                                                         |
| Deleted: e ratios of the standard and calibration gases were                                     |
| Formatted: Not Highlight                                                                         |
| Deleted: (seeChen et al. (2010);Lee et al. (2006);Nara et al. (2012)                      |
| Formatted: Not Highlight                                                                         |
| Formatted: Not Superscript/ Subscript                                                            |
| Formatted: Subscript                                                                             |
| Formatted: Not Superscript/ Subscript                                                            |
| Formatted: Subscript                                                                             |
| Formatted: Not Superscript/ Subscript                                                            |
| Formatted: Subscript                                                                             |
| Formatted: Not Highlight                                                                         |
| Formatted: Not Highlight                                                                         |
| Formatted: Not Highlight                                                                         |
| Formatted: Not Superscript/ Subscript                                                            |
| Formatted: Subscript                                                                             |
| Formatted: Subscript                                                                             |
| Deleted: for further discussion).                                                                |
| Formatted: Highlight                                                                             |
| Deleted: 3                                                                                       |
| Formatted: Font color: Auto                                                                      |
|                                                                                                  |

the ECD detector response to be extremely stable over time and very linear for both  $SF_{\underline{6}}$  and  $N_{\underline{2}}O$  in the mole fraction range typical of the HFD and BSD stations. As such, we expect the uncertainty of the nonlinearity correction to be very small.

GC-ECD and CRDS standards and calibration cylinders were, where possible, calibrated both before and

- 5 after deployment at the sites. If these two measurements agreed then a mean mole fraction was used, otherwise a linearly drift corrected mole fraction was used. The CRDS cylinders were calibrated through WMO linked calibration centres (either WCC-EMPA, Dübendorf, Switzerland or GasLab MPI-BGC, MPI, Jena, Germany). This ties the ambient measurements to the WMO CO2 x2007 (Zhao and Tans, 2006), CH4 x2004A (Dlugokencky et al., 2005) and CO x2014A (Novelli et al., 1991) scales. The
- 10 calibration of the GC-ECD standards was conducted at either the AGAGE Mace Head laboratory or the University of Bristol laboratory and are reported here on the recently released SIO-16 N2O scale and the SIO SF6 scale. Most cylinders were or will be calibrated before and after deployment and the mean of the two values used. Some cylinders, due to logistical constrains were only calibrated once (Table 4).

**2.3.9 Instrument short-term precision and long-term repeatability**

- 15 The short-term (1-minute) precision of the CRDS data was determined as the mean of the standard deviations of the 1-minute mean data. This was calculated from measurements of the standard cylinder and the calibration suite allowing the relationship between CO2, CH4 and CO mole fraction and short-term precision to be examined. This analysis included 18 cylinders covering a wide range of mole fractions (Table 3).
- 20 The mean absolute short-term precision for all cylinders was consistent between the two sites across all three gases. At BSD the short-term precision was 0.024 µmol mol-1 CO2, 0.18 nmol mol-1 CH4 and 4.2 nmol mol-1 CO while at HFD it was 0.021 µmol mol-1 CO2, 0.22 nmol mol-1 CH4 and 6 nmol mol-1 CO2 Both sites showed a small trend with the mean absolute precision worsening with increasing CO2 and CH4 mole fraction. However, this was not observed in the relative precision which remained unchanged
- 25 at ~ 0.005 % for CO2 and ~ 0.01 % for CH4. This was not the case for CO where the relative precision improved with increasing mole fraction from ~ 4 % at CO < 100 nmol mol-1 to < 1.5 % at CO > 250 nmol mol-1. We suspect that this tendency is inherent in the spectroscopic approach as the CO peak measured

| J | Formatted: | Subscript |
|---|------------|-----------|
| Î | Formatted: | Subscript |

| X |
|---|
|---|

**Moved (insertion) [7] Deleted: 2 Deleted: Long-term repeatability of 20 minute means were calculated by examining the standard deviation of the daily standard cylinder measurements. Moved up [7]: This analysis included 18 cylinders covering a wide range of mole fractions (Table 2). Deleted: ± 0.004 Deleted: ± 0.04 Deleted: ± 0.04 Deleted: ± 0.04 Deleted: ± 0.04 Deleted: ± 1 Deleted: ±1 Deleted: ±1 Deleted: (x ± 1σ) Deleted: increasing (i.e. becoming less precise)**

by the Picarro CRDS is much smaller than those of the CO2 and CH4 (Chen et al., 2013) and hence more susceptible to noise in the baseline particularly at low mole fractions.

The long-term reproducibility of a 20-minute mean was estimated as the mean standard deviation of the daily 20-minute measurements of 4 standard cylinders at each site. Like short-term precision, mean longterm repeatability (calculated over a period of approximately a year) is consistent between the two sites, 5 0.018 and 0.013 µmol mol-1 CO2, 0.20 and 0.20 µmol mol-1 CH4 and 1.1 and 1.7 µmol mol-1 CO at BSD and HFD respectively.

Repeatability of individual injections on the GC instruments were calculated as the standard deviation of the hourly standard injection. These were found to be < 0.3 nmol mol-1 and < 0.05 pmol mol-1, for N2O and SF6 respectively, and did not differ between the two sites.

10

**2.4 Data analysis**

**2.4.1 Data quality control**

A three-stage data flagging and quality control system was used for the HFD and BSD data. Initially, automated flags based on the stability of key parameters including cell temperature & pressure and

- 15 instrument cycle time (the time taken to collect and process each measurement) were applied. Here, data with a cycle time > 8 seconds were filtered out along with any data with cell temperature outside the range  $45 \pm 0.02$  °C or cell pressure outside  $140 \pm 0.1$  Torr. Secondly, a daily manual examination of the GC chromatograms and key GC/CRDS parameter values of each site were made. Data points were flagged if instrument parameters varied beyond thresholds determined to reduce their accuracy and a
- 20 reason for the removal was logged. Finally, all sites were reviewed simultaneously and the mixing ratio of the same gas from each site are overlaid to look for differences between sites. Any significant differences between the background values at each site were investigated by examining key instrumental parameters, calibration pathways and 4-hourly air mass history maps to ensure that these differences represent true signals rather than instrumental or calibration driven artefacts. The hourly air
- 25 mass history maps were produced using the Numerical Atmospheric dispersion Modelling Environment (NAME) Lagrangian dispersion model (Manning et al., 2011).

| Deleted: ± 0.009                             |  |
|----------------------------------------------|--|
| Deleted: ± 0.001                             |  |
| Deleted: ± 0.04                       |  |
| Deleted: ± 0.01                              |  |
| Deleted: ± 0.3                               |  |
| Deleted: ± 0.3                               |  |
| Deleted: $(\overline{x} \pm 1\sigma)$ |  |

**Deleted: - quality controlstatistical processing**

| Deleted: pressure     |  |
|-----------------------|--|
| Deleted: Torr         |  |
| Deleted: temperatures |  |
| Deleted: °C.          |  |

**2.4.2 Statistical processing, baseline fitting and seasonal cycles**

[revised manuscript text omitted]

| Deleted: A known weakness of the statistical fitting procedure is that, due to the FFT used, any gaps in the record must be filled using interpolation before analysis. This can lead to spuriously high (or low) values near gaps within the fitted seasonal cycle if the ends of the gaps occur during a significantly elevated (or reduced) event. These "end effects" are very evident in the HFD 50 m intake record for CH4 (and CO and SF6) in late 2017 leading to the large spikes in the seasonal cycle evident in Figure 5c, e and 1.1 Of the 5 gases measured at HFD and BSD, CO is the only gas to show a decrease in the long-termn trend of the statistically defined baselineole fraction between 2013 and 2017, roughly-6 mmol mol 11 yr -1 (Figure 5f) |
|------------------------------------------------------------------------------------------------------------------------------------------------------------------------------------------------------------------------------------------------------------------------------------------------------------------------------------------------------------------------------------------------------------------------------------------------------------------------------------------------------------------------------------------------------------------------------------------------------------------------------------------------------------------------------------------------------------------------------------------------------------------------------------------------------|
| Deleted: approximately 3µmol mol -1 yr -1 and 7 [5]                                                                                                                                                                                                                                                                                                                                                                                                                                                                                                                                                                                                                                                                                                                            |
| Formatted: Not Highlight                                                                                                                                                                                                                                                                                                                                                                                                                                                                                                                                                                                                                                                                                                                                                                             |
| Formatted: Not Highlight                                                                                                                                                                                                                                                                                                                                                                                                                                                                                                                                                                                                                                                                                                                                                                             |
| Deleted:                                                                                                                                                                                                                                                                                                                                                                                                                                                                                                                                                                                                                                                                                                                                                                                             |
| Formatted [6]                                                                                                                                                                                                                                                                                                                                                                                                                                                                                                                                                                                                                                                                                                                                                                                        |
| Deleted: (Figure 5b and d)                                                                                                                                                                                                                                                                                                                                                                                                                                                                                                                                                                                                                                                                                                                                                                           |
| Formatted[7]                                                                                                                                                                                                                                                                                                                                                                                                                                                                                                                                                                                                                                                                                                                                                                                         |
| Deleted: . The HFD long-term trend curves in CO2, CO and CH4                                                                                                                                                                                                                                                                                                                                                                                                                                                                                                                                                                                                                                                                                                                                         |

**Deleted:** The HFD long-term trend curves in CO2, CO and CH4 are generally positively offster lealive to BSD - most likely driven by the prevalence of polluted air at the HFD site. The ... hile the range of minute mean CO mole fractions was significantly larger at a...St BS..., D averaged 269 mm0 mol3 but varied between ... 3 and ... [5]

**Formatted: Not Superscript/ Subscript**

**Deleted: 2.3.2**

**Formatted: Heading 2**

**Deleted:** Mean ...2O mole fractions observed from the two intakes of comparable height, 108 m at BSD and 100 m at HFD, were found to be identical (329.5 nmol mol-1) and the range of N2O observations were also ...ery similar, 326.6 to 340.0 and 326.4 to 338.5 nmol mol-1 for BSD and HFD, respectively (Figures 3g ...d and 4g...d). The N2O data from the higher (248 m) intake at BSD, has has not only a higher mean value, 331.1 nmol mol-1 but a ...arrower range, 329.3 to 335.0 nmol mol-1 ... [10]

**(Formatted: Not Highlight**

**Deleted:** The 248m data analysed in this paper was from the 17th March 2017 (compared with late 2013 for the other heights) till the 19th June 2017. Hence this data is likely to be higher in N2O relative1 **Deleted:** 5g

**Eormatted**

| V  | ronnatteu                                                          | [12] |  |  |
|----|--------------------------------------------------------------------|------|--|--|
| Ì  | Deleted:are similar at both sites. and the long-term trend,        |      |  |  |
| ļί | ~0.8 (Figure 5h)                                                   | [13] |  |  |
| l  | Formatted                                                          | [14] |  |  |
| Ì  | Deleted: are lso agrees well between the two sites and very |      |  |  |
| V  | similar between the two sites                                      | [15] |  |  |
| Ì  | Deleted:                                                           |      |  |  |

spring. This was attributed to the winter intrusion of N2O depleted stratospheric air and its delayed mixing into the lower troposphere. In contrast, a UK focused inversion study Ganesan et al. (2015), found that N2O flux seasonality is driven not just by seasonal changes in temperature but by agricultural fertilizer application and post-rainfall emissions. They predict the largest net N2O fluxes will occur between May

- 5 and August while agricultural fluxes will peak during spring for eastern England and summer time for central England. However, the exact timings of these fluxes can vary year-to-year as they depend not only on the scheduling of agricultural fertilizer application but on rainfall and temperature. Like MHD, BSD and HFD are expected to experience a decrease in N2O driven by stratospheric intrusion, which would account for the springtime maximum and summer minimum. However, both BSD and HFD are located
- 10 much closer to significant agricultural sources of N2O than MHD. Hence, it is likely that they are much more influenced by agricultural N2O fluxes. As such, it is possible that although a summer time maximum in N2O flux is completely offset by stratospheric intrusion, this summer time maximum may be so large that the residual autumn tail of this event appears as a second maximum at BSD and HFD.

Clear diurnal cycles in N2O were observed at the HFD for the spring, summer and autumn months with the maximum N2O mole fraction occurring 2 hours after sunrise and the minimum in the mid-afternoon

(Figure 6h). These cycles were not as apparent at BSD (Figure 6g). The long-term trend in the SF6 mole fraction at BSD and HFD shows a gradual increase of 0.3 pmol mol-1 yr-1 again agreeing well with MHD which showed an identical growth rate. Although the predominant sources of SF6 are electrical switchgear, which is not expected to have significant seasonality, there was

- 20 a small seasonal cycle observed (Figure 7i & j). This cycle is more apparent in the 108 m BSD data and appears as a slight (0.1 to 0.15 pmol mol-1) enhancement in SF6 in the winter months. This seasonal shift occurs across the wider DECC-GAUGE network and air history maps suggest that it is not associated with an obvious UK or continental region. As such, instead of an atmospheric transport driven shift we believe this to be a true change in emissions and hypothesise that this may be due to increased load on,
- 25 and hence increased failure of, the electrical switchgear during the colder months. SF6 mole fractions averaged 8.9 pmol mol-1 at both BSD and HFD. While HFD, located closer to large conurbations than BSD, typically saw higher SF6 pollution events. This was reflected in its larger range of 8.1 to 34.2 pmol mol-1 compared with 8.1 to 22.9 pmol mol-1 at BSD (Figures 4e and 5e).

| Deleted: 3h)   |        |  |  |
|----------------|--------|--|--|
| ()             |        |  |  |
| Deleted: (Figu | re 5j) |  |  |
|                |        |  |  |
| Deleted: 5i    |        |  |  |
|                |        |  |  |
|                |        |  |  |
|                |        |  |  |
| Deleted:       |        |  |  |
| Deleted: We    |        |  |  |
|                |        |  |  |
|                |        |  |  |
|                |        |  |  |

**3.3 Site specific water corrections**

The annually determined instrument specific water corrections are typically very similar at each site, often within the 95 % confidence interval of the triplicate runs (Table 1), suggesting that the corrections are fairly stable between years and instruments. The residuals of the instrument specific water corrections are

- 5 generally quite small, with 25th and 75th quartiles of -0.03 and 0.05,  $\mu$ mol mol-1 CO2, and -0.4 and 0.3 pmol mol-1 CH4 (Table 1). The mean absolute residuals are, on average, smaller than those of the inbuilt correction and are notably smaller at higher H2O content (see Figure S5). For example, the mean absolute residuals for 2015 data from HFD with H2O > 2 % are 0.04 and 0.09  $\mu$ mol mol-1 CO2 and 0.4 and 1.2 nmol mol-1 CH4 for the new and inbuilt correction, respectively.
- 10 While instrument specific CO water corrections were calculated, the large minute-mean variability inherent in the G2401 CO measurements (> 4 nmol mol-1) meant that the difference between data corrected using the instrument specific and in-built correction was not statistically significant. As such, these corrections were not presented in the body of the paper, however, further information can be found in Figure S5 of the supplementary.
- 15 Plots of the residuals typically show a common pattern, with the residual of zero at 0 % H2O, before dipping below zero and then returning to zero at H2O between 0.2 and 0.5 % (Figure \$5). Unlike other tests, the depth and width of this dip is more pronounced for BSD 2017. However, the BSD 2017 data both spans a wider range of H2O contents than the earlier BSD tests (0 to 3.5 % ys. 0 to 2.2 %) and has far fewer data points in the 0.1 to 1 % H2O range (0.9 % of all data points ys. 34 % and 27 % for BSD
- 20 2015 and 2016, respectively). The BSD 2017 0.1 to 1.0 % minute mean data also have an average standard deviation an order of magnitude larger than those of 2015 and 2016 (Figure \$5a, b & c). Refitting the BSD 2017 correction using only data H2O < 2.2 % decreases the depth of the deviation by 0.05 µmol mol-1 CO2 and 0.3 nmol mol-1 CH4 as well as decreasing its width slightly but the deviation remains. This suggests that the presence of the dip is robust but the change in its shape between 2017 and 2016 may
- 25 well be a fitting artefact.

Reum et al. (2018) previously identified this pattern in water correction residuals and linked it to a pressure sensitivity at low water vapour mole fractions. They proposed an alternative fitting function incorporating the "pressure bend" although they do not recommend using this fit for data collected during

| 1                 | Moved a   | own [1]: <#> cross membrane transport for ambient                  |
|-------------------|-----------|---------------------------------------------------------------------------|
| _                 | Deleted:  | <#>Experiment 1 —                                                         |
|                   | Moved d   | own [6]: <#> At least 15 minutes of data were                             |
|                   | Deleted:  | <#>3.2.2                                                                  |
| 1                 | Deleted:  | <#>Quantifying CO2 and CH4 cross membrane[16]                             |
| $\langle \rangle$ | Deleted:  | <pre><#>with varying sample H2O mole fraction [17]</pre>            |
|                   | Deleted:  | <#>of                                                                     |
|                   | Deleted:  | <#>Nation                                                                 |
|                   | Deleted   | A areas membrane transport for embiant CO2 and of                         |
|                   | Deleted.  | <pre>closs memorane transport for amolent CO2_mu[18]</pre>                |
|                   | Maurad d  | ····· [19]                                                                |
|                   | Moved d   | own [z]: <#>                                                              |
|                   | Deleted:  | <#> [20]                                                                  |
|                   | Deleted:  | <#>.                                                                      |
|                   | Moved d   | own [3]: <#> ¶                                                            |
|                   | Moved d   | own [4]: <#> Flow meters F1 and F2 had a range of                         |
|                   | Moved d   | own [5]: <#> to a second identical valve, V2, while                       |
|                   | Deleted:  | <#> [21]                                                                  |
|                   | Deleted:  | <#>.                                                                      |
|                   | Deleted:  | <pre><#> to a second identical valve, V2, while the second []</pre> |
|                   | Deleted:  | <#> Flow meters F1 and F2 had a range of 0.1-1.L[22]                      |
|                   | Deleted:  | <#>One of the outputs of V1 went via the Nafion                           |
|                   | Deleted:  | <#> dry and Nafion wet sample streams.                                    |
|                   | Deleted:  | <#> At least 15 minutes of data were collected after that                 |
|                   | Deleted:  | [24]                                                                      |
|                   | Deleted:  | #Sobserve gas exchange across the Nation membranes                        |
|                   | Deleted:  | (#>) For this experiment, two changes were made that                      |
|                   | Deleted.  | (#>). For this experiment, two enanges were made [27]                     |
|                   | Deleted.  |                                                                           |
|                   | Deleted:  | *#>was placed between v2 and the CRDS intake, [29]                        |
|                   | Deleted:  | <#> connected to the                                                      |
|                   | Deleted:  | <#>multiport valve allowing the sampling                                  |
|                   | Deleted:  | <#> of the Nation                                                         |
|                   | Deleted:  | <#>                                                                       |
|                   | Deleted:  | <#>counter purge                                                          |
|                   | Deleted:  | <#>gas                                                                    |
|                   | Deleted:  | <#> prior to                                                              |
|                   | Deleted:  | <#>(CPin) and after                                                       |
|                   | Deleted:  | <#>the Nafion                                                             |
|                   | Deleted:  | <#> (CPout). A fourth flowmeter (F4) was placed after )                   |
|                   | Deleted:  | <#> to                                                                    |
|                   | Deleted:  | <#>heat                                                                   |
|                   | Deleted:  | <#>the                                                                    |
|                   | Deleted:  | <#> new                                                                   |
|                   | Deleted:  | <#> multiport valve to > 25 °C only dew points                            |
|                   | Deleted:  | <#>of                                                                     |
|                   | Deleted:  | <#>25 °C                                                                  |
|                   | Deleted:  | <#>or less                                                                |
|                   | Deleted:  | <pre><#> were used in this experiment.¶</pre>                       |
|                   | Formatted |                                                                           |
|                   | Deleted:  | ====================================                                      |
|                   | Deleted   | <pre><#> experiment was conducted in three stages Firstbal</pre>    |
|                   | Deleted   | =                                                                         |
|                   | Deleted:  | stating                                                                   |
|                   | Delatad:  |                                                                           |
|                   | Deleted:  | -m-stage                                                                  |
|                   | Deleted:  | we use rize use removed from the Dewar of [34]                            |
|                   | Deleted:  | \# .                                                               |

the droplet test due to the paucity of stable data typically obtained between 0.02 and 0.5 % H2O and the effect of rapidly changing H2O on the cell pressure sensor. Implementing a more controlled water test at the sites would allow the use of the new fitting function. But due to the complexity of such a test this would be logistically difficult at remote field sites.

- 5 It is also important to note that the magnitude of the dip observed by Reum et al. (2018) in their controlled water tests, ~ 0.04 μmol mol-1 CO2 and 1 nmol mol-1 CH4, are roughly half those observed for the HFD, BDS and UoB droplet tests. As such the increased residuals observed for our water corrections between 0.02 and 0.5 % H2O are likely to be primarily driven by the rapidly changing H2O content inherent in the droplet test rather than represent a true error in the water correction.
- 10 The poor performance of the CRDS pressure sensor at low H2O mole fractions, 0.02 to 0.5 % H2O, is not expected to be a large source of error for undried samples as the majority of these, 92 % of the BSD and HFD data, contain > 0.5 % H2O. But this is likely to be a large source of error for Nafion® dried samples where 95 % of HFD and 92 % of BSD are < 0.5 %. Calibration gases, although partially humidified to < 0.015 % H2O, when flowed through a damp Nafion® are still far drier than Nafion® dried air samples
- 15 which averaged 0.2 % H2O. As such this effect will not be accounted for as part of the calibration process. It is difficult to quantify this error, as it will vary with sample water content and the sensitivity of the individual instrument's pressure sensor to low H2O mole fractions. However, for BSD and HFD2 assuming that the residuals of the droplet water tests are an accurate reflection of the likely error (Figure S5), we expect there to be a systematic offset of the order of -0.05 to -0.1 µmol mol-1 CO2 and -1 to -2
- 20 nmol mol-1 CH4.

The sample mole fraction dependence of the CRDS water correction was examined by conducting water droplet tests using dry cylinders of above and below ambient mole fractions (Section 2.3.5). Specific above and below ambient water corrections were calculated based on these data sets (Table1 and Figure S6). Jf the water correction was independent of sample mole fraction then the residuals should be identical

25 for both correction types. Although the above and below ambient residual plots are similar they do differ slightly with the residual of the above mole fraction sample becoming more positive at higher H2O mole fractions while the below ambient mole fraction residuals become more negative. This is reflected in the

| Deleted: Fitting the HFD and BSD data using this function reduced      |
|------------------------------------------------------------------------|
| the dip in the residual but did not significantly alter the mean       |
| residual. However, due to the limitations of the droplet water test as |
| outlined by Reum et al. (2017) we have decided not to implement        |
| this.                                                                  |

[revised manuscript text omitted]

to measurements made at higher water contents.

| Deleted: | 10 |  |
|----------|----|--|

eleted: This is not the case for samples dried with a Nafion drier. re the CRDS is measuring samples with low water contents, for ample 38 % at BSD and 42 % at HFD of the Nafion dried air mples had  $H_2O < 0.1$  %. leted: 10

| _   |      |   |     |
|-----|------|---|-----|
| Dal | -t   | • | 10  |
| De  | eleu |   | IU. |

leted: Alternatively, Deleted: could

**3.4 Quantifying the CRDS water correction error using the dew point generator**

The change in the CRDS water correction with sample  $H_2O$  content was characterised using the difference between the *Wet* and *Dry* DPG runs. This error typically had a shallow negative parabolic trend for both CO2 and CH4 (Figure 2) and was similar to the shape seen in the residual of the CRDS water corrections

5 (Figure \$5 and \$6) with the error negative at H2O mole fractions near 0.5 %, becoming more positive between 1 and 2 % H2O before dropping at higher H2O contents.
 Although the UoB CRDS was not deployed in the field we expect the results of the DPG tests to be typical

of most Picarro G2401 CO2/CH4 CRDS instrumentation. The DPG tests show that for ambient and below ambient mole fraction samples the CH4 error remained within the WMO internal reproducibility 10 guidelines (WMO, 2016) at all water contents examined, that is 0.6 to 3.5 % H2O, while the CO2 error

- increased outside the guidelines for  $H_2O > 2.5$  %.  $CO_2$  errors increased rapidly outside this range reaching 0.3 µmol mol-1 at 3.5 % H2O. These results are broadly consistent with those of the droplet test residuals. Unlike the ambient and below ambient samples, the CRDS water correction error of the above ambient sample, UoB-04, exceeded the WMO internal reproducibility guidelines for both  $CO_2$  and  $CH_4$  at most
- 15 H2O mole fractions. For the H2O range of the BSD and HFD sites the error peaked at 0.1 µmol mol-1 for CO2 near 1.75 % H2O and at 2 nmol mol-1 CH4 near 2.25 % H2O. As discussed earlier in Section 3.3, the absolute error in the CRDS water correction will scale with the absolute mole fraction of the sample due to the structure of the correction. The UoB CRDS correction was also optimised using a cylinder of significantly lower mole fraction (397.38 µmol mol-1 CO2 and 1918.73 nmol mol-1 CH4 compared with
- 20 515.4 μmol mol-1 and 2579.5 nmol mol-1). This shift in error/residual was also observed in the H2O droplet tests using higher mole fraction cylinders although it appears larger for the DPG tests, most likely due to the higher mole fractions used within these tests (515.4 and 2579.5 compared with 449.55 μmol mol-1 CO2 and 2148 nmol mol-1 CH4, respectively).

 The full range H2O mole fractions observed at the HFD and BSD sites, 0.05 to 2.5 % H2O, a were not

 25
 examined in these tests which due to limitations inherent in the experimental set up were restricted to a

 $H_2O$  range of 0.6 - 3.5 %. However, it is possible to conclude that for observations of ambient and below ambient CO2 and CH4 mole fractions with  $H_2O > 0.6$  % the water driven error in the CRDS water correction is not likely to be a major source of uncertainty. Even at other DECC sites that are subject to

| 1 | 1 |
|---|---|
|   | - |
| ~ | • |

| Deleted: $(Wet - Dry = E_{X\%})$ |
|-----------------------------------------|
|                                         |
| Deleted: 11                             |
| Formatted: Not Highlight                |
| Deleted: S3                             |

| (   | Deleted: Considering the typical |
|-----|----------------------------------|
| λ   | Formatted: Subscript             |
| λ   | Formatted: Subscript             |
| 1   | Formatted: Subscript             |
| (   | Formatted: Subscript             |
| ~~( | Deleted: ny                      |
| (   | Deleted: for these sites         |

higher humidity, for example the Angus site (Stanley et al., 2018) periods of high (> 2.5 % H2O) water content are rare, < 0.03 % of the data record. In contrast, as elevated CO2 and CH4 mole fractions are regularly observed at both the HFD and BSD sites, the increase in CRDS error with mole fraction is a source of concern and must be quantified as part of a full uncertainty analysis.

**5 3.5 Quantifying Nafion® cross membrane transport**

Nafion® membranes, when combined with a dry counter purge gas stream, can be used to effectively dry air samples. This drying process is driven by the moisture gradient between the "wet" sample and the dry counter purge. In a similar manner, as long as the membrane is permeable to the gas, a sample to counter purge gradient in any other trace gas species will also drive exchange. In an effort to quantify the

- 10 magnitude of CO2 and CH4 exchange a series of experiments measuring the composition of the Nafion® counter purge gas were conducted. During these experiments all measurement of the Nafion® counter purge ( $CP_{in}$  and  $CP_{out}$ ) were cryogenically dried to < 0.002 % H2O prior to CRDS analysis. Hence the need to use an empirical CRDS water correction and any error associated with the correction was removed and differences between the  $CP_{in}$  and  $CP_{out}$  samples can be solely attributed to transport across the
- 15 Nafion membrane (NX%). The results of these experiments are shown in Figure 10. The counter purge experiments conducted with both the ambient (UoB-15) and above ambient (UoB-16) mole fraction cylinders show identical changes in CO2 and CH4 mole fractions, respectively. The wet sample NX% difference is consistently positive for CO2 with the CPout mole fraction an average of 0.021  $\pm 0.002 \ \mu$ mol mol-1 ( $\bar{x} \pm 95 \%$  conf. int., n > 19) higher than CPin, reflecting a loss from the sample to the
- 20 counter purge across the Nafion® membrane (Figure 10a). Although small, this value is an order of magnitude larger than the average standard deviation of the 15 min block means (0.002 µmol mol-1 CO2) making it well within the typical measurement precision. This difference decreases slightly with decreasing sample water content but it is never zero. Even with a dry sample, the  $CP_{out}$   $CP_{in}$  difference (Nx%),  $0.015 \pm 0.003$  µmol mol-1 CO2, is still positive. This is in line with previous studies, which have

25

| found that, although water substantially increases membrane perme | ability, even dry membranes are   |
|-------------------------------------------------------------------|-----------------------------------|
| permeable to CO2 (Ma and Skou, 2007; Chiou and Paul, 1988). As    | s earlier studies have found that |

| Deleted: the        |  |
|---------------------|--|
| Deleted: hence      |  |
| Deleted: it         |  |
| Deleted: . As such, |  |
| Deleted: solely     |  |
| Deleted: 4          |  |

membranes can take more than a week to fully dry out (Chiou and Paul, 1988), it is also highly likely that the relatively brief length of this study (4 to 5 hours) was too short to remove all H2O from the membrane. The CH4 CPin and CPout mole fraction difference for both dry and wet samples is also slightly positive,  $0.03 \pm 0.01$  and  $0.04 \pm 0.02$  nmol mol-1 CH4, respectively (Figure 10c). This value is very close to the

5 measurement precision, with the average CH4 standard deviation of the 15-min block means of the order of 0.02 nmol mol-1 CH4.

The ~ 0.02  $\mu$ mol mol-1 loss of CO2 across the Nafion® membrane from the sample stream to the counter purge observed here, although small, is of the order of the WMO internal reproducibility guidelines, 0.05  $\mu$ mol mol-1 in the northern hemisphere and 0.025  $\mu$ mol mol-1 in the southern hemisphere (WMO, 2016),

- 10 and must be acknowledged. However, the calibration gases are also passed through the Nafion. These cylinders are very dry,  $H_2O < 0.0001$  %, equivalent to the driest conditions studied in the DPG experiments (Figure 10a and b) and as such would be expected to show similar CO2 loss across the Nafion. membrane, ~ 0.015 µmol mol-1. Hence, as the bias is constant with sample CO2 and H2O mole fractions and as a bias would be present in both the calibration gases (~ 0.015 µmol mol-1) and samples
- 15 (~ 0.02 µmol mol-1) the majority of the bias will be calibrated out, with only a very small (≤ 0.005 µmol mol-1) constant bias, of the order of the instrumental precision, remaining.
   In contrast, the mean CH4 Nafion bias, 0.04 ± 0.02 nmol mol-1, is at least an order of magnitude smaller than the WMO internal reproducibility guidelines (WMO, 2016) and is extremely close to the typical measurement precision suggesting that it is not a bias of concern.

**20 4 Conclusions and future work**

The newly established Bilsdale and Heathfield tall tower measurement stations provide important new data sets of GHG observations. These high-precision continuous in situ measurements show clear long term increases in baseline CO2, CH4, N2O and SF6 mole fraction and capture the seasonal and diurnal cycles of these key gases. It is expected that these observations, when combined with regional inversion

25 modelling, will significantly improve our ability to quantify UK greenhouse gas emissions — both reducing the uncertainty and improving the spatial and temporal resolution. Future work using this data

| Deleted: t                    |
|-------------------------------|
|                               |
|                               |
|                               |
| Deleted: requirements         |
| Palatad: (WMO 2016)           |
| Deleted. (wM0, 2010)          |
| Deleted:                      |
|                               |
| Deleted: the 0 °C data points |
| Deleted: used in              |
| Deleted: Experiment 3         |
|                               |

**Deleted:**

Considering how poorly the results from Experiment 1 and 2 characterise the NxNs value it is interesting to see that the CO2 NxNs value obtained here is within the bounds of those estimates while the CH4 estimates are not. It is not clear whether this is a fortuitous quirk or because the CRDS water correction error is two-fold larger for CH4 than CO2 relative to the typical atmospheric mole fractions

is focusing on better estimates of UK GHG emissions with a particular emphasis on the UK carbon budget.

An examination of the Nafion R drying method found it to have a small inherent CO2 bias of 0.02 µmol mol-1; however, this bias did not vary significantly with sample water content > 0.7 % H2O or CO2 mole

5 fraction. Even samples as dry as the calibration gases were affected by this Nafion® bias, although to a smaller degree — ~ 0.015 µmol mol-1 for H2O < 0.0001 % — as residual moisture remained in the membrane. Thus, as calibration gases are dried in an identical manner to the samples, this bias is mostly calibrated out with only a very small (≤ 0.005 µmol mol-1) constant residual bias of the order of the instrumental precision. As such, the Nafion® drier itself when, used in this manner, does not contribute 10 a significant bias to the resulting CO2 observations.

In contrast, the errors associated with the CRDS water correction for samples with low water contents (< 0.5 %), like those dried using a Nafion® drier, can be significant and difficult to adequately quantify using the current in field techniques. Hence, even though Nafion® driers are not themselves an inherent source of bias, for the CRDS instrumentation examined in this study the incomplete drying of the sample

- 15 is a significant source of error. Altering the drying method to better match the moisture content of the calibration gases to the sample may minimise this error. It is also important to note that this may not be the case for other CRDS instrumentation or optical techniques that use alternative cell pressure sensors. Similarly, samples with high water contents > 2.5 % H2O or CO2 or CH4 mole fractions significantly above ambient mole fractions are also subject to larger water correction errors.
- 20 Estimates of these types of errors for Bilsdale and Heathfield have been given in Table 5 and range between 0.2 and 0.05 μmol mol-1 CO2 and 3 and 1 nmol mol-1 CH4. While these errors are significant relative to the WMO internal reproducibility goals they are typically smaller than the extended WMO measurement compatibility goals (± 0.2 μmol mol-1 CO2 and ± 5 nmol mol-1 CH4). It is also important to note that they are orders of magnitude smaller than baseline excursions observed at the sites (see Figures)
- 25 4 & 5). They are also a factor of 10 smaller than the CHd model-data mismatch within the UK DECC network as estimated by Ganesan et al. (2015) at ~ 20 nmol mol-1.
   While, drift in the instrumental water correction typically small it is important that it is identified and accounted for through regular water tests. The necessary frequency of these water tests will depend on

| Formatted: Subscript |  |
|----------------------|--|
| Formatted: Subscript |  |
| Formatted: Subscript |  |

| Formatted: Superscript |  |
|------------------------|--|
| Formatted: Subscript   |  |
|                        |  |
| Formatted: Subscript   |  |
| Formatted: Superscript |  |

**Deleted:**

**Deleted: Undried**

the stability of the individual instrument and the typical CO2, CH4 and H2O mole fractions at the given location and should be determined on a case by case basis.

Future improvements to the Bilsdale and Heathfield records include the addition of target tanks at the sites. Although the use of target tanks do not directly influence measurement uncertainty they allow

5 independent long-term monitoring of instrument performance and are a useful tool for assessing measurement uncertainty. The development of a full uncertainty analysis incorporating such target tank measurements, along with an assessment of the calibration strategy and any instrumental, water correction and sampling errors is also planned.

**Deleted:** Future improvements to the Bilsdale and Heathfield records include the addition of target tanks at the sites to allow independent long-term monitoring of instrument performance and the development of a full uncertainty analysis incorporating calibration, instrumental, water correction and sampling errors.

**5 Acknowledgements**

- 10 This study was funded under the NERC Greenhouse Gas Emissions and Feedbacks Program as part of the Greenhouse gAs UK and Global Emissions (GAUGE) area grant number NE/K002449/1NERC. This grant also covered the establishment and early running costs of the stations. Operating costs of the Bilsdale site after 17th September 2016 were funded by the UK Department of Business, Energy and Industrial Strategy (formerly the Department of Energy and Climate Change) through contract TRN1028/06/2015.
- 15 The National Physical Laboratory (NPL) took responsibility for the Heathfield site on the 30th September 2017 by the UK's Department for Business, Energy and Industrial Strategy as part of the National Measurement System Programme.

The authors would also like to acknowledge the support of Dr Carole Helfter and Dr Neil Mullinger from the NERC Centre for Ecology and Hydrology (CEH), Edinburgh, Scotland who helped to maintain and

20 run the Bilsdale site. Lastly the authors would like to thank Dr Joseph Pitt from the University of Manchester for the use of the dew point generator used during a series of preliminary studies.

**29**

**Deleted: 5.**

---

## Editor Decision (ED1)

[revised manuscript text omitted]

---

## Author Response (AR2)

**Reply to reviewer #1**

Ann R. Stavert[1], Simon O'Doherty[2], Kieran Stanley[2], Dickon Young[2], Alistair J. Manning[3], Mark F. Lunt[5], Christopher Rennick[4] and Tim Arnold[4,5].

[1] Climate Science Centre, CSIRO Oceans & Atmosphere, Aspendale, VIC, 3195, Australia

5   [2] School of Chemistry, University of Bristol, Bristol, BS8 1TS, United Kingdom

[3] Met Office, Exeter, Devon, EX1 3PB, United Kingdom

[4] National Physical Laboratory, Teddington, Middlesex, TW11 0LW, United Kingdom

[5] School of GeoSciences, University of Edinburgh, Edinburgh, EH9 3FF, UK

10   The authors would like to thank the reviewer for their helpful comments. We have endeavoured to incorporate the suggestions that they have made. The locations of changes to the paper are given as Page Number, Line number of the new paper while the reviewers comments contain the locations in the original paper. Our responses are shown here in blue text and the reviewers' comments in black.

15   **Overall framing:**

There are several other papers already describing CRDS installations at field sites. The novel aspect of this paper is that two different drying methods were deployed -- each for a significant amount of time, and laboratory tests have been performed to quantify their errors. Currently the manuscript does not adequately describe the extent to which systematic errors introduced by the CRDS water correction are mitigated by calibrating through the nafion. This is a crucial question and should be

20   relatively easily addressed with existing data. Specific suggestions for figures to address this question are given below. If it is the case that passing calibration gases through the nafion substantially mitigates the systematic errors, then the initial configuration would seem to be clearly superior in terms of ease of implementation/maintenance and remaining systematic errors. It would be especially useful for the discussion to consider implications of complex and frequent instrument specific annual water correction strategy versus ease of operations for the original configuration.

25   If it is the case that dried air samples versus humidified calibration gases have nearly the same water content, then any bias associated with the picarro water correction should be nearly or entirely calibrated out (see specific comments below). To what extent does the daily measurement of the ambient standard remove bias introduced by the CRDS water correction? I think the systematic errors should be estimated based on the difference between the humidity of humidified drift-tracking standard versus the humidity of the dried ambient sample. It would be useful to provide some details in section 2.3.8 about

the extent to which standards are humidified and how much variability in ambient humidity is observed between daily (drift) calibration episodes.

It would be interesting to include a figure in the paper or in the supplement showing the complete h2o time series prior to removal of the nafion, i.e. showing the CRDS measurements of post-nafion humidity with enough detail to show how it varied over different timescales, including seasonally and diurnally (some variability in drying performance is expected given that the effectiveness of nafion drying is strongly dependent on the temperature of the membrane).

Also consider including a panel(s) showing typical humidity difference between ambient sample air and during a daily drift calibrations (i.e. how much water does the calibration gas pick up and does it dry out perceptibly during the course of a 20 minute calibration interval?).

It is critically important to understand the extent to which calibrating through the nafion once per day mitigates errors associated with the crds water correction. If the amount of water stored in the nafion membrane is sufficient to (1) humidify the standard to the same humidity as the ambient sample is dried and (2) to largely mitigate within-day variability of sample h2o, then errors related to h2o are likely negligible. It seems that the original nafion strategy may have had only very small systematic errors and that these could likely be further reduced if necessary by increasing the frequency of the daily CRDS analyzer drift calibrations through the nafion.

For the second phase, where the nafion dryers were removed, the authors have gone to great trouble to track the instrument specific water corrections, and the findings are very interesting. In the end, I wonder if perhaps the errors associated with using a single pre-deployment water correction are actually tolerable for the application of estimating UK scale fluxes (at least in the case where some drying is used to keep h2o < ~2%). Looking at Figure 8 a and b, and given the large differences among the h2o corrections for humidity > ~2%, it seems that the errors are unacceptably large, and unlikely to be improved even with weekly testing.

It would be useful to discuss the significance of systematic errors with respect to the current model errors and goals of the UK network (i.e. for national-scale evaluation of reported emissions by atmospheric monitoring to be useful). Based on all of tests and analysis, what are the recommendations about how to simplify the operations and reduce the systematic errors going forward? My reading is that the systematic errors would be greatly reduced if nafion dryers were to be reinstalled, so long as the calibration gases are also routed through the nafion dryer and so long as the membrane provides an adequate reservoir of h2o such that the humidity of cal gases and ambient samples is the same. It is worth reemphasizing the need to calibrate through the nafion for any system that has a large partial pressure gradient of co2 and/or ch4 across the membrane. The authors would like to thank the reviewer for their detailed response to the paper. The reviewer outlines four key improvements to the paper and provides some suggests as to how these may be addressed.

1. *A better description of the extent to which systematic errors introduced by the CRDS water correction are mitigated by calibrating through the Nafion®. Specifically, including information on the humidification of the daily standards, their ability to track changes in ambient humidity and how this would reduce systematic errors.*

2. *A more detailed discussion of the advantages and disadvantages of the two drying techniques namely the advantage to the Nafion® based drying system in terms of minimising systematic error and practicality and the use of an annually determined water correction for samples with $H_2O < 2\%$.*

3. *A discussion of the significance of the systematic errors in terms of the goals of the UK tall tower network and a conclusion in terms of the optimum drying process for the network.*

4. *A reemphasis of the importance when using Nafion® drying systems of drying the calibration and standards gases in an identical manner to the sample to account for any loss of $CO_2$ across the membrane.*

We thank the reviewer for the suggestions and discuss them each in turn.

1. *A better description of the extent to which systematic errors introduced by the CRDS water correction are mitigated by calibrating through the Nafion®. Specifically, including information on the humidification of the daily standards, their ability to track changes in ambient humidity and how this would reduce systematic errors.*

As the reviewer notes passing the calibration and the standard gases through the Nafion® does substantially mitigate systematic errors in the CRDS water correction. These systematic errors, as demonstrated in Figure S8, tend to peak at low (0.05 to 0.5% $H_2O$) moisture contents - the typical moisture content of samples dried using a Nafion® based drying system. However, as the moisture content of the standard closely follows that of the sample (see Figure S4) these systematic errors should be substantially mitigated. Following the suggestion of the reviewer Section 2.3.9 (previously Section 2.3.8) now includes a description of the moisture content of the standard, its variability over time and relationship to the moisture content of the ambient air samples. We have also included three new plots, Figure S1, S2 and S3. Additions to the text and references to the new figures are listed below.

Page 7, lines 3 – 7: "When functioning correctly this drying method resulted in air samples with water mole fractions between 0.05 and 0.2 % $H_2O$ depending on the original moisture content of the air and temperature. However, at Bilsdale, due to problems with the TOC and the tubing initially installed at the site, the Nafion® was not drying optimally and significant periods of 2014 had far higher moisture contents (Figure S1)."

Page 7, lines 13 – 15: "Plots of the water content of all air samples along with the comparisons of the diurnal and seasonal cycles in sample moisture content can be found in the supplement (Figures S1 and S2)."

Page 15, line 23 – Page 16, line 13: "During the period that the Nafion® drying system was used these standards were partially humidified as they passed through the wet Nafion® dryer. The level of humidification is dependent on that of the air samples measured prior to the standard. The moisture content of the standard closely tracks that of the air samples with variations in

the humidity of the samples clearly reproduced in the standard (Figure S1). However, the moisture content of the standard is generally slightly lower. On average the standard has a mean moisture content 88 % that of the average of the 30 mins of air sample either side of the standard (on average 0.02 % $H_2O$ lower). The moisture content of the standard also decreases slightly during the 20-minute measurement period as the dry standard air dries out the Nafion® membrane. The size of this decrease

5  is dependent on the moisture content of the prior air samples with larger decreases during the more humid times of the year. As a worst-case example, the change in the water content of the Heathfield standard during each run of August 2014 is shown in Figure S6. This shows a maximum drift of 0.07 % $H_2O$ equating to 30 % of the mean moisture content of air observations collected 30 minutes either side of the standard.

In contrast, due to the time taken to take replicate measurements of the calibration cylinders (at least 240 mins) only the first

10  20-minute measurement block of each calibration cylinder is significantly humidified. with the water content of the calibration measurement dropping rapidly to < 0.02 % $H_2O$ (10 to 20 % of the typical ambient air measurements). However, the exact level of humidification varies with ambient humidity and temperature. As such, in an effort to maintain consistency between calibration runs all runs with > 0.02 % $H_2O$ were excluded from analysis."

Page 25, line 13 – Page 26, line 2: "The poor performance of the CRDS pressure sensor at low $H_2O$ mole fractions, 0.02 to 0.5

15  % $H_2O$, is not expected to be a large source of error for undried samples as the majority of these, 92 % of the BSD and HFD data, contain > 0.5 % $H_2O$. But this could be a source of error for Nafion® dried samples where low moisture contents are typically obtained. However, for this study, where 95 % of HFD and 92 % of BSD Nafion® dried samples contain < 0.5 % $H_2O$, this effect is expected to be substantially mitigated by the humidification of the daily standard. As described earlier (Section 2.3.9 and Figure S1) the moisture content of the daily standard closely tracks that of the ambient air with the standard

20  mean moisture content almost 90% that of the ambient air. Hence the bulk of the error in the $H_2O$ correction at lower water contents should be accounted for during the drift correction process.

In contrast, without the humidification of the standard the error when Nafion® drying may well be significant. It is difficult to quantify this error, as it will vary with sample water content and the sensitivity of the individual instrument's pressure sensor to low $H_2O$ mole fractions. However, assuming that the residuals of the droplet water tests are an accurate reflection of the

25  likely error (Figure S8), we expect there to be a systematic offset of the order of -0.05 to -0.1 $\mu$mol mol$^{-1}$ $CO_2$ and -1 to -2 nmol mol$^{-1}$ $CH_4$. Assuming that a 90 % match in sample and standard moisture content equates to a 90% reduction in offset then we can estimate the offset in the BDS and HFD data as between 0.005 and 0.01 $\mu$mol mol$^{-1}$ $CO_2$ and -0.1 to -0.2 nmol mol$^{-1}$ $CH_4$, negligible in comparison to the WMO reproducibility guidelines."

30  *2. A more detailed discussion of the advantages and disadvantages of the two drying techniques namely the advantage to the Nafion® based drying system in terms of minimising systematic error and practicality and the use of an annually determined water correction for samples with $H_2O$ < 2%.*

While we have incorporated the reviewer's suggestion of additional discussion of the pros and cons of the two drying systems and the use of an annually determined water correction for samples with $H_2O < 2\%$ we politely disagree with their conclusion that the Nafion® based drying system is superior. Instead, we would contend that both approaches have advantages and the choice of method should be based primarily on the humidity and accessibility of the site in question. We have broadened the discussion in the conclusions section to make this apparent.

[revised manuscript text omitted]

3. *A discussion of the significance of the systematic errors in terms of the goals of the UK tall tower network and a conclusion in terms of the optimum drying process for the network.*

Page 32, lines 1 - 8: "While these errors are significant relative to the WMO internal reproducibility goals they are for the majority of observations smaller than the extended WMO measurement compatibility goals (± 0.2 µmol mol$^{-1}$ $CO_2$ and ± 5 nmol mol$^{-1}$ $CH_4$). It is also important to note that they are orders of magnitude smaller than baseline excursions observed at the sites (see Figures 4 & 5). They are also a factor of 10 smaller than the $CH_4$ model-data mismatch within the UK DECC network as estimated by Ganesan et al. (2015) at ~ 20 nmol mol$^{-1}$. Considering this difference, it is highly unlikely that, without significant improvement in modelled atmospheric transport, the systematic errors reported here would significantly alter estimates of UK-scale GHG fluxes or impede national emissions verification efforts."

Page 31, lines 22 - 26: "Considering the relatively narrow humidity range observed at Bilsdale and Heathfield, with no observations > 2.4 % $H_2O$ and > 95 % of observations > 0.5 % $H_2O$ (> 99 % > 0.35 % $H_2O$) and the relative remoteness of the locations the decision to remove the Nafion® based drying systems and rely on the annual empirical water correction appears justified. In contrast, at other more easily accessible or more humid sites the use of a Nafion® based drying system may be more advantageous."

4. *A reemphasis of the importance when using Nafion® drying systems of drying the calibration and standards gases in an identical manner to the sample to account for any loss of $CO_2$ across the membrane.*

This has been highlighted on:

Page 31, lines 3-4: "As such, when using a Nafion® based drying method it is essential that the calibration and standard gases are dried in an identical manner to the samples."

Page 31, lines 20-21: "Again this re-emphasises the importance of treating the sample, standards and calibration gases in an identical manner."

Specific comments:

page 4, line 19: uncertainties related to other components (such as?) sample collection

Page 4, line 19: The text "such as" has been inserted in to the sentence

page 7, line 2: Nafion drying performance is a strong function of temperature (see product literature). How much did nafion membrane temperature vary seasonally and/or diurnally. Might nafion temperature be the driver of output sample humidity variability rather than input sample humidity? In order to understand the extent to which water correction systematic errors are calibrated out, it would be useful to include a figure showing the extent to which humidified calibration gas air typically resembles dried sample air (and/or worst case difference between sample and standard humidity as measured by the CRDS). To what extent do daily drift correction calibrations capture variations in humidity at the CRDS? (see Figure 7 of Andrews et al., https://www.atmos-meas-tech.net/7/647/2014/amt-7-647-2014.pdf)) If nafion output humidity is very smooth in time, then the daily drift corrections may entirely mitigate errors due to the CRDS water correction. If this is the case, then the original configuration with the nafion would seem to be the superior method, since instrument-specific and time-dependent h2o corrections would be unnecessary. What can you say about the extent to which variations in the dried sample air humidity drive day-to-day drift in the CRDS signal during the period when the nafion was installed?

See the response to point 1 of "Overall framing" (above) and alterations/additions to the text found on Page 7, lines 3 – 7, Page 7, lines 13 – 15, Page 15, line 23 – Page 16, line 13 and Page 25, line 13 – Page 26, line 2 as given above.

page 8, line 10: wet/mean(dry)?

Page 8, line 20: This has been corrected to mean "wet"/"dry"

page 9 line 12: Here, dry air from four cylinders with varying CO2... was humidified.

Page 9, line 22: Corrected

page 9, line 22: An experiment was designed *to* observe

Page 11, line 20: Corrected

page 9, line 24: Not sure why it is necessary to mention the series of inconclusive experiments

Page 11, line 18: Sentence removed.

5  page 11, line 15, "all components of the cylinder air path between the DPG and the multiport valve, excluding the water trap, and the pump"

--> does the water trap refer to the cryotrap? This is confusing because according to the text and the drawing neither the cryotrap nor the pump are located between the DPG and the multiport valve.

Page 10, line 24: We thank the reviewer for noting this. It has been corrected.

Page 12, equation (1), it seems unnecessary/trivial to define TrueCP = CPin. I can't think of a reason to suspect that it would be otherwise.

Page 13, equation (1), we can understand the perspective of the reviewer however for completeness and to maintain

15  consistency in the notation we would prefer to leave the equation as it stands.

Page 13, line 12, "These experiments assume that any changes in the $CO_2$ or $CH_4$ mole fraction are driven solely by the Nafion drying process..."

--> Is it not the case that the experiment was designed to isolate the impact of the other possible sources of error or bias that

20  are identified, i.e. I think the setup aims to measure differences that are solely due to the Nafion. Maybe I do not understand how the data are analyzed.

Yes, experiment 2.3.7 was designed to isolate the differences due solely to the Nafion®. However, as with all experiments there is the possibility that small biases or errors may occur. This section explores these. The opening paragraph has been rewritten to clarify this.

25  Page 14, lines 4-8: "While experiments 2.3.5, 2.3.6 and 2.3.7 were designed to isolate key processes, for example experiment 2.3.7 examined changes in the $CO_2$ or $CH_4$ mole fraction driven by the Nafion® drying processes other possible sources of error or bias may exist. These include, adsorption and desorption effects within the regulator and walls of the tubing, gas solubility within the condenser of the dew point generator and instrumental drift."

30  page 14, line 4: dependent, not depended

Page 14, line 25: Corrected

page 14, line 25: according to Stanley et al 2018, the daily standard is measured for 20 minutes. Please state that here also. Is the standard always measured at the same time of day? Or does it run on e.g. a 21 or 23 hour cycle so that it rotates throughout the day in order to check for sensitivity to room temperature variations or similar.

5   Page 15, line 20: Yes, the daily standard is measured for 20 minutes. This information has now been included in the text. This is conducted at a fixed time of the day. However, that time of day is manually adjusted sometimes to allow for instrument servicing or when specific events (e.g. flight over passes) are taking place. The suggestion of adjusting it to a 21- or 23-hour cycle is interesting and something to consider.

10   page 15, line 16: Can you rule out the worst case scenario of -0.155 ppm bias (i.e. based on the likely fraction of added industrial air in your standards)? Possible bias of -0.155 is obviously quite large compared to other systematic errors related to water that are the focus of this paper. Why has there not been more effort to quantify this possible source of bias for these installations? It could be easily addressed by having the cylinders measured for 13CO2. Perhaps the labs that supplied the calibration gases could provide information about the isotopic abundance of their spiking/dilution cylinders so that you could
15   make a specific estimate of the errors (and correct them) for each set of unique calibration gases used at the sites. Given the amount of verbiage devoted to the 0.02 ppm nafion error above, the authors seem remarkably unconcerned about the possible impacts of isotopic composition of the standards on the reported ambient CO2 values.

Due to the, already considerable, length of the paper and its focus on the drying strategies a more detailed discussion of the impact of the isotopic composition of the calibration gases will be included in a future paper.

20   Page 32, Lines 12-14: "The development of a full uncertainty analysis incorporating such target tank measurements, along with an assessment of the calibration strategy, instrumental, water correction and sampling errors and errors induced by the isotopic composition of the calibration gases is also planned."

page 26, line 20, is the average standard deviation of the 15 min block means really 0.002 ppm, or is that the standard error
25   (Where the std dev has been divided by sqrt(N)).

It's the average standard deviation of the 15-minute block means not the standard error.

page 26: A nafion membrane can store quite a lot of water. I think the results of this test may depend on how the membrane was conditioned. A membrane that has been conditioned with ambient air at e.g. 2.5% humidity for several days may be
30   more permeable than a membrane that has been conditioned with dry air from a cylinder. The permeability of the membrane might also depend on its temperature. How might these tendencies map onto the dataset (i.e. how was the membrane conditioned prior to the experiment and how does the level of water stored in the membrane compare to what is expected in

the field in the summer during a high-humidity event)? Might the membrane be more permeable if it were equilibrated with high-humidity air). Perhaps the day-to-day variations of the ambient standard for the period when the nafion membrane was installed at the field sites might reflect changes in nafion permeability. It would be useful to repeat this experiment to explore whether permeability is impacted by the amount of water stored in the membrane (i.e. flow air with >2% h2o through the

5    nafion for several days prior to starting the test). Andrews et al. also reported permeability of nafion for co2 (page 652,653), but with a somewhat different configuration. They found cross-membrane transport of 0.1 ppm $CO_2$ for a partial pressure gradient of ~1700 hPa - 265 hPa = 1453, which is approximately proportional to your finding of 0.02 ppm co2 difference for a ~400 hPa gradient.

Indeed, as reported in other papers the permeability of the Nafion® is can be influenced by temperature and moisture content

10    along with membrane age and other factors (e.g. Naudy et al. (2014);Collette et al. (2009);Ma and Skou (2007)). However, as shown by Ma and Skou (2007), who compared the permeability of a water soaked and water equilibrated membrane, the moisture content dependence is small, typically of the order of the variability of the permeability measurement (see Figure 4, Ma and Skou (2007)). While comparisons between the wet membranes and a completely dry Nafion® showed significant differences in $CO_2$ permeability (c.f. Figure 4 and Figure 5, Ma and Skou (2007)) this dry membrane was prepared for 48

15    hours in a vacuum oven at 40°C and is, therefore, not representative of the laboratory or field conditions described in this paper. Similarly, considering the expected range of temperatures (20 – 40°C) experienced by the Nafion® membranes deployed in our experiments the temperature driven shifts in permeability observed by Ma and Skou (2007) were far smaller than the measurement precision. As such, while future experiments looking into the effects of these parameters may be of interest we consider them outside the scope of the current article.

20    Page 28, line 3: I don't agree with the current wording. So long as calibration gases are passed through the nafion then no significant bias is introduced by this implementation of the nafion drying method. The rest of the paragraph is fine, but I think the opening sentence is confusing.

Page 30, line 25 – Page 31, line 3: The paragraph has been reworded, now reading "However, care must be taken with the implementation of the method as, for samples with water content > 0.7 %, an additional 0.02 µmol mol⁻¹ of $CO_2$ was lost

25    from the sample across the membrane. Even samples as dry as the calibration gases were affected by this loss, although to a smaller degree (~ 0.015 µmol mol⁻¹ for $H_2O$ < 0.0001 %) as residual moisture remained in the membrane. However, as in this application of the method the calibration gases are dried in an identical manner to the samples, this loss is mostly calibrated out with only a very small (≤ 0.005 µmol mol⁻¹) constant residual bias of the order of the instrumental precision."

30    Page 28, line 15: I am confused about how these data are being post-processed to apply the calibration data from the field analyzer. The text says: "Altering the drying method to better match the moisture content of the calibration gases to the sample may minimise this error." For the early period where the nafion was deployed, is it not the case that the nafion humidified standards and the nafion-dried ambient air samples have the same humidity by the time they enter the CRDS? Or does the

nafion     membrane     dry     out     significantly     during     the     20-minute     calibration     interval.
The Nafion® membrane dries out only slightly during the 20-minute measurement of the standards, with the standard and the ambient air samples having very similar moisture contents.

Page 15, line 28 – Page 16, line 1: "On average the standard has a mean moisture content 88 % that of the average of the 30 mins of air sample either side of the standard (on average 0.02 % $H_2O$ lower)."

Page 28: In this conclusions section, it would be helpful to make a clear distinction between the early period where the nafion was installed versus the later period with no nafion. It seems that the systematic errors associated with the h2o correction are likely substantially larger during the later period. Also, it would be appropriate to include a brief discussion/summary of the pros and cons of each method, including complexity of implementation for the latter period when annual water calibrations for each crds analyzer were used but evidently with little reduction in uncertainty (given the relatively large variability seen in the daily and weekly lab water corrections.)

See the response to point 2 of "Overall framing" (above) and additions/alterations to the text found on Page 30, Line 10 – Page 31, Line 26.

page 28, line 20: How are you computing the max errors reported in Table 5? It looks like they are perhaps based on the residuals plots shown in Fig S5. But how are you accounting for uncertainty in the annual h2o correction itself? For cases such as BSD 2016 vs 2017 in Fig 8a, there are large differences between subsequent H2O corrections. Given that and the weekly variability for UoB in the weekly tests, how do you estimate the errors associated with assuming that a single realization of the H2O correction equation is valid on a particular day?

The maximum errors reported in Table 5 are an estimate based on both the residuals plotted in Fig S5, which represent an estimate of the error at a specific point in time, while the temporal uncertainty in the annual $H_2O$ correction for each site is estimated from the absolute difference between consecutive annual water tests. Specifically, we combine the maximum residual-based error with the drift in the $H_2O$ correction in quadrature for the given water content range. This is a worst-case scenario estimate. The temporal uncertainty in the $H_2O$ correction could also be approximated by examining the variability in the weekly UoB $H_2O$ tests as suggested by the reviewer. Estimating this uncertainty using the standard deviation of the weekly runs gives, to a single significant figure, the same uncertainty estimates as those in Table 5. A summary of this method is now included in the text (Page 31, lines 8-10).

page 28, line 27: "While drift in the instrumental water correction typically small it is important that it is identified and accounted for through regular water tests." This could be clarified and elaborated upon. My reading is that the annual h2o tests don't seem to provide much reduction in uncertainty, given the results in Fig 8e and f versus Fig 8a and b. Weekly tests would

apparently provide little benefit given that the week to week scatter is comparable to the differences among the annual tests. I do think it's a good idea to check the h2o correction at least annually if possible, but the droplet test has documented problems and returning an instrument to the lab for a more rigorous test could lead to lengthy data gaps. Given that the nafion errors are quite small based on tests done to date, it would be useful to include a few sentences about whether they should be reinstalled in the future.

Due to the restructure of the conclusion the line referred to here has been removed, however, we completely agree with the reviewer that an increase in the frequency of the water correction tests above that of an annual check is unlikely to significantly reduce the uncertainty, particularly for moisture contents between 0.5 and 2.5% $H_2O$. As discussed in the text rather than attempting to capture the temporal variability of the water correction we suggest that:

Page 26, Lines 23-25: "In light of the temporal variability of the water correction over time at higher water contents for sites with high humidity (> 2 % $H_2O$) using a Nafion® dryer or alternative drying method to obtain a relatively low and stable sample water content would be an advantage."

We also note that:

Page 27, Lines 2-8: "…within the bounds of the data typically observed at the BSD and HFD sites, the use of annually derived instrument specific water corrections are sufficient. This may not be the case at sites with higher levels of humidity and $CO_2$ and $CH_4$ mole fractions where water corrections may need to be determined more frequently, perhaps even weekly. The impracticality of such a frequent testing regime along with the apparent unreliability of the droplet test at $H_2O$ > 2.5 % (for example Figure S8g) mean that an alternative method, possibly partial drying, or a higher level of uncertainty may need to be applied to measurements made at higher water contents."

Finally, while the systematic errors of the Nafion® drying approach appear small there are no plans to reinstate these drying systems. As:

Page 31, Lines 22-25: "Considering the relatively narrow humidity range observed at Bilsdale and Heathfield, with no observations > 2.4 % $H_2O$ and > 95 % of observations > 0.5 % $H_2O$ (> 99 % > 0.35 % $H_2O$) and the relative remoteness of the locations the decision to remove the Nafon® based drying systems and rely on the annual empirical water correction appears justified."

page 29: line 6: Perhaps reprocessing of the data to implement a post-hoc water correction to remove/mitigate the systematic and humidity dependent errors that are evident in Fig S5 should be considered. Such a correction would be different for the nafion and no-nafion periods. Perhaps no correction would be required for the initial period with the nafion driers.

This is indeed an interesting option for future work. We have added the following text to the conclusions in an effort to incorporate this suggestion.

Page 32, Line 15 – 17: –" Further work to fully characterise the humidity dependent error in the water correction of each instrument, like that of (Reum et al., 2018), possibly using a piecewise post hoc correction, would also be beneficial in an effort to reduce the estimated error associated with the observations."

5    Figure 3: define DPG in the legend or caption.

Figure 3: Done

Fig 4: Have these been filtered? It is surprising that 42m $CO_2$ is not significantly higher than upper levels during summer night time.

10    Fig 4: These have not been filtered. As can be seen in the diurnal cycle shown in Figure 6 the 42m intake (blue) is often 1-2 ppm higher than the higher intakes during summer night time periods.

Fig 5: caption refers to mean diurnal cycle, but that seems to appear in Fig 6.

Fig 5: Corrected

Fig 6, 7. Figures are hard to read due to small size and overlapping symbols/bars. Maybe it is not practical to show figures for all molecules in the body of the paper.

We completely agree with the reviewer that the figures are difficult to read. However, as we feel that all gases are of interest rather than removing some from the figures we have instead adopted a simpler plotting strategy. This new strategy, based on

20    that of Satar et al. (2016), presents diurnal and seasonal cycles based on the means of trimmed linearly detrended data. These plots are far easier to read and have the advantage of retaining all the gases of interest.

Page 19, line 18 - 23: - Updated to describe the new plotting strategy.

"The long-term trend in mole fraction at each site was estimated as the mean linear trend in the minute mean data over the period 2014-2017, inclusive. Seasonal and diurnal trends in the data were assessed using monthly and hour-of-day means of

25    trimmed detrended minute-mean data developed using the Python numpy package. Here the long-term trend was removed by using a least-squares fit between a quadratic and the minute mean data. The data for each hour (or month for the seasonal plots) were trimmed following the approach of Satar et al. (2016) who removed the highest and lowest 5% of all data points."

New Figures 6 & 7

Figure 8: references to the individual panels in the caption are awkward and confusing. Also, (c & d) look to be daily and (e &f )look to be weekly, which seems to be opposite from the caption

Figure 8: The caption has been clarified now reading "The change with water in the difference in $CO_2$ and $CH_4$ dry mole fraction between: the first annual mean instrument specific water correction and subsequent annual corrections (a & b); the first individual water correction and subsequent daily corrections (c & d) and the first individual water correction and subsequent weekly tests (e & f). The daily and weekly tests were conducted using only the UoB instrument while the annual tests were conducted using all three instruments."

Figure 10: I don't understand the point of Figure 10 b & d, which seem to show only that the cryogenic drier was working.

Figure 10 b and d and references to these figures have been removed. Figure 10 c has been relabelled as Figure 10 b.

Supplement page 1, line 16: error in stated range for F3 0.5 - 0.5 L per min

Corrected to $0.1 – 0.5$ L $min^{-1}$.

Figure S1 caption: instead of writing "TOC" in the caption, consider "writing gas generator used to supply the counterpurge flow"

Caption changed. It now reads "Figure S1: (a) $CO_2$, (b) $CH_4$, (c) CO and (d) $H_2O$ mole fractions of the gas generator used to supply the Nafion® counter purge flow. Note the HFD gas generator was powered up just before to analysis in contrast to the other sites where the TOCs had been running for at least 12 hours prior to analysis."

Figure S5g and Figure S6: What is the explanation for the negative values at high water in the CO2 water correction residuals at U of Bristol but not seen at the field sites?

While much less common at sites other than the University of Bristol (UoB) negative residual values are evident at high water contents for both the Heathfield and Bilsdale sites (see Figure S8 a, c and e). We expect they appear more common at UoB as the data shown in Figure S9 g for UoB is from 13 droplet tests compared to a maximum of three for the other sites.

Figure S5: Given that the residuals show similar structure for most sites/years (i.e. CO2 and CH4 residuals are negative at low H2O), could the water correction be improved by using a piecewise correction or a post-hoc correction to remove the systematic residuals that are typical with the current approach?

This is indeed an interesting option for future work. We have added the following text to the conclusions in an effort to incorporate this suggestion.

Page 32, Line 15 – 17: –" Further work to fully characterise the humidity dependent error in the water correction of each instrument, like that of (Reum et al., 2018), possibly using a piecewise post hoc correction, would also be beneficial in an effort to reduce the estimated error associated with the observations."

5    Collette, F. M., Lorentz, C., Gebel, G., and Thominette, F.: Hygrothermal aging of Nafion®, Journal of Membrane Science, 330, 21-29, http://dx.doi.org/10.1016/j.memsci.2008.11.048, 2009.
     Ganesan, A. L., Manning, A. J., Grant, A., Young, D., Oram, D. E., Sturges, W. T., Moncrieff, J. B., and O'Doherty, S.: Quantifying methane and nitrous oxide emissions from the UK and Ireland using a national-scale monitoring network, Atmos. Chem. Phys., 15, 6393-6406, 10.5194/acp-15-6393-2015, 2015.
10   Ma, S., and Skou, E.: $CO_2$ permeability in Nafion® EW1100 at elevated temperature, Solid State Ionics, 178, 615-619, dx.doi.org/10.1016/j.ssi.2007.01.030, 2007.
     Naudy, S., Collette, F., Thominette, F., Gebel, G., and Espuche, E.: Influence of hygrothermal aging on the gas and water transport properties of Nafion® membranes, Journal of Membrane Science, 451, 293-304, dx.doi.org/10.1016/j.memsci.2013.10.013, 2014.
15   Reum, F., Gerbig, C., Lavric, J. V., Rella, C. W., and Göckede, M.: Correcting atmospheric $CO_2$ and $CH_4$ mole fractions obtained with Picarro analyzers for sensitivity of cavity pressure to water vapor, Atmos. Meas. Tech. Discuss., 2018, 1-29, 10.5194/amt-2018-242, 2018.
     Satar, E., Berhanu, T. A., Brunner, D., Henne, S., and Leuenberger, M.: Continuous CO2/CH4/CO measurements (2012–2014) at Beromünster tall tower station in Switzerland, Biogeosciences, 13, 2623-2635, 10.5194/bg-13-2623-2016, 2016.

**Reply to editors' comments**

Ann R. Stavert[1], Simon O'Doherty[2], Kieran Stanley[2], Dickon Young[2], Alistair J. Manning[3], Mark F. Lunt[5], Christopher Rennick[4] and Tim Arnold[4,5].

[1] Climate Science Centre, CSIRO Oceans & Atmosphere, Aspendale, VIC, 3195, Australia
[2] School of Chemistry, University of Bristol, Bristol, BS8 1TS, United Kingdom
[3] Met Office, Exeter, Devon, EX1 3PB, United Kingdom
[4] National Physical Laboratory, Teddington, Middlesex, TW11 0LW, United Kingdom
[5] School of GeoSciences, University of Edinburgh, Edinburgh, EH9 3FF, UK

The authors would like to thank the editor, Dominik Brunner, for his helpful comments. We have endeavoured to incorporate all the suggestions that he has made. The locations of changes to the paper are given as Page Number, Line number of the new paper while his comments contain the locations in the original paper. Our responses are shown here in blue text and the editors' comments in black.

Associate Editor Decision: Publish subject to minor revisions (review by editor) (04 Apr 2019) by Dominik Brunner
Comments to the Author:

Dear authors,

Thank you very much for the amendments to the manuscript and detailed responses to the reviewers. By dropping the inverse modelling part the publication has clearly gained in focus and conciseness.

Reviewer #1 reassessed your revised manuscript and, overall, was satisfied with the changes. However, the reviewer has a number of further comments I would like you to consider.

In particular, the reviewer would like to see more discussion of the advantages and disadvantages of active drying (original configuration) versus no drying but compensating for errors through frequent droplet tests (current configuration).

Following the suggestion of Reviewer #1 we have incorporated a fuller discussion of the advantages of an active drying configuration in the "Conclusions and future work" section of the paper. Specifically, we have added

[revised manuscript text omitted]

In addition, I read your manuscript carefully myself and have a few additional comments as detailed in the following:

Main points:

5  A new section 2.3.6 an "Assessing the CRDS water correction" has been added. Details of this experiment are reported in the supplement section S1 and Figure S3. Actually, the text in this supplement is partly identical to the text presented later in the main body of the manuscript in Section 2.3.7. I actually think, it would make more sense to present that text here in the new Section 2.3.6 and to shorten Section 2.3.7 accordingly. I thus propose the following changes:

1. Move most of the text currently in Supplement S1 to Section 2.3.6 "Assessing the CRDS water correction." Furthermore, I

10  don't see a reason for placing Figure S3 in the supplement but Figure 3 in the main body of the text. Why not combining the two into a single Figure 3 with panels a) and b). Like this one would immediately see the differences in the setups.

Page 9, line 20 to Page 11, 12 - As suggested by the editor the text in Section S1 has been moved to Section 2.3.6 and Figures 3 and S3 have been combined into a new Figure 3 in the main body of the paper. Figure S3 has been removed from the supplement.

2. Remove all parts of the text in Section 2.3.7, that would now be described under "Assessing the CRDS water correction". Actually, in Section 2.3.7 you could simply state that the experimental setup is very similar to 2.3.6, but with modifications XYZ.

As far as I can see, those parts in Section 2.3.7 that could be dropped to avoid repetition include P10, Lines 10-17, P11,

20  Lines13-19, P12, Lines 7-11.

As suggested by the editor we have removed the lines listed above.

3. Please also make clear that the same CRDS G2301 at UoB was used in all experiments. Currently, you state in 2.3.6 that "a Picarro G2301" was used, but it remains unclear whether this is the same as in 2.3.5.

25  This has now been made clear.

Page 9, lines 24: "…using the same Picarro G2301 CRDS used in Sections 2.3.5 and 2.3.7."

Page 11, lines 19-20: "… UoB laboratory Picarro CRDS as used in Sections 2.3.5 and 2.3.6"

Another concern I have is that the loss of CO2 across the Nafion membrane may not be quantified from the experiment as it

30  is done currently. The loss is currently quantified as 0.02 umol mol-1 (e.g. page 27, Line 7), based on the differences between CP_in and CP_out measurements in the experiment described in 2.3.7. However, if the flow rate (and pressure?) of the counter purge air is not the same as the flow rate of the sample, the change in mole fraction in the counter purge air is not identical to the corresponding change in the sample.

We thank the editor for this note, however we feel it is important to note that CRDS used in these experiments does not

sample the whole counter purge gas stream. Instead it takes a highly pressure-controlled subsample of the counter purge gas (See Figure 3). As such, while the native flow rates and pressures of the counter purge gas and air samples may differ the pressure of the gas within the CRDS sample cell is held at $140 \pm 0.05$ Torr. Hence, the pressures of the sample and counter purge are matched during measurement and therefore a change in the mole fraction of the counter purge air should be

5    identical to the change in the sample.

Small points:

- Page 7, line 1: Here it is stated that the Nafion drying results in air samples with water mole fractions between 0.05 and 0.2%. At the end of Section 2.3.1 (P6, L23), however, a value of <0.0002% $H_2O$ is mentioned for the samples with Nafion

10   drying. I have difficulties bringing these numbers together. In addition, how can the samples become dryer than the counter-purge air (which has humidity of <0.005% $H_2O$)?

The figure of 0.0002% $H_2O$ is taken directly from Young, 2007. This measurement was made of the GC-ECD Nafion using a Xentuar portable dewpoint meter. In comparison the reference on Page 7, line 1 (now Page 8, lines 4-5) is to the counter purge of the CRDS Nafion® and was made using the CRDS. Similarly, measurements of the counter purge air given in the

15   text were made using the CRDS. As such differences between these numbers may be due to calibration and sensitivity differences between the CRDS and the Xentuar portable dewpoint meter. It is also important to note that the counter purge flow rates differ significantly between the two, with the GC-ECD counter purge flow rate typically 5 times higher than the CRDS counter purge. The large difference in relative flow rates between the sample and counter purge of the GC-ECD Nafion® greatly increases the drying capacity of the counter purge air stream. The text has been altered to provide more

20   details.

Page 6, Lines 24 – 26: "Previous examinations of this drying method using a Xentuar portable dewpoint meter have found that samples are dried to dew points of around -40°C when using a counter purge at approximately -70°C (Young, 2007)."

- P7, L4: Change "membrane this drying" to "membrane, this drying"

25   Page 7, Line 10: Done

- P7, L3-5: Long sentence that is difficult to understand. Should be reformulated.

Page 7, lines 10-13: Reformulated to:

"The CRDS Nafion® drying systems were removed on the 30th of September 2015 & 17th of June 2015 at BSD and HFD,

30   respectively. Following this undried air was analysed and the data post corrected with an instrument specific water correction."

- P7, L14-15: Sentence "While the CO mole …" is incomplete.

Page 7, lines 23-14: Rewritten. Now reads:

"Similarly, the zero air CO mole fraction is significantly lower than the minimum CO mole fractions typically observed at the HFD and BSD sites (~ 60 nmol mol$^{-1}$)."

- P8, L5: Change "effected" to "affected"

Page 8, line 15: Corrected

- P8, L7: Change "(Rella et al. 2013)" to "Rella et al. (2013)"

Page 8, line 17: Corrected

- P8, L16: Change "for CO however this used" to "for CO. However, this used"

Page 8, line 26: Corrected

- P8, L23: Add comma in "measuring dry air, they often"

Page 9, line 5: Corrected

- P9, L7-8: Make clear that the "sub- and above ambient air mole fraction cylinders" refer to CO2 below or above ambient, not to H2O.

Page 9, lines 15-17: Now reads "A set of instrument specific water corrections was also determined in triplicate, using dried sub- and above ambient $CO_2$ and $CH_4$ mole fraction cylinders."

- P10, L2: Change to "cylinders, one of dry air near"

Page 10, line 20 – Corrected

- P10, L10: Why do you say "but"?

Sentence removed

- P10, L13 and later lines: The figure does not show any labels for the T-pieces. There is just no use in calling them T1, T2, etc. in the text.

These have been removed.

- P10, L20: delete "valve" after "Switzerland)"

Page 10, Line 11 - Corrected

- P12, L3: Is this where the third stage starts? Thus, wouldn't it make sense to state "As third stage, the multiport valve was

used to switch .."?

The role of the cryogenic trap in this third stage should probably also be described, or not?

Page 12, line 28 – Page 13, line 2: Corrected to read:

"At the start of the third stage, the multiport valve was used to switch between the $CP_{in}$ or $CP_{out}$ flows, measuring each for

5    repeated 20-minute blocks (n > 3) at each dew point (see Figure S3 red and blue). The $H_2O$ trap remained inserted in the silicone oil throughout the third stage."

- P13, L2: You mention that the difference between CP_in and CP_out is also affected by the water correction. Shouldn't you add ".. and the effect of the water correction AT LOW HUMIDITIES."?

10   Page 13, Line 22: Corrected.

- P13, L12-L28: To what extent are these factors specific for experiment 2.3.7 only? The question of regulator and tubing adsorption/desorption effects, for example, seems to be applicable to other experiments as well. Thus why discussing all this here?

15   We thank the editor for this note and have now broadened this section to encompass experimental assumptions relating to experiments 2.3.5, 2.3.6 and 2.3.7. The section is now a stand-alone section (2.3.8 Key experimental assumptions) rather than a subsection of section 2.3.7.

Page 14, lines 4-8: "While experiments 2.3.5, 2.3.6 and 2.3.7 were designed to isolate key processes, for example experiment 2.3.7 examined changes in the $CO_2$ or $CH_4$ mole fraction driven by the Nafion® drying processes other possible sources of

20   error or bias may exist. These include, adsorption and desorption effects within the regulator and walls of the tubing, gas solubility within the condenser of the dew point generator and instrumental drift."

- P15, L18: It is unclear how you came up with the number of "<0.04%".

This number was determined based on the typical ambient sample mole fraction of 400 µmol mol$^{-1}$ $CO_2$ and 2000 nmol mol$^{-1}$

25   $CH_4$ and error range of 0.001 – 0.155 µmol mol$^{-1}$ $CO_2$ and 0.1 – 0.7 nmol mol$^{-1}$ $CH_4$ as discussed in Griffith (2018). We have edited the text to make this more apparent.

Page 17, lines 2-4: "Based on this we expect a worst-case scenario estimate of the error associated with our typical ambient measurements to be < 0.04 % for both $CO_2$ and $CH_4$."

30   - P17, L4: I am somewhat confused about the frequency of calibrations. Here you talk about "daily measurements of 4 standard cylinders at each site", but I thought, that full calibrations were only done once a month.

Yes, full calibrations are completed only monthly. However, a single 20-minute measurement of a standard cylinder is made each day. As these standard cylinders are consumed over time 4 of these types of cylinders have been used at the HFD and

BSD sites, respectively, during the time period examined in this paper (i.e. 8 cylinders in total). We have rewritten the text to clarify this.

Page 18, lines 17-20: "The long-term reproducibility of a 20-minute mean was estimated as the mean standard deviation of the daily 20-minute measurements of the standard cylinders used at each site. A total of 8 standard cylinders have been used in succession at the two sites with the usage periods and $CO_2$, $CH_4$ and $CO$ mole fractions listed in Table 3."

- P17, L5: Should "repeatability" be changed to "reproducibility" also here?

Page 18, line 20: Corrected

- P18, L17-19: Strange sentence. Should probably be reformulated (and split in 2 or more sentences).

Page 20, Lines 11-13: Reformulated to read: "In contrast, events that do not show corresponding high CO and $CH_4$ mole fractions tend to occur in the higher two intakes. As such, they are likely to be driven by more remote $CO_2$ sources, for example power plants."

- P18, L13: Delete "suggesting a change in the background mole fraction" as this basically repeats what was said in the same sentence before.

Deleted.

- P19: Discussion of the vertical gradients and diurnal profiles at the towers: You could/should discuss your results also in the context of other tall tower studies, notably Winderlich et al. (2010; https://www.atmos-meas-tech.net/3/1113/2010/amt-3-1113-2010.pdf) and Satar et al. (2016; https://www.biogeosciences.net/13/2623/2016/), especially their Figure 3, which also shows the seasonal evolution of diurnal profiles of CO2, CH4 and CO at different vertical levels similar to your Figure 6.

References to these papers have now been incorporated:

Page 20, line 20 to Page 21, line 10: "Both sites show a clear relationship between $CO_2$ mole fraction and intake height with the lowest height generally having the most elevated mole fractions, followed by the higher heights (Figures 4a & 5a). This trend, also apparent for $CH_4$ and CO (Figures 4b & c and Figure 5b & c), is typical of tall tower measurements and is driven by proximity to surface sources (Bakwin et al., 1998;Winderlich et al., 2010;Satar et al., 2016). This gradient in $CO_2$ and $CH_4$ mole fraction is most apparent in the warmer seasons and during the early hours of the morning (Figure 6a, b, c & d) when the boundary layer is the lowest, a trend observed previously by Winderlich et al. (2010). While a reversal of this gradient, lower heights having lower $CO_2$ mole fractions, occurs in the middle of the day (Figure 6 a and b). As described in Satar et al. (2016) this decrease in near surface $CO_2$ is most likely driven by local photosynthetic activity. Interestingly, this trend is also apparent in spring, summer and autumn $CH_4$ mole fractions at BSD (Figure c) but not HFD (Figure 6d). This suggests a midday sink of $CH_4$ local to BSD but not HFD. Considering that BSD is located high in the Yorkshire moors (379.1 m a.s.l) while HFD is

located in a lower agricultural region (157.3 m a.s.l) a large difference in soil moisture, and therefore methanotrophic activity (Topp and Pattey, 1997), between the two sites is possible."

- P19, L13: change ". Suggesting" to ", suggesting"

Page 21, lines 15-16: Corrected

- P20, L7: Saying that this comparison is "meaningless" is a bit far stretched. I wouldn't expect the effect of using different calibration scales to be so large to make a comparison meaningless.

Page 22, line 9: Changed to "difficult to interpret".

- P20, L17: I guess that agriculture is the main source of N2O. I wouldn't call this a "point source".

Page 22, line 19: Changed from "local point sources" to "local sources".

- P21, L2: Change to "In contrast, in a UK focused inversion study Ganesan et al. (2015) found that .."

Page 22, line 4: Corrected

- P22, L2: I think it would be useful to say again how the instrument specific water correction was made. E.g. with "The annual instrument-specific water corrections, determined through regular droplet tests, are typically very similar … "

Page 23, line 4: Corrected

- P24, L26: Change to "may need to be applied"

Page 25, line 7: Corrected

- P25, L24: Change to "The full range of H2O".

Page 28, line 5: Corrected

- P28, L17: The role of the "cell pressure sensors" will not be clear to a reader reading only the conclusions. Probably you should add a reference to Reum et al. here.

Page 31, lines 14-15: Corrected

- P28, 27: Change to "While the drift in the instrumental water correction is typically small, it is .."

Sentence removed

- Caption of Figure 8: I think weekly and daily are mixed up here in the last line of the caption.

Corrected

[revised manuscript text omitted]

**8 Supplementary**

[Figure]

Figure S1: The minute mean water observations for (a) Bilsdale and (b) Heathfield. Blue, green and purple data points represent the Bilsdale 42m, 108m and 248m intakes and the red and yellow data points represent the Heathfield 50m and 100m intakes, respectively. The black data points are the 20min mean of the daily standards. The dashed grey line indicates when the TOC system at the Bilsdale site was repaired. The solid grey line indicates when the Nafion® drying systems were removed from the sites.

[Figure]

**Figure S2: Seasonal and diurnal patterns in minute mean water observations. Blue, green and purple represent the Bilsdale 42m, 108m and 248m intakes and red and yellow represent the Heathfield 50m and 100m intakes, respectively. Figure (a) and (c) show a full year's worth of data at Bilsdale, with and without the Nafion® drying system, respectively. Similarly, Figures (e) and (g) show a full year's worth of data at Heathfield, with and without the Nafion® drying system. Figures (b) and (d) show week-long periods of data in summer (dashed lines) and winter (solid lines) at Bilsdale with and without the Nafion®, respectively. Similarly, Figures (f) and (h) show week-long periods of data in summer (dashed lines) and winter (solid lines) at Heathfield with and without the Nafion®, respectively.**

[Figure]

**Figure S3: (a) CO₂, (b) CH₄, (c) CO and (d) H₂O mole fractions of the gas generator used to supply the Nafion® counter purge flow. Note the HFD gas generator was powered up just before to analysis in contrast to the other sites where the TOCs had been running for at least 12 hours prior to analysis.**

In brief, the output of the cylinder regulator was plumbed to the input of the DPG. A T-piece connected prior to the DPG input vented any excess gas via a flow meter (F1, Fig. S2) ensuring that the DPG input remained at close to ambient atmospheric pressure throughout the experiment. The output of the DPG passed through a second T-piece with the over flow outlet also connected to a flow meter (F2) to ensure that the CRDS input pressure remained near ambient. A third flowmeter (F3) was placed on the outflow of the Nafion counter purge. Flow meters F1 and F2 had a range of 0.1–1 L min⁻¹ (VAF-G1-05M-1, Swagelok, UK) while F3 had a smaller flow range 0.5–0.5 L min⁻¹ (FR2A12BVBN-CP, Cole-Palmer, USA). Typical output flows were 0.1, 0.3 and 0.3 L min⁻¹ for F1, F2 and F3 respectively. After F2 the sample flow was further split using a T-piece, with half the flow passing through a cryogenic water trap before reaching a 4-port 2-position valve, V1 (EUDA-2C6UWEPH, VICI Valco AG International, Switzerland, actually a 6-port valve configured as a 4-port valve). The other half bypassed the water trap and connected directly to V1. One of the outputs of V1 went via the Nafion to a second identical valve, V2, while the second output went directly to V2. The first output of V2 connected directly to the input of the CRDS while the second connected to a pump (PICARRO Vacuum pump S/N PB2K966-A) set to a flow rate matching that of the CRDS (0.3 L/min) to ensure uniform flow through both branches of the system. These valves were controlled manually using a VALCO electronic controller and universal actuator.¶
The water trap consisted of a coil of ¼" diameter (I.D. 3.36mm) stainless steel tubing immersed in a Dewar of silicone oil (Thermo Haake SIL 100, Thermo Fisher Scientific, USA). The silicone oil was cooled using an immersion probe (CC-65, NESLAB) to less than -50 °C. Other than the water trap and two short sections (< 10 cm) of ¼" (O.D.) plastic tubing immediately prior to and post the DPG, 1/16 " stainless steel tubing was used throughout the system.¶
The experiment was conducted in a temperature-controlled laboratory at 19 °C, and thus, at temperatures lower than a number of the dew points used within the experiment. Hence, in order to avoid condensation forming on the walls of the tubing all components of the system between the cylinder, excluding the water trap, and the outputs of V2 were contained within a chamber heated to > 32 °C. Tubing between the heated chamber and the input of the CRDS was also heated with heating tape to > 32 °C while the internal temperature of the CRDS was > 32 °C throughout the experiment. ¶
Multiple measurement blocks of each sample treatment were conducted after a lengthy initial stabilisation period. This period allowed for the establishment of equilibrium between the water in the condenser block of the DPG and the sample gas and lasted at least 2 hours (sometimes up to 5 hours). The treatment blocks varied in … [2]

[Figure]

Cylinder

Syringe

Multiport valve

T-Piece with septum

—— 1/16" stainless steel tubing

━━ 3/8" Dekabon tubing

**Figure S4: Droplet test instrument specific water correction flow path and procedure**

[Figure]

(a)

(b)

(c)

**Counter purge
Prior to Nafion**

**Counter purge
After Nafion**

**Sample
Direct**

**Sample
Via $H_2O$ trap**

[Figure]

F1 F2

DPG

Cylinder —— Sample flow ·····

► Direction of flow

¶

**Figure S5: The (a) CO$_2$, (b), CH$_4$ and (c) H$_2$O minute mean data obtained during the Nafion® counter purge experiments for cylinder UoB-16 at a dewpoint of 10C. Error bars are ± 1 standard deviation of each minute mean. Purple and grey data points are the sample without and with the H$_2$O trap, respectively, while blue and red data points are the Nafion® counter purge before and after the Nafion®, respectively.**

[Figure]

**Figure S6: The change in the moisture content of standards measured at Heathfield during August 2014. Each colour represents a different standard run. The date of each run is given in the legend.**

[Figure]

**Figure S7: The (a) CO$_2$, (b), CH$_4$ and (c) H$_2$O m**inute mean data for 24 hours starting 16:00 11/3/2016 at Bilsdale. Blue, green and purple data points represent the 42m, 108m and 248m intakes, respectively.

[Figure]

**Figure S8: Droplet test residual, wet corrected data less dry mean, plots for Bilsdale (a) 2015, (b) 2016 and (c) 2017, Heathfield (d) 2015 and (e) 2016 and University of Bristol (f) 2015, (g) 2016 and (h) 2017. The instrument specific CO$_2$ residual values (red) are shown in the upper plots, CH$_4$ (blue) in the middle plots. And CO (orange) in the lower plots. The residuals of the factory determined water correction are also shown in grey. The mean ± 1σ$^2$ of the residuals are given for each plot for both the instrument specific (black) and the factory (grey) corrections.**

[Figure]

[Figure]

**Figure S9: Droplet test residual, wet corrected data less dry mean, plots for (a) below ambient cylinder and (b) the above ambient cylinder at the University of Bristol. The instrument specific CO$_2$ residual values (red) are shown in the upper plots and CH$_4$ (blue) in the middle plots. The residuals of the factory determined water correction are also shown in grey. The mean ± 1σ$^2$ of the residuals are given for each plot for both the instrument specific (black) and the factory (grey) corrections.**

| Page 41: [1] Deleted | Stavert, Ann (O&A, Aspendale) | 5/14/19 4:48:00 PM |
|---|---|---|

| Page 41: [1] Deleted | Stavert, Ann (O&A, Aspendale) | 5/14/19 4:48:00 PM |
|---|---|---|

| Page 41: [1] Deleted | Stavert, Ann (O&A, Aspendale) | 5/14/19 4:48:00 PM |
|---|---|---|

| Page 41: [1] Deleted | Stavert, Ann (O&A, Aspendale) | 5/14/19 4:48:00 PM |
|---|---|---|

| Page 41: [1] Deleted | Stavert, Ann (O&A, Aspendale) | 5/14/19 4:48:00 PM |
|---|---|---|

| Page 41: [1] Deleted | Stavert, Ann (O&A, Aspendale) | 5/14/19 4:48:00 PM |
|---|---|---|

| Page 41: [1] Deleted | Stavert, Ann (O&A, Aspendale) | 5/14/19 4:48:00 PM |
|---|---|---|

| Page 55: [2] Deleted | Stavert, Ann (O&A, Aspendale) | 5/28/19 3:55:00 PM |
|---|---|---|

---

## Author Response (AR3)

**Response to technical corrections**

**Ann R. Stavert1**

1 Climate Science Centre, CSIRO Oceans & Atmosphere, Aspendale, VIC, 3195, Australia *Correspondence to*: Ann R Stavert (ann.stavert@csiro.au)

5 Please note the suggested changes are written in plain font and the author's response in italics. Where the page/line number has changed between the corrected and original text the new page/line number has been given in parentheses following the author's response.

Page 7, Line 6 - Change to "TOC gas generator"

10 Changed

Page 7, Line 11 – Change "&" to "and" *Changed*

Page 9, Line 6 – I assume these values are in %?
Yes, the values are in %. These units have been inserted into the text.

Page 9, Line 20 - Maybe it would be clearer to say

"The accuracy of the CRDS water correction determined through the water droplet tests as described in Sect. 2.3.4 was assessed

**20**

.."

The text has been clarified as suggested.

Page 11, Line 25 – Insert "The" between "as" and "multiport". *Inserted*

**25**

Page 13, Line 15 – Why a minus sign? Isn't CP\_out higher than CP\_in because additional CO2 is transported from the sample to the counter purge air? *Changed to a plus sign.*

30 Page 13, Lines 16-19 – no line breaks, separate by commas Corrected Page 14, Lines 4-5 – Remove "for example... drying processes". *Removed*

5 Page 21, Line 9 – Change "Suggesting" to "This suggests". *Corrected*

Page 21, Line 10 – Remove "necessary" *Removed*

**10**

Page 24, Line 7 – Remove "instrument specific water" *Removed*

Page 24, Line 24 – Insert "with" between "data" and "H2O".

15 Inserted (Page 24, Line 18)

**UK greenhouse gas measurements at two new tall towers for aiding emissions verification**

Ann R. Stavert1, Simon O'Doherty2, Kieran Stanley2, Dickon Young2, Alistair J. Manning3, Mark F. Lunt5, Christopher Rennick4 and Tim Arnold4,5.

[revised manuscript text omitted]